# CMTM4 is a subunit of the IL-17 receptor and mediates autoimmune pathology

Daniela Knizkova[1,2,4], Michaela Pribikova [1,4], Helena Draberova[1], Tereza Semberova [1], Tijana Trivic [1], Alzbeta Synackova [1], Andrea Ujevic [1], Jana Stefanovic [1], Ales Drobek[2], Martina Huranova[2], Veronika Niederlova [2], Oksana Tsyklauri [2], Ales Neuwirth [2], Jolana Tureckova[3], Ondrej Stepanek [2]✉ and Peter Draber [1,2]✉

Interleukin-17A (IL-17A) is a key mediator of protective immunity to yeast and bacterial infections but also drives the pathogenesis of several autoimmune diseases, such as psoriasis or psoriatic arthritis. Here we show that the tetra-transmembrane protein CMTM4 is a subunit of the IL-17 receptor (IL-17R). CMTM4 constitutively associated with IL-17R subunit C to mediate its stability, glycosylation and plasma membrane localization. Both mouse and human cell lines deficient in CMTM4 were largely unresponsive to IL-17A, due to their inability to assemble the IL-17R signaling complex. Accordingly, CMTM4-deficient mice had a severe defect in the recruitment of immune cells following IL-17A administration and were largely resistant to experimental psoriasis, but not to experimental autoimmune encephalomyelitis. Collectively, our data identified CMTM4 as an essential component of IL-17R and a potential therapeutic target for treating IL-17-mediated autoimmune diseases.

Interleukin-17A (IL-17A) is a proinflammatory cytokine produced by CD4[+] T helper 17 cells (T$_H$17 cells) and several other immune cell types[1–3]. IL-17A amplifies inflammation by inducing the production of various proinflammatory cytokines. It is a fundamental component of the immune defense against fungal and bacterial infections. Patients deficient in the IL-17A signaling pathway suffer from recurrent candidiasis and staphylococcal infections[4–7]. On the other hand, dysregulated IL-17A production and/or signaling can trigger chronic inflammation and tissue damage, eventually leading to the development of autoimmune disorders or malignant transformation[8,9].

The IL-17 receptor (IL-17R) is composed of two subunits: the ubiquitously expressed IL-17RA and IL-17R subunit C (IL-17RC), whose expression is relatively low and restricted to nonhematopoietic cells[10–12]. In unstimulated cells, IL-17RA and IL-17RC are spatially separated on the cell surface. Stimulation of cells with dimeric IL-17A triggers concomitant binding of IL-17RA and IL-17RC to the same IL-17A dimer, crosslinking of these two receptors and formation of the IL-17R signaling complex (IL-17RSC)[13]. Both IL-17RA and IL-17RC contain a SEFIR domain in the cytoplasmic tail. Dimerization of IL-17RA and IL-17RC within IL-17RSC enables recruitment and oligomerization of cytoplasmic adapter ACT1, which also contains a SEFIR domain. ACT1 subsequently recruits the trimeric nondegradative ubiquitin ligase TRAF6. TRAF6-induced formation of K63-polyubiquitin chains within IL-17RSC promotes recruitment of various signaling components, such as linear ubiquitin ligase LUBAC, the NEMO-IKKα/β complex and the TAK1-TABs complex, which triggers activation of the transcription factor NF-κB and the protein kinase MAPK signaling pathways and induction of cellular responses. Ubiquitin linkages also recruit regulatory proteins, such as the deubiquitinase A20 or the kinases TBK1 and IKKε, which promote disassembly of the IL-17RSC and prevent hyperactivation of IL-17A-induced responses[14,15].

IL-17RC is essential for cell responsiveness to IL-17A, and also to the related cytokine IL-17F, or the IL-17A-IL-17F heterodimer[16].

[1]Laboratory of Immunity & Cell Communication, BIOCEV, First Faculty of Medicine, Charles University, Vestec, Czech Republic. [2]Laboratory of Adaptive Immunity, Institute of Molecular Genetics of the Czech Academy of Sciences, Prague, Czech Republic. [3]Czech Centre for Phenogenomics and Laboratory of Transgenic Models of Diseases, Institute of Molecular Genetics of the Czech Academy of Sciences, Vestec, Czech Republic. [4]These authors contributed equally: Daniela Knizkova, Michaela Pribikova. ✉e-mail: ondrej.stepanek@img.cas.cz; peter.draber@lf1.cuni.cz

Therefore, modulating the amount of IL-17RC on the surface of the cells might represent an attractive therapeutic target for the treatment of IL-17-mediated diseases. Here we identified CMTM4 as a new component of the IL-17RSC and we showed that CMTM4 was crucial for plasma membrane localization of IL-17RC and subsequently, for IL-17A and IL-17F signaling.

## Results

### CMTM4 is constitutively associated with IL-17RC

To identify new regulators of IL-17A signaling, we established a methodology for mass spectrometry (MS) analysis of the IL-17RSC[15]. We stimulated mouse stromal ST2 cells with recombinant Strep-Flag-tagged mouse IL-17A (SF-IL-17A) at a concentration of 0.5 µg ml$^{-1}$, at which slightly less than half of IL-17 receptors on the cell surface were occupied (Extended Data Fig. 1a,b). This concentration of SF-IL-17A induced a strong signaling response in ST2 cells, which was comparable to that induced by commercial recombinant IL-17A (Extended Data Fig. 1c). Subsequently, the entire IL-17RSC was isolated by tandem affinity purification of SF-IL-17A, followed by MS detection. Analysis using intensity-based absolute quantification (iBAQ)[17] found that the transmembrane protein CMTM4 was a component of the IL-17RSC (Fig. 1a and Supplementary Table 1). CMTM4 belongs to a family of eight structurally similar proteins, all containing four transmembrane helices. To test whether CMTM4 bound directly to the transmembrane receptors IL-17RA and/or IL-17RC, we stably expressed SF-IL-17RA or SF-IL-17RC in ST2 cells using retroviral transduction and we subjected the cellular lysates to anti-Flag immunoprecipitation (IP). SF-IL-17RC, but not SF-IL-17RA, coprecipitated with CMTM4 (Fig. 1b), indicating a constitutive interaction between them. CMTM4 did not interact with other members of the IL-17R family, namely IL-17RB, IL-17RD and IL-17RE (Extended Data Fig. 1d).

IL-17RA and IL-17RC are present as single transmembrane proteins in unstimulated cells and associate together as a part of IL-17RSC upon IL-17A stimulation[13]. The constitutive interaction between CMTM4 and SF-IL-17RC did not change upon IL-17A stimulation (Fig. 1c). In contrast, SF-IL-17RA became associated with IL-17RC and CMTM4 only upon IL-17A stimulation (Fig. 1d). In addition, CMTM4 coimmunoprecipitated with endogenous IL-17RC in both unstimulated and IL-17A-stimulated ST2 cells (Fig. 1e). Together, these data established that CMTM4 constitutively bound IL-17RC and both proteins became an integral part of the IL-17RSC upon IL-17A stimulation.

Next, we aimed to characterize the association between CMTM4 and IL-17RC. MS analysis of SF-IL-17RC-associated proteins confirmed the interaction between IL-17RC and CMTM4, while no other CMTM family members were detected (Supplementary Table 2). The quantification of IL-17RC-associated proteins through iBAQ indicated that CMTM4 was the most abundant IL-17RC interacting partner (Fig. 1f). The analysis of stoichiometry between IL-17RC and CMTM4 within IL-17RSC based on MS iBAQ values showed nearly equimolar interaction of these two proteins (Fig. 1g). In addition, when transfected in *Drosophila* S2 cells, which are devoid of IL-17 signaling components[18], Myc-tagged CMTM4 (Myc-CMTM4) associated with SF-IL-17RC, but not with SF-IL-17RA (Fig. 1h).

To map the interaction site between IL-17RC and CMTM4, we stably expressed various SF-IL-17RA and SF-IL-17RC chimeric proteins or empty vector (EV) in ST2 cells using retroviral transduction. SF-IL-17RC$^{T-mRA}$, in which the transmembrane domain of IL-17RC was exchanged with that of IL-17RA, which has substantially different amino acid composition (Extended Data Fig. 1e), did not bind endogenous CMTM4 (Fig. 1i), despite comparable surface expression to wild-type SF-IL-17RC (SF-IL-17RC$^{WT}$) (Extended Data Fig. 1f). In accord, while SF-IL-17RA did not bind endogenous CMTM4, the chimeric SF-IL-17RA$^{TmRC}$ protein, which harbored the transmembrane domain of IL-17RC, associated with endogenous CMTM4 (Fig. 1i). These data indicated that CMTM4 interacted with the transmembrane domain of IL-17RC.

We employed a similar approach to identify the IL-17RC interaction motif on CMTM4. Because SF-IL-17RC could not bind the CMTM4 paralog, CMTM6, we generated chimeric molecules in which each of the four individual transmembrane domains of Myc-CMTM4 were replaced with the respective transmembrane domain from CMTM6 (termed Myc-CMTM4$^{1-Tm6}$, Myc-CMTM4$^{2-Tm6}$, Myc-CMTM4$^{3-Tm6}$ and Myc-CMTM4$^{4-Tm6}$). We used retroviral transduction to express SF-IL-17RC together with Myc-CMTM4, Myc-CMTM6 or the chimeric mutants. IP of SF-IL-17RC indicated a strong interaction with Myc-CMTM4, Myc-CMTM4$^{3-Tm6}$ and Myc-CMTM4$^{4-Tm6}$, but not with Myc-CMTM4$^{1-Tm6}$ and Myc-CMTM4$^{2-Tm6}$ (Fig. 1j). These results indicated that CMTM4 constitutively associated with IL-17RC through the transmembrane domains of both proteins.

### CMTM4 mediates IL-17RC glycosylation and localization

Because CMTM4 was reported to stabilize the plasma membrane localization of the immunomodulatory receptor PD-L1 (ref. [19]), we tested whether CMTM4 was required for IL-17RC expression at the cell surface. All four *Cmtm4*$^{-/-}$ ST2 cell lines prepared using CRISPR–Cas9 system had markedly reduced surface expression of IL-17RC compared to wild-type cells (Fig. 2a and Extended Data Fig. 2a). Retroviral transduction of expression vector coding Myc-CMTM4 in two different *Cmtm4*$^{-/-}$ ST2 cell lines rescued the surface expression of IL-17RC compared to the EV (Fig. 2b and Extended Data Fig. 2b). Real-time quantitative PCR analysis showed similar relative amounts of *Il-17rc* messenger RNA in wild-type and *Cmtm4*$^{-/-}$ ST2 cell lines (Extended Data Fig. 2c), suggesting that CMTM4 did not impact the transcription of *Il-17rc*. The surface expression of IL-17RA, IL-17RD or TNF receptor 1 (TNFR1) in *Cmtm4*$^{-/-}$ ST2 cells were similar to wild-type cells (Extended Data Fig. 2d). In addition, the surface localization of ectopically expressed IL-17RC$^{TmRA}$, which did not bind CMTM4, was similar in wild-type and *Cmtm4*$^{-/-}$ ST2 cells (Extended Data Fig. 2e), suggesting that CMTM4 specifically bound to IL-17RC transmembrane domain to promote its surface expression.

Next we investigated whether CMTM4 was required for IL-17RC plasma membrane localization in human cells. Deletion of *CMTM4* in human HeLa cells using CRISPR–Cas9 system led to a substantial decrease in surface expression of IL-17RC compared to wild-type cells (Extended Data Fig. 2f). Re-expression of Myc-CMTM4 in *CMTM4*$^{-/-}$ HeLa cells through retroviral transduction rescued the surface expression of IL-17RC, while the surface expression of IL-17RA was not changed compared to cells transfected with EV (Extended Data Fig. 2g). Similarly, four different *CMTM4*$^{-/-}$ human A549 cell lines obtained using CRISPR–Cas9 system had strongly reduced surface expression of IL-17RC, but not IL-17RA, compared to wild-type A549 cells or control cell lines in which the transfection with CRISPR–Cas9 did not lead to the deletion of CMTM4 (Extended Data Fig. 2h). These data showed that CMTM4 was a major regulator of IL-17RC surface expression in human and mouse cell lines.

In unstimulated and IL-17A-stimulated ST2 cells, endogenous IL-17RC was detected in two forms with different molecular weights of approximately 100 kDa and 150 kDa (Fig. 2c). Expression of both forms was reduced in *Cmtm4*$^{-/-}$ ST2 cells compared to wild-type ST2 cells, with the 150 kDa form showing a more pronounced loss of expression in four different *Cmtm4*$^{-/-}$ ST2 cell lines (Fig. 2c and Extended Data Fig. 3a,b). Furthermore, *Cmtm4*$^{-/-}$ ST2 cells retrovirally transduced with Myc-CMTM4 expressed the 150 kDa form of IL-17RC while cells transduced with EV did not (Extended Data Fig. 3c), suggesting that CMTM4 regulated the posttranslational processing of IL-17RC.

The extracellular domain of IL-17RC contains several putative N-glycosylation sites[16]. To test whether IL-17RC was glycosylated, we immunoprecipitated SF-IL-17RC from ST2 cells ectopically expressing this protein and treated these samples with different glycosidases: PNGase F glycosidase, which removes all N-linked glycans, Endoglycosidase H (Endo H), which cleaves N-linked glycans not modified by *trans*-Golgi network-resident enzymes[20], O-glycosidase, which removes

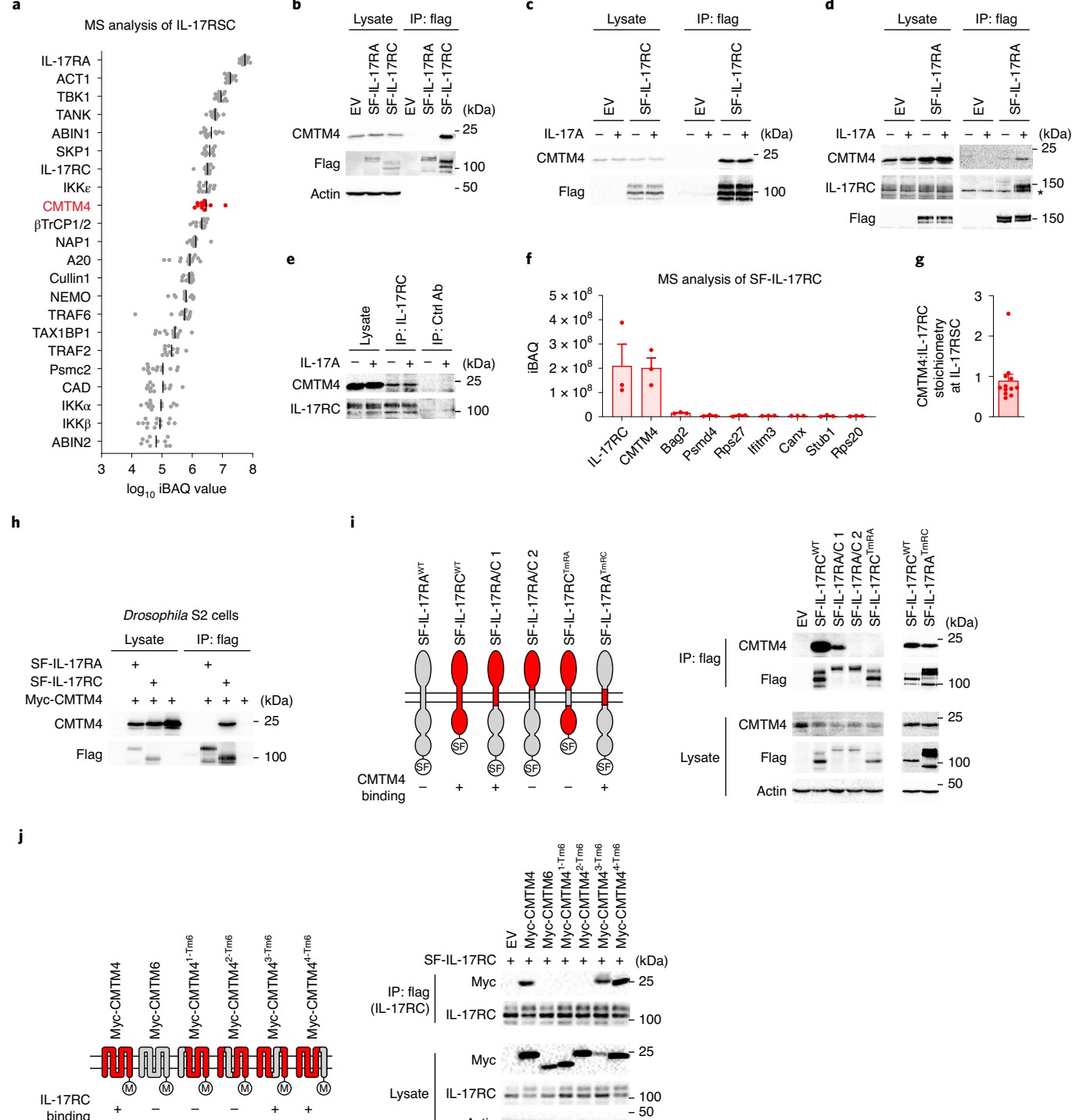

**Fig. 1 | CMTM4 is constitutively associated with IL-17RC transmembrane domain. a**, MS analysis of IL-17RSC isolated via tandem affinity purification from lysates of ST2 cells that were stimulated with SF-IL-17A (500 ng ml⁻¹) for 15 min. Proteins identified in stimulated samples, but not in control samples in which SF-IL-17A was added after lysis, are listed. Relative quantification is based on iBAQ values from 13 experiments. **b**, Immunoblotting analysis of samples isolated by Flag IP from lysates of ST2 cells transduced with expression vectors coding for SF-IL-17RA, SF-IL-17RC or EV. **c,d**, Immunoblotting analysis of samples isolated by Flag IP from lysates of ST2 cells expressing EV or SF-IL-17RC (**c**) or SF-IL-17RA (**d**) that were stimulated with IL-17A (500 ng ml⁻¹) for 15 min or left unstimulated. *, unspecific band. **e**, Immunoblotting analysis of samples isolated through IP with IL-17RC antibody or isotype control antibody from the lysates of ST2 cells stimulated with IL-17A (500 ng ml⁻¹) for 15 min or left unstimulated. **f**, MS analysis of proteins isolated via tandem affinity purification from the lysates of ST2 cells expressing SF-IL-17RC. Relative quantification is based on iBAQ values from three experiments. Only proteins that were not detected in control samples isolated from ST2 cells expressing SF-EGFP and that had an average iBAQ value >2 × 10⁶ are listed. **g**, Mean + s.e.m. of iBAQ ratio between CMTM4 and IL-17RC in IL-17RSC calculated for individual MS experiments shown in **a**. **h**, Immunoblotting analysis of samples isolated by Flag IP from lysates of *Drosophila* S2 cells transduced with expression vectors coding for SF-IL-17RA, SF-IL-17RC or EV together with the expression vector coding for Myc-CMTM4. **i**, Immunoblotting analysis of samples isolated through Flag IP from lysates of ST2 cells transduced with expression vectors coding for SF-IL-17RA, SF-IL-17RC or indicated chimeric proteins. **j**, Immunoblotting analysis of samples isolated by Flag IP from lysates of ST2 cells expressing CMTM4-Myc, CMTM6-Myc or indicated chimeric proteins together with SF-IL-17RC. Data are representative of two (**d,i,j**), three (**c,h**) or four (**b,e**) independent experiments.

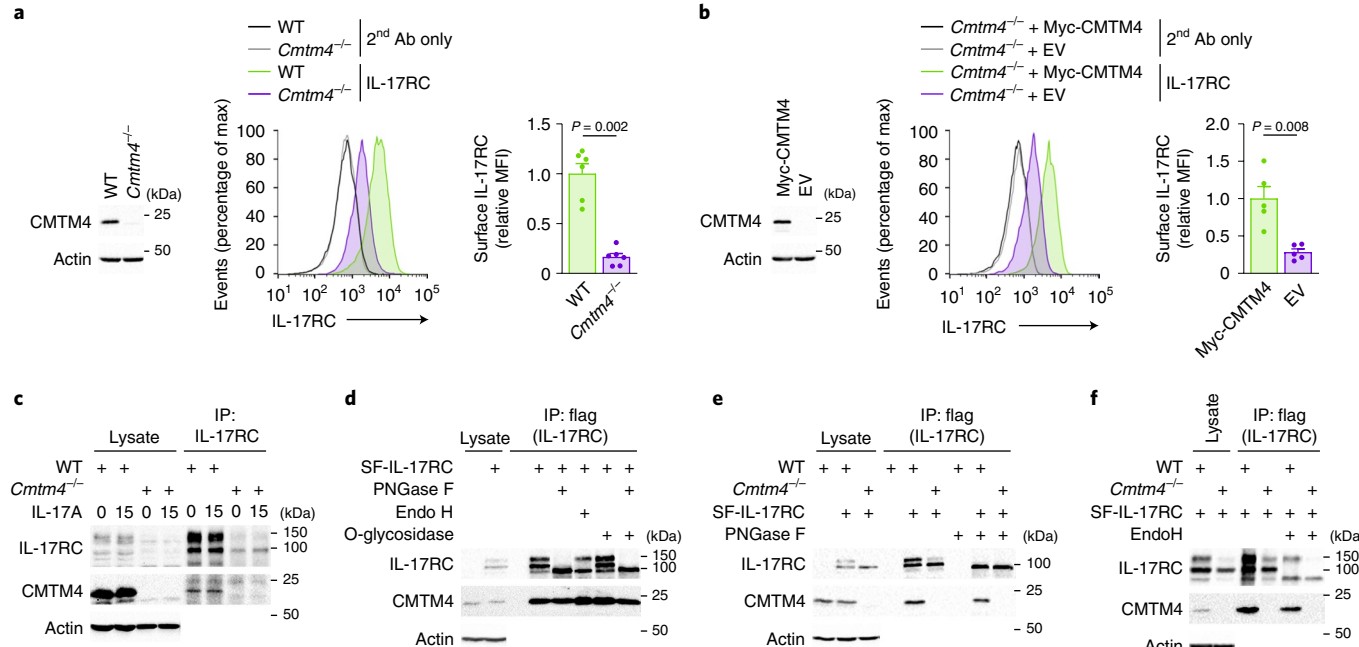

**Fig. 2 | CMTM4 mediates IL-17RC glycosylation and plasma membrane localization. a,b,** Immunoblotting analysis of CMTM4 and actin and flow cytometry analysis of IL-17RC surface expression in wild-type and *Cmtm4*^−/− ST2 cells (**a**) or *Cmtm4*^−/− ST2 cells transduced with expression vectors coding for Myc-CMTM4 or EV (**b**). **c,** Immunoblotting analysis of samples isolated via IP with an IL-17RC antibody from lysates of wild-type and *Cmtm4*^−/− ST2 cells stimulated with IL-17A (500 ng ml⁻¹) for 15 min or left unstimulated. **d–f,** Immunoblotting analysis

of samples isolated through Flag IP from lysates of wild-type and *Cmtm4*^−/− ST2 cells expressing SF-IL-17RC that were treated with O-glycosidase (**d**), PNGase F (**d**, **e**) or Endo H (**d**, **f**) as indicated. Data are representative of two (**c**–**f**) independent experiments. Flow cytometry data are represented as mean + s.e.m. from six (**a**) or five (**b**) independent experiments. Two-tailed Mann–Whitney test. MFI, median fluorescence intensity.

O-linked glycans or a combination of PNGase F and O-glycosidase. Treatment with PNGase F resulted in a single SF-IL-17RC band with an apparent molecular weight of approximately 85 kDa (Fig. 2d). In contrast, Endo H cleaved the 100 kDa form, but not the 150 kDa form, of SF-IL-17RC (Fig. 2d). Treatment with O-glycosidase did not lead to the decrease of SF-IL-17RC molecular weight nor did it further decrease the molecular weight of SF-IL-17RC that was previously treated with PNGase F (Fig. 2d), indicating that SF-IL-17RC was not substantially modified by O-glycans. Thus, both the 100 kDa and 150 kDa forms of SF-IL-17RC were N-glycosylated and the 150 kDa form of SF-IL-17RC was the more mature protein that did reach the *trans*-Golgi network.

SF-IL-17RC expressed in *Cmtm4*^−/− ST2 cells did not undergo full glycosylation corresponding to the 150 kDa band compared to SF-IL-17RC expressed in wild-type cells (Fig. 2e). Furthermore, the 100 kDa form of SF-IL-17RC predominantly detected in *Cmtm4*^−/− ST2 was fully cleaved with Endo H (Fig. 2f), indicating that CMTM4 was necessary for the full maturation of IL-17RC in the *trans*-Golgi compartment. Treatment of wild-type and *Cmtm4*^−/− ST2 cells with the proteasome inhibitor bortezomib led to the accumulation of the fully deglycosylated 85 kDa isoform of IL-17RC in both cell lines (Extended Data Fig. 3d), in agreement with a role of glycosylation in protein trafficking and stability[21].

To test whether any other member of the CMTM family can promote IL-17RC glycosylation, we expressed various Myc-tagged members of the CMTM family (CMTM1–8) together with SF-IL-17RC in *Cmtm4*^−/− ST2 cells using a retroviral transduction system. Only Myc-CMTM4 associated with SF-IL-17RC and rescued its glycosylation (Extended Data Fig. 3e). Finally, we expressed SF-IL-17RA, SF-IL-17RB, SF-IL-17RD, SF-IL-17RE or SF-TNFR1 in both wild-type and *Cmtm4*^−/− ST2 cells using retroviral transduction, immunoprecipitated these proteins and performed PNGase F treatment to confirm glycosylation. None of these proteins interacted with CMTM4, and CMTM4 deficiency did not impact their glycosylation (Extended Data Fig. 3f–j). These data

suggested that CMTM4 is specifically required for IL-17RC glycosylation and trafficking.

**Glycosylation regulates IL-17RC plasma membrane trafficking**

IL-17RC contains nine putative glycosylation sites defined as NxS/T (where x is any amino acid) (Fig. 3a). Expression constructs coding SF-IL-17RC or harboring mutations of different putative glycosylation sites (N₁₁₄Q, N₁₈₂Q, N₂₀₉Q, N₂₄₉/₂₅₅/₂₅₉Q, N₃₄₅Q and N₄₀₁/₄₀₂Q) or combination of multiple putative glycosylation sites (termed SF-IL-17RC^MutA to SF-IL-17RC^MutE) were retrovirally transduced in ST2 cells to assess how particular mutations in SF-IL-17RC impacted its glycosylation, judged as a shift in the apparent molecular weight. Only mutation of all nine potential glycosylation sites in SF-IL-17RC^MutE led to a fully deglycosylated form of 85 kDa (Fig. 3a), indicating that IL-17RC was heavily glycosylated on several residues. The surface expression of SF-IL-17RC^MutE in ST2 cells was strongly reduced compared to SF-IL-17RC^WT (Fig. 3b), showing that IL-17RC glycosylation is required for its plasma membrane expression. However, SF-IL-17RC^MutE construct bound endogenous CMTM4 to the same extent as SF-IL-17RC^WT (Fig. 3c), indicating that the interaction between CMTM4 and IL-17RC preceded IL-17RC glycosylation.

To directly address the role of CMTM4 in the localization of IL-17RC, we expressed a IL-17RC fused to enhanced green fluorescent protein (IL-17RC-EGFP) in wild-type or *Cmtm4*^−/− ST2 cells via retroviral transduction. While IL-17RC-EGFP expressed in ST2 wild-type cells was detected as a fully glycosylated, 150 kDa form, and localized to the plasma membrane, it appeared as a partially glycosylated, 100 kDa form, and maintained a cytoplasmic localization in *Cmtm4*^−/− ST2 cells (Fig. 3d–f). Coexpression of IL-17RC-EGFP and CMTM4-mCherry in *Cmtm4*^−/− ST2 cells rescued IL-17RC-EGFP glycosylation, judged by the appearance of the 150 kDa form and led to markedly higher surface expression of IL-17RC-EGFP compared to *Cmtm4*^−/− ST2 expressing

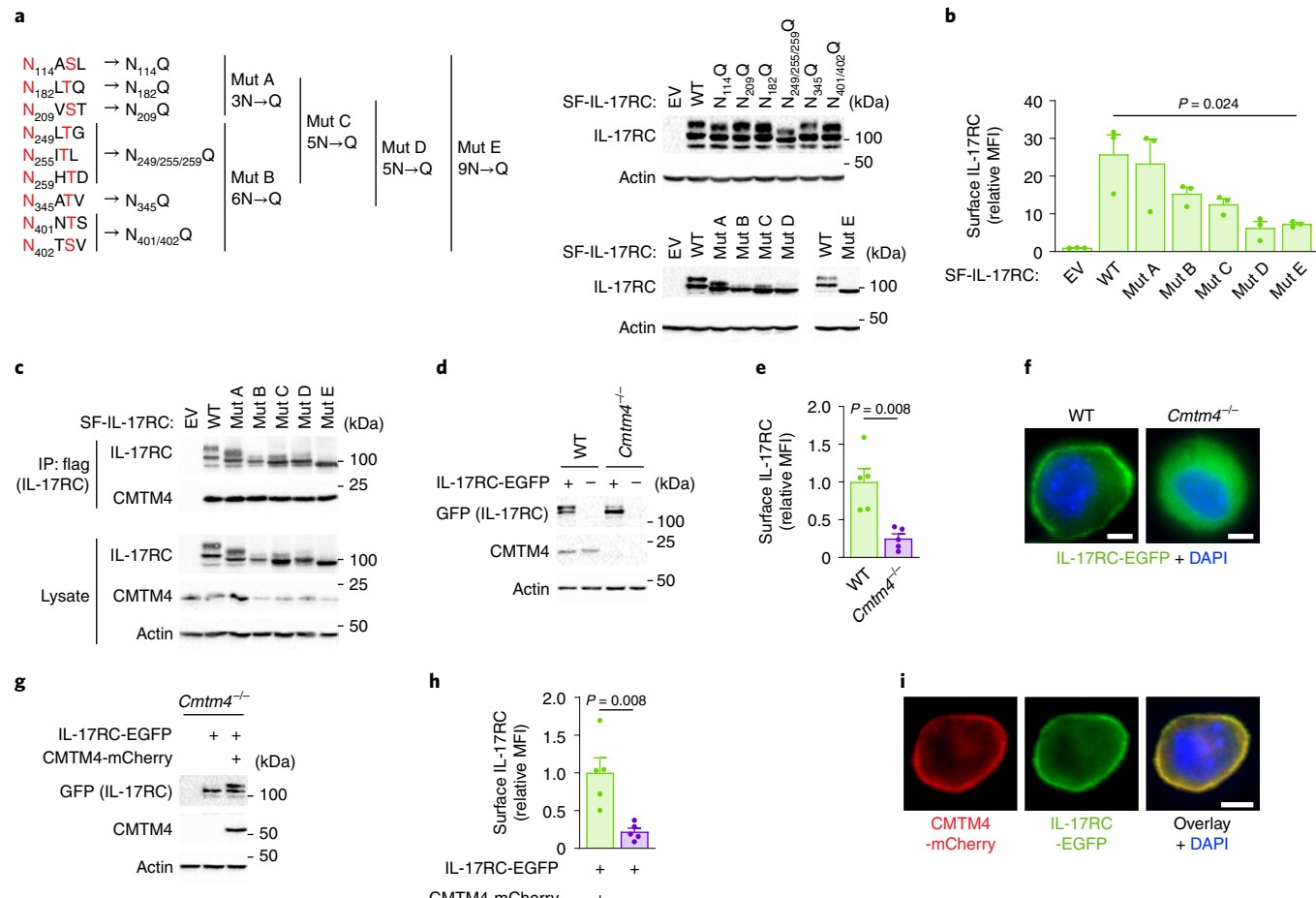

**Fig. 3 | IL-17RC glycosylation is required for trafficking to the plasma membrane. a**, Immunoblotting analysis of lysates of ST2 cells transduced with expression vectors coding for SF-IL-17RC harboring indicated mutations of the putative glycosylation sites. **b**, Flow cytometry analysis of IL-17RC surface expression in cells expressing indicated SF-IL-17RC constructs. **c**, Immunoblotting analysis of samples isolated through Flag IP from lysates of ST2 cells expressing indicated SF-IL-17RC constructs. **d**–**i**, Immunoblotting (**d**,**g**), flow cytometry of surface IL-17RC (**e**,**h**), and fluorescence microscopy (**f**,**i**) in *Cmtm4⁻/⁻* ST2 cells expressing IL-17RC-EGFP only (**d**,**e**,**f**) or together with CMTM4-mCherry (**g**,**h**,**i**). Scale bar, 5 μm. Data are representative of three (**a**–**c**,**f**,**i**) or two (**d**,**g**) independent experiments. Flow cytometry data show mean + s.e.m. from three (**b**) or five (**e**,**h**) independent experiments. Two-tailed Mann–Whitney test (**e**,**h**) or two-tailed unpaired *t*-test (**b**).

IL-17RC-EGFP only (Fig. 3g,h). Both IL-17RC-EGFP and CMTM4-mCherry colocalized at the plasma membrane in unstimulated cells (Fig. 3i) and remained colocalized also upon IL-17A stimulation (Extended Data Fig. 4a). In addition, confocal microscopy showed cytoplasmic localization of IL-17RC-EGFP that overlapped with endoplasmic reticulum marker Calnexin staining in *Cmtm4⁻/⁻* ST2 cells (Extended Data Fig. 4b). In contrast, *Cmtm4⁻/⁻* ST2 cells retrovirally transduced with vectors encoding IL-17RC-EGFP and CMTM4-mCherry showed that both these proteins were separated from Calnexin and localized on the cell surface (Extended Data Fig. 4b). These results showed that CMTM4 was not required for the initial steps of IL-17RC glycosylation, but its binding to IL-17RC was crucial for the full maturation of the glycan moieties on IL-17RC in the *trans*-Golgi compartment and the transport of mature IL-17RC to the plasma membrane.

## CMTM4 is required for IL-17A-induced signaling responses

Because IL-17RC is critical for IL-17A-induced signaling and subsequent cytokine production[11,22], we examined whether loss of surface IL-17RC in *Cmtm4⁻/⁻* ST2 cells impaired the formation of IL-17RSC and downstream signaling. Wild-type and *Cmtm4⁻/⁻* cells were stimulated with SF-IL-17A and the IL-17RSC was isolated through SF-IL-17A IP. *Cmtm4⁻/⁻* ST2 cells had a substantial defect in the recruitment of the adapter ACT1 and ubiquitin ligase TRAF6, two key proximal components of the IL-17RSC[23–25], compared to wild-type cells (Fig. 4a). The assembly of the IL-17RSC was rescued in *Cmtm4⁻/⁻* cells transduced with Myc-CMTM4, but not with EV (Extended Data Fig. 5a).

Next, we evaluated the impact of CMTM4 deficiency on the IL-17-induced cellular responses. Activation of NF-κB, p38 and JNK signaling pathways, as well as activation of negative regulatory loop mediated by kinase TBK1, were strongly reduced in IL-17A-stimulated *Cmtm4⁻/⁻* ST2 cells as compared to wild-type ST2 cells (Fig. 4b), and were rescued by ectopic expression of Myc-CMTM4 (Fig. 4c). In contrast, the activation of these signaling pathways by the proinflammatory cytokines IL-1α or TNF was not impacted in *Cmtm4⁻/⁻* ST2 cells (Extended Data Fig. 5b,c). IL-17A signaling is negatively regulated by TBK1 and IKKε-mediated phosphorylation of ACT1, which leads to the release of TRAF6 from the IL-17RSC[15,26]. Inhibition of TBK1 and IKKε kinases with the inhibitor MRT67307 led to the strong stimulation of NF-κB, p38 and JNK in wild-type, but not in *Cmtm4⁻/⁻* ST2 cells (Extended Data Fig. 5d), suggesting a role for CMTM4 upstream of the ACT1-TRAF6 interaction.

The transcription of the IL-17-responsive genes encoding the proinflammatory cytokines CCL2, CXCL1, TNF and IL-6 was markedly reduced in *Cmtm4⁻/⁻* compared to wild-type ST2 cells (Fig. 4d).

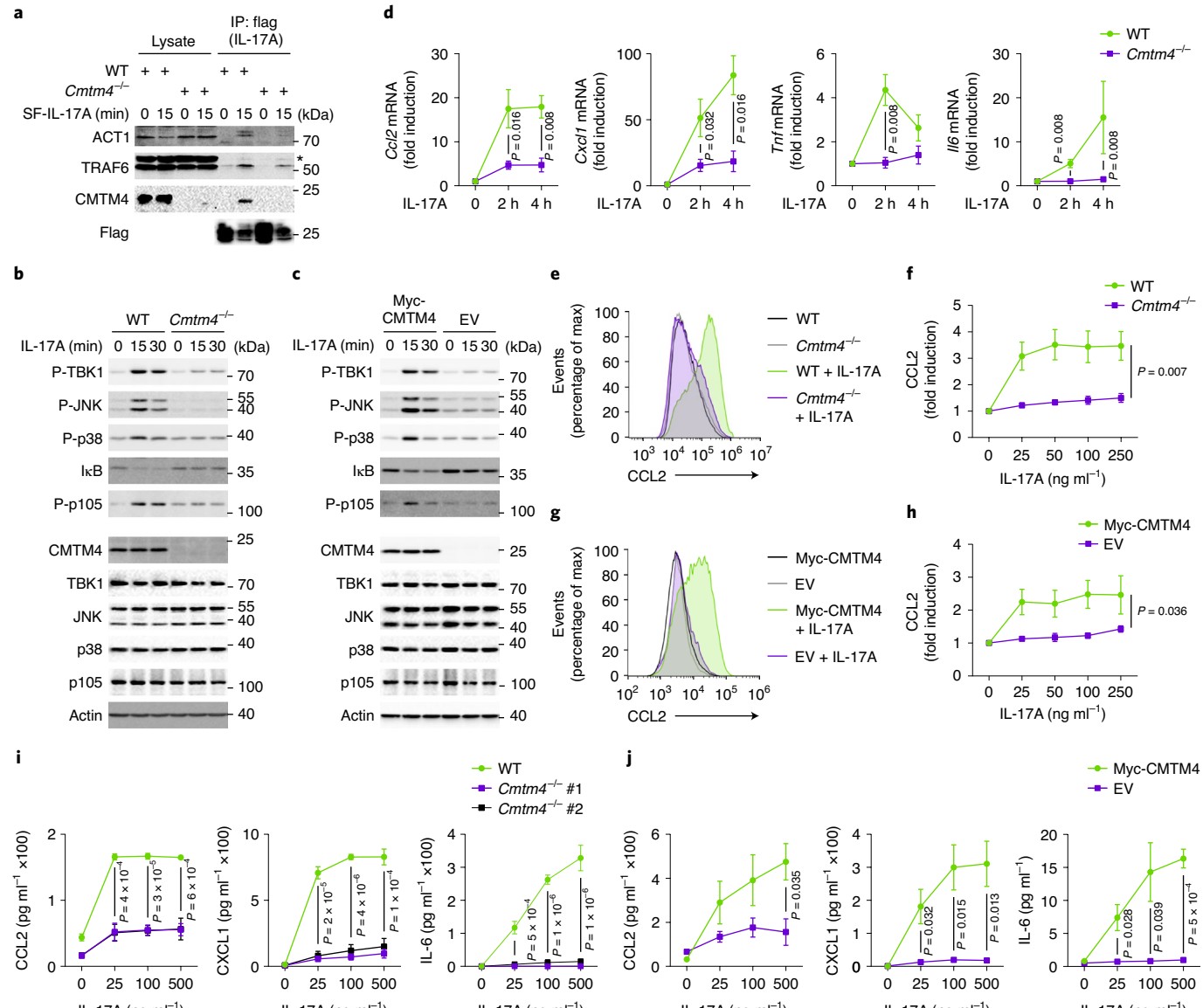

**Fig. 4 | CMTM4 is required for IL-17A-induced signaling and cytokine production. a,** Immunoblotting of samples isolated by Flag IP from lysates of wild-type or *Cmtm4*$^{-/-}$ ST2 cells stimulated with SF-IL-17A (500 ng ml$^{-1}$) for 15 min or left unstimulated, with SF-IL-17A added post lysis. *, unspecific band. **b,c,** Immunoblotting of lysates of wild-type or *Cmtm4*$^{-/-}$ ST2 cells (**b**) or *Cmtm4*$^{-/-}$ ST2 cells expressing Myc-CMTM4 or EV (**c**) and stimulated with IL-17A (500 ng ml$^{-1}$) for indicated time points. **d,** Real-time quantitative PCR analysis of the fold induction of *Ccl2*, *Cxcl1*, *Tnf* and *Il6* transcripts upon stimulation of wild-type or *Cmtm4*$^{-/-}$ ST2 cells with SF-IL-17A (500 ng ml$^{-1}$) for the indicated time points. Mean ± s.e.m. from five independent experiments. Two-tailed Mann–Whitney

test. **e–h,** Flow cytometry analysis of CCL2 induction in wild-type or *Cmtm4*$^{-/-}$ ST2 cells (**e,f**) or *Cmtm4*$^{-/-}$ ST2 cells expressing Myc-CMTM4 or EV (**g,h**) stimulated with IL-17A (25 ng ml$^{-1}$) (**e,g**) or with the indicated concentration of IL-17A (**f,h**) for 4 h in the presence of protein transport inhibitor brefeldin A. Mean ± s.e.m. from five independent experiments, two-way ANOVA. **i,j,** ELISA detection of cytokines CCL2, CXCL1, IL-6 in supernatants from wild-type or *Cmtm4*$^{-/-}$ ST2 cells (**i**) or *Cmtm4*$^{-/-}$ ST2 cells expressing Myc-CMTM4 or EV (**j**) stimulated with the indicated concentration of IL-17A for 24 h. Mean ± s.e.m. from three independent experiments. One-way ANOVA (**i**) or two-sided unpaired *t*-test (**j**). Data (**a–c**) are representative of three independent experiments.

Similarly, the production of CCL2 protein measured by flow cytometry was strongly decreased in *Cmtm4*$^{-/-}$ ST2 cells upon stimulation with a broad range of IL-17A concentrations (Fig. 4e,f), and was rescued by retroviral transduction of Myc-CMTM4 (Fig. 4g,h). Enzyme-linked immunosorbent assay (ELISA) indicated the impaired production of CCL2, CXCL1 and IL-6 proinflammatory cytokines in two different *Cmtm4*$^{-/-}$ ST2 cell lines compared to wild-type ST2 cells upon stimulation with a wide range of IL-17A concentrations (Fig. 4i), which was rescued by re-expression of Myc-CMTM4 (Fig. 4j). IL-17RC is also a receptor for IL-17F (ref. [16]). The IL-17F-induced transcription of *Ccl2* and *Cxcl1* was markedly inhibited in *Cmtm4*$^{-/-}$ ST2

cells compared to wild-type ST2 cells (Extended Data Fig. 5e). Immunoprecipitation of IL-17RSC from human *CMTM4*$^{-/-}$ HeLa cell line showed severely decreased recruitment of adapter ACT1 and ubiquitin ligase TRAF6 compared to wild-type cells (Extended Data Fig. 5f). In addition, the activation of p38, JNK and TBK1 and transcription of genes encoding proinflammatory cytokines CCL20, TNF, IL-6 and CXCL2 upon IL-17A stimulation were markedly decreased in *CMTM4*$^{-/-}$ HeLa as compared to wild-type cells (Extended Data Fig. 5g,h). Altogether, these data indicated that the formation of IL-17RSC and downstream signaling were severely reduced in CMTM4-deficient human and mouse cells.

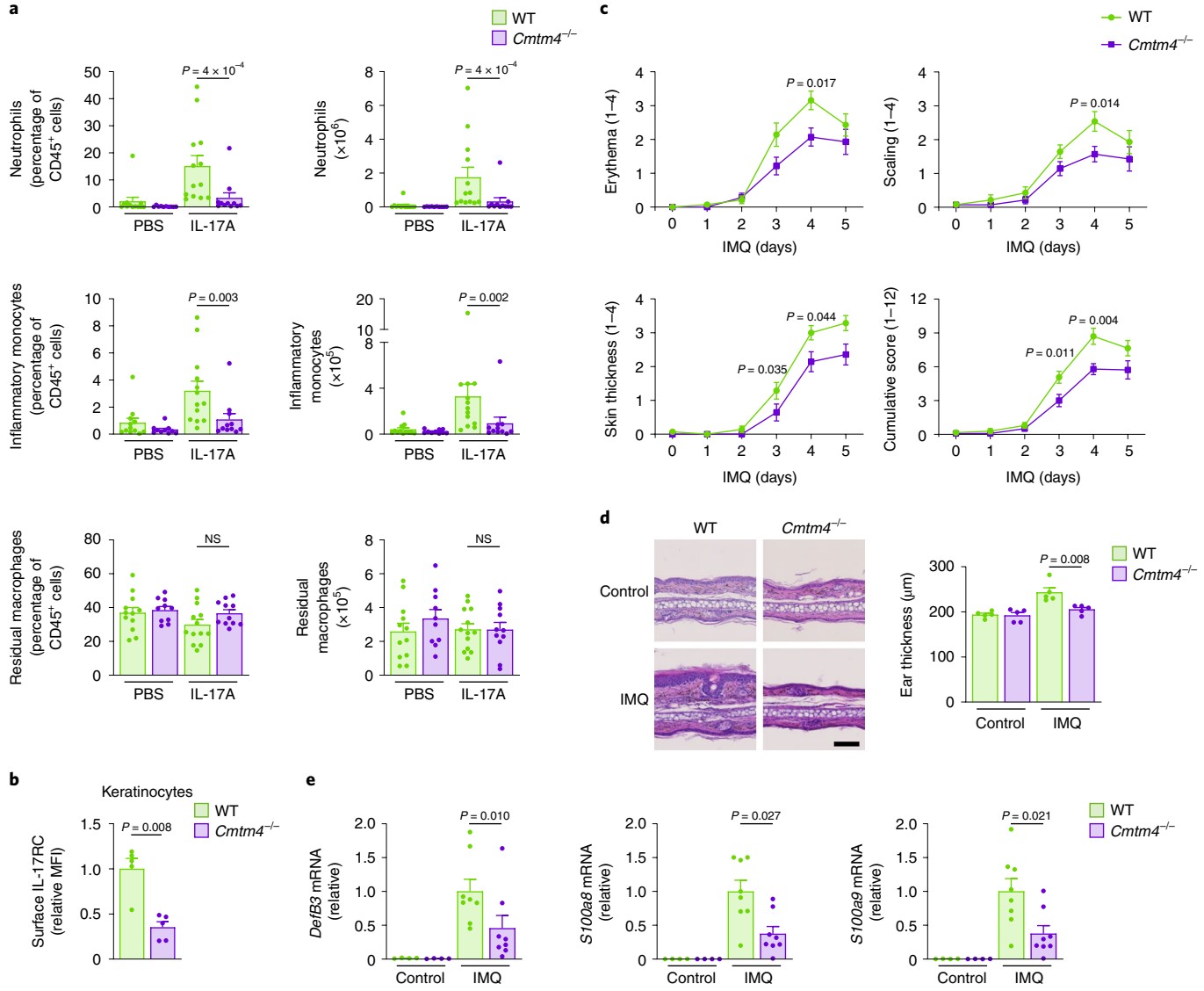

**Fig. 5 | CMTM4 regulates IL-17R-mediated inflammation and autoimmune pathology. a**, Flow cytometry analysis of intraperitoneal lavage from *Cmtm4*⁺/⁺ (WT) or *Cmtm4*⁻/⁻ littermate mice that were intraperitoneally injected with IL-17A (500 ng) or PBS only for 4 h. CD45⁺ leukocyte subsets were gated as CD11b⁺F4/80⁺ residual macrophages, F4/80⁻CD11b⁺Ly6G⁺ neutrophils and F4/80⁻Ly6G⁻CD11b⁺Ly6C⁺ inflammatory monocytes. *n* = 12 (*Cmtm4*⁺/⁺ control), 13 (*Cmtm4*⁺/⁺ + IL-17A), 10 (*Cmtm4*⁻/⁻ control) and 11 (*Cmtm4*⁻/⁻ + IL-17A) in three independent experiments. Mean + s.e.m. Two-tailed Mann–Whitney test. NS, not significant. **b**, Flow cytometry analysis of IL-17RC surface expression in differentiated keratinocytes isolated from *Cmtm4*⁺/⁺ or *Cmtm4*⁻/⁻ littermate mice. Mean + s.e.m. from five independent experiments. Two-tailed Mann–Whitney test. **c**, Analysis of experimental psoriasis disease progression in *Cmtm4*⁺/⁺ or *Cmtm4*⁻/⁻ littermate mice treated with Aldara cream containing 5% IMQ on the

shaved skin of back and ears daily for 5 consecutive days and monitored daily. *n* = 14 mice per group in three independent experiments. Mean ± s.e.m. score for erythema, scaling, dorsal skin thickness or combined cumulative score. Two-tailed Mann–Whitney test. **d**, Hematoxylin and eosin staining of sections of ear skin treated with Aldara for 4 days or the opposite ear in each mouse, which was left untreated, isolated from *Cmtm4*⁺/⁺ or *Cmtm4*⁻/⁻ littermate mice. Scale bar, 100 μm, *n* = 6 mice per group. Mean ± s.e.m., two-tailed Mann–Whitney test. **e**, Real-time quantitative PCR analysis of the relative expression of *DefB3*, *S100a8* and *S100a9* transcripts in the back skin isolated from *Cmtm4*⁺/⁺ or *Cmtm4*⁻/⁻ littermate mice treated or not with Aldara cream for 4 consecutive days. *n* = 4 for control groups, and 8 for IMQ-treated groups. Mean ± s.e.m., two-tailed Mann–Whitney test.

## CMTM4 regulates IL-17R-mediated inflammation in vivo

Analysis of publicly available RNA sequencing data from various mouse and human tissues indicated a correlation between the expression of mouse *Cmtm4* and *Il17rc* and human *CMTM4* and *IL17RC* (Extended Data Fig. 6a,b and Supplementary Table 3). Analysis of public single-cell RNA sequencing data of human skin and small intestine indicated the coexpression of *CMTM4* and *IL17RC*, but not of *CMTM4* and the ubiquitously expressed *IL-17RA* (Extended Data Fig. 6c,d), suggesting an evolutionary link between *CMTM4* and *IL17RC*.

To investigate the role of CMTM4 in vivo, we generated *Cmtm4*⁻/⁻ mice based on a *Cmtm4*-targeted mouse strain from the European Mouse Mutant Archive repository[27] (Extended Data Fig. 7a,b) and analyzed both male and female littermates at the age of 8–12 weeks. *Cmtm4*⁻/⁻ mice were born in a Mendelian ratio (Extended Data Fig. 7c), appeared healthy, and had no major changes in their T cell and B cell compartment, except very mildly increased frequencies of B cells and decreased frequencies of naïve CD8⁺ T cells in the lymph nodes, but not in the spleen (Extended Data Fig. 7d). *Cmtm4*⁻/⁻ mouse embryonic

fibroblasts (MEFs) showed impaired IL-17-induced signaling measured as phosphorylation of p38, JNK and TBK1, which was accompanied by the decreased amount of IL-17RC protein and IL-17RC glycosylation in comparison to wild-type MEFs (Extended Data Fig. 7e).

The injection of IL-17A intraperitoneally induced massive recruitment of CD45⁺CD11b⁺Ly6G⁺ neutrophils and CD45⁺CD11b⁺Ly6G⁻Ly6C⁺ inflammatory monocytes in wild-type mice as expected[28], whereas the response of Cmtm4⁻/⁻ mice was much weaker (Fig. 5a). As a control, the amount of intraperitoneal residual CD45⁺CD11b⁺F4/80⁺ macrophages in Cmtm4⁻/⁻ mice remained unchanged compared to Cmtm4⁺/⁺ littermates (Fig. 5a). These data demonstrated that CMTM4 mediates IL-17A responses in vivo.

The primary keratinocytes isolated from Cmtm4⁻/⁻ mouse tail had markedly reduced surface expression of IL-17RC in comparison with keratinocytes from Cmtm4⁺/⁺ littermates (Fig. 5b). Therefore, we tested the role of CMTM4 in the imiquimod (IMQ)-induced experimental model of psoriasis, in which the IL-17 signaling in keratinocytes is one of the main drivers of the disease progression[29,30]. The IMQ was applied on the ears or shaven backs of Cmtm4⁻/⁻ or Cmtm4⁺/⁺ female littermates for 5 consecutive days and the progression of the disease was monitored daily. Cmtm4⁻/⁻ mice had substantially milder back skin erythema, scaling and thickening on days 3 and 4 (Fig. 5c and Extended Data Fig. 8a) and ear skin thickness on days 2–5 (Extended Data Fig. 8b) as compared to Cmtm4⁺/⁺ mice. Histological analysis of ears upon 4 days of IMQ treatment indicated less severe skin pathology (Fig. 5d), and real-time quantitative PCR indicated markedly lower expression of the IL-17A target genes, DefB3, S100a8 or S100a9 (ref. [31]) in IMQ-treated skin (Fig. 5e) in Cmtm4⁻/⁻ mice compared to Cmtm4⁺/⁺ littermates. IL-17A signaling in keratinocytes induces the production of IL-23 leading to the 'feed-forward' amplification of inflammation[32]. The transcription of Il23, Il17a, Il17f and Il17c genes was decreased, albeit not significantly, in the IMQ-treated skin from Cmtm4⁻/⁻ mice compared to Cmtm4⁺/⁺ littermates (Extended Data Fig. 8c). Irradiated wild-type mice transplanted with bone marrow grafts from Cmtm4⁺/⁺ or Cmtm4⁻/⁻ littermates showed similar progression of IMQ-induced psoriasis (Extended Data Fig. 8d), indicating that CMTM4 had a limited role in the immune cell compartment in this model. Analysis of publicly available expression databases[33,34] showed a slight decrease in the expression of both IL17RC and CMTM4 in the psoriatic skin in human patients (Extended Data Fig. 8e), which could be due to the infiltration of immune cells that lack these two proteins (Extended Data Fig. 6).

IL-17A was reported to modulate the severity of myelin oligodendrocyte glycoprotein (MOG)-induced experimental autoimmune encephalomyelitis (EAE), although the magnitude of this effect varies substantially between studies[35–37]. Mice were immunized with MOG peptide 35-55 to induce EAE and monitored daily for clinical signs of their disease progression. Cmtm4⁻/⁻ mice had a slightly milder EAE progression, based on the combined severity score, compared to Cmtm4⁺/⁺ littermates, albeit the difference did not reach statistical significance (Extended Data Fig. 8f). Overall, our data indicated that CMTM4 regulates IL-17A signaling in vivo and mediates IMQ-induced psoriasis, while it had only a limited role in MOG-induced EAE.

## Discussion

Here, using the proteomic approach, we identified the tetra-transmembrane protein CMTM4 as a new subunit of IL-17R and showed that CMTM4 was required for IL-17RC plasma membrane expression, assembly of IL-17RSC and activation of IL-17A signaling in mouse and human cell lines and animal models.

CMTM4 was strongly associated with the transmembrane domain of IL-17RC early in the secretory pathway and we detected a strong association between ectopically expressed mouse CMTM4 and IL-17RC in a Drosophila cell line. We interpret these observations as proof that the interaction between CMTM4 and IL-17RC was direct. If another protein was required to mediate the interaction between CMTM4 and IL-17RC, this protein would need to be expressed in insect cells as well,

although this remains a possibility. CMTM4 enabled IL-17RC glycosylation by trans-Golgi resident enzymes and subsequent transport to the plasma membrane. However, Cmtm4⁻/⁻ ST2 cells showed normal glycosylation and localization of several transmembrane proteins, indicating that CMTM4 was not a general regulator of glycosylation or protein trafficking. Our analysis of publicly available RNA sequencing data suggested that CMTM4 has a relatively broad tissue distribution, although it seemed missing in immune cells. If CMTM4 regulated global protein N-glycosylation, we would expect an altered function of a larger number of glycosylated proteins, possibly manifested in a more severe phenotype in Cmtm4⁻/⁻ mice[38].

CMTM4 was required for the activation of IL-17A signaling response, likely due to its role in promoting IL-17RC plasma membrane expression, although it is possible that CMTM4 has additional functions in the assembly of IL-17RSC. CMTM4 was an important regulator of IL-17A signaling in vivo, as the inflammatory response to intraperitoneal IL-17A injection was severely compromised in Cmtm4⁻/⁻. Furthermore Cmtm4⁻/⁻ mice were partially protected from IMQ-induced psoriasis to an extent comparable with Il17ra⁻/⁻ mice[39]. In contrast, we observed only mild and statistically nonsignificant improvement of EAE progression in Cmtm4⁻/⁻ mice as compared to Cmtm4⁺/⁺ mice. The role of IL-17A in this model is controversial, with reports ranging from important to almost negligible[35–37]. Mice deficient in IL-17A and IL-17F (termed IL17⁻/⁻) were reported to be protected from disease progression in a MOG-induced EAE model, mainly due to differences in microbial colonization between the wild-type and IL17⁻/⁻ mice and this effect was partially reverted by cohousing wild-type and IL17⁻/⁻ mice together[40]. Because we used littermates resulting from the cross of Cmtm4⁺/⁻ mice, the microbiota was likely equalized in our experimental Cmtm4⁺/⁺ and Cmtm4⁻/⁻ mice, which might explain the similar disease progression in the MOG-induced EAE model. Alternatively, CMTM4 might regulate the expression of additional protein(s) that might have a positive or negative regulatory role in the progression of EAE. CMTM4 was reported to regulate the expression of the immunoregulatory protein PD-L1 in several human cell lines[19,41]. However, CMTM6 largely complemented CMTM4 in the trafficking of PD-L1[19,42] and therefore it is unlikely that the membrane expression of PD-L1 is substantially affected in Cmtm4⁻/⁻ mice.

Altogether, our results showed that CMTM4 may function as a subunit of the IL-17R. CMTM4 assembled together with IL-17RC and IL-17RA to form the ligand-engaged IL-17RSC and was required for the cellular responses to IL-17. In this respect, CMTM4 is analogous to a tetraspanin protein CD81, which is an established subunit of the B cell complement receptor (CD19/CD21/CD81), although it does not directly bind the ligand and has no direct role in the signal transduction[43]. Given its role in mediating IL-17A-induced responses, CMTM4 represents a potential new target for the therapy of IL-17A-mediated autoimmune diseases.

## Online content

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

## Methods

### Cell lines

ST2 cells were kindly provided by J. Balounova. HeLa and HEK293FT, Phoenix-Eco and Phoenix-Ampho cells were kindly provided by T. Brdicka (both from the Institute of Molecular Genetics, Prague, Czech Republic) and A549 cells were kindly provided by Z. Novakova (Institute of Biotechnology, Vestec, Czech Republic). MEF cells were derived from E11.5 mouse embryos and immortalized by lentiviral infection with the SV40 large T antigen. Cell lines were cultured at 37 °C, 5% $CO_2$ in a complete Dulbecco's Modified Eagle Medium (DMEM) medium containing 10% fetal bovine serum (FBS) (Gibco) and antibiotics penicillin/streptomycin (Biosera). All cell lines were tested for Mycoplasma contamination by PCR regularly. S2 cells were kindly provided by Petr Draber (Institute of Molecular Genetics, Prague, Czech Republic). S2 cells were cultured at 28 °C without $CO_2$ in complete Schneider's *Drosophila* Medium (Gibco) containing 10% heat-inactivated fetal bovine serum and penicillin/streptomycin.

### Antibodies and reagents

Antibodies detecting β-actin (Cat. number 3700), IκBα (9242), P-p105 (Ser932) (4806), P-p38 (Thr180/Tyr182) (9216), P-JNK (Thr183/Tyr185) (4671), P-TBK1(S172) (5483), Myc (2276), TBK1 (3013), JNK2 (9258), p105 (4717 or 13586) were purchased from Cell Signaling Technology; CMTM4 (HPA014704), FLAG (F3165), GFP (SAB4301138), human IL-17RC (HPA019885) were from Sigma-Aldrich; ACT1 (sc-398161) and p38 (sc-728) were from Santa Cruz Biotechnology; murine IL-17RA (MAB4481), IL-17RC (AF2270), IL-17RD (AF2276) and TNFR1 (AF-425-PB) were from R&D Systems; and TRAF6 (ab40675) was from Abcam. The standard dilution of antibodies for immunoblotting was 1:1,000.

Secondary antibodies Anti-Rabbit IgG (H+L) HRP (711-035-152), Goat Anti-Mouse IgG1 HRP (115-035-205), Goat Anti-Mouse IgG2a HRP (115-035-206), Goat Anti-Mouse IgG2b HRP (115-035-207), Donkey Anti-Goat IgG H+L HRP (705-035-147), Goat Anti-Rabbit IgG Fc fragment specific HRP (111-035-046), Goat Anti-Rat IgG (H+L) HRP (112-035-167), Donkey Anti-Rabbit IgG (H + L) AF488 (711-545-152) and Donkey Anti-Goat IgG (H+L) AF647 (705-605-147) were purchased from Jackson Immunoresearch, dilution 1:10,000 for HRP-conjugated antibodies, 1:500 for fluorophore-conjugated antibodies. For detection of ACT1, we used Mouse TrueBlot ULTRA (Rockland), dilution 1:5,000.

Fluorescently labeled antibodies against following antigens were used: CD1d Pe-Cy7 (123524), CD4 PerCP (100538), CD8a BV421 (100738), CD8a BV510 (100752), CD11b BV421 (101251), CD11c AF700 (117320), CD19 PE (115508), CD25 PE-Cy7 (102016), CD44 PE (103008), CD44 PerCP-Cy5.5 (103032), GITR APC (126312), F4/80 PE (123110), IgM BV421 (406518), IgD PerCP-Cy5.5 (405710), KLRG1 BV510 (138421), Ly6C PE-Cy7 (128018), Ly6G AF647 (127610), TCRβ PerCP (109212), MCP-1/CCL2 PE (505903), B220 AF700 (103232), TCRβ PE (109208), CD45.1 BV650 (110735), Flag APC (637307) and human IL-17RA-APC (372305) were purchased from BioLegend; CD3 FITC (553062), CD19 FITC (553785), CD25 FITC (553071), CD45.2 FITC (553772), CD49d PE (553157) and NK1.1 FITC (553164) were from BD Pharmingen; CD23 APC (1108095) was from SONY; FOXP3 PE-Cy7 (25-5773-82) was from eBioscience; and Calnexin AF647 (MA3-027-A647) was from Invitrogen. The standard dilution of fluorescently conjugated antibodies was 1:200. LIVE/DEAD Fixable Near-IR Dead Cell Stain Kit was from Thermo Fisher Scientific.

The following recombinant proteins were used in this study: SF-IL-17A was composed of 6xHis, 2xStrep tag, 1xFlag tag and either murine IL-17A (AA 26-158) or human IL-17A (AA 24-155). IL-17A was composed of 6xHis, 1xMyc followed by murine or human IL-17A. IL-17F was composed of 6xHis, 2xStrep tag, 1xFlag tag and murine IL-17F (AA29-161). TNF was composed of 6xHis, 2xStrep tag, 1xFlag tag and murine TNF (AA60-215). Recombinant proteins were produced in HEK293FT cells and purified via His GraviTrap TALON column (GE Healthcare) as described previously[15]. Commercial murine IL-17A was

purchased from R&D Systems (7956-ML). Murine IL-1α was from PeproTech (211-11A).

### DNA cloning and viral transduction

Murine CMTM1–8, IL-17RD, IL-17RE, TNFR1 and IL-17RA$^{TmRC}$ coding sequences were prepared using the GeneArt Gene Synthesis service (Thermo Fisher Scientific). Coding sequences of murine IL-17RA, IL-17RB and IL-17RC were amplified from ST2 wild-type cells. Constructs were fused at C-terminus to 2xStrep-3xFlag tag, Myc, EGFP or mCherry as indicated. Swapping mutants between IL-17RC extracellular (AA1-461), transmembrane (AA462-491) and cytoplasmic (AA492-698) parts and IL-17RA transmembrane (AA315-349), and cytoplasmic (AA350-864) parts, swapping mutants between transmembrane domains of CMTM4 (TM1 AA1-83, TM2 AA84-110, TM3 AA111-147, TM4 AA148-208) and CMTM6 (TM1 AA1-67, TM2 AA68-94, TM3 AA95-131, TM4 AA132-183), and constructs coding for various mutants of putative IL-17RC glycosylation sites were prepared by fusion PCR approach using Phusion polymerase (New England BioLabs). Constructs were inserted in retroviral pBabe vector (kindly provided by M. Hrdinka, University Hospital Ostrava, Czech Republic) expressing EGFP marker, or puromycin-resistance marker for constructs fused to EGFP or mCherry, under SV40 promoter. All constructs were sequenced.

Platinum-Eco or Phoenix-Ampho cells were employed to produce viruses for transduction of mouse or human cells, respectively. Cells were transfected with pBabe vector harboring coding sequences of indicated proteins or EV using Lipofectamine 2000 (Invitrogen). Virus-containing supernatants were collected, passed through a 0.2 μm filter, and added to target cells in the presence of 6 μg ml$^{-1}$ polybrene followed by spinning at 2,500 r.p.m. for 45 min. Transduced cells were isolated by fluorescence-activated cell sorting (FACS) using FACSAria IIu (BD Biosciences).

Constructs for transfection of S2 cells were cloned in the pMT-Puro inducible expression vectors. The S2 cells were seeded in 60-mm plates and transfected with 20 μg of vectors using calcium phosphate according to the manufacturer's protocol (Gibco). Cells were cultivated for 14 days in the presence of 2.5 μg ml$^{-1}$ puromycin to select stably transfected cells. The expression of the transfected genes was induced with 500 μM $CuSO_4$ for 3–4 days and verified by immunoblotting.

### Generation of knockout-cell lines

CMTM4-deficient cell lines were generated using the CRISPR–Cas9 system. Single-guided RNA (sgRNA) targeting selected genes were designed using the web tool CHOPCHOP[44] and inserted in the pSpCas9(BB)-2A-GFP (PX458) vector kindly provided by Feng Zhang (Addgene plasmid no. 48138)[45]. Below is the list of sgRNA target sequences with PAM motif indicated in bold and underlined:

Mouse CMTM4 5′-GAAGTAGAGGCCTTCGCACG**GGG** (used in ST2 cells)

Human CMTM4 5′-GAGCGCGCCGCGCAGGTAGT**CGG** (used in HeLa cells and A549 Series no. 2)

Human CMTM4 5′-GCTGCTGGCGCCCGAGATCA**TGG** (used in A549 Series no. 1)

Cell lines were transfected using Lipofectamine 2000. EGFP-expressing cells were sorted as single cells in 96-well plates on FACSAria III (BD Bioscience). Clones were analyzed for the expression of target proteins by immunoblotting and sequencing of DNA surrounding the sgRNA target site.

### Cell stimulation

Before stimulation, ST2 or HeLa cells were washed and incubated in a serum-free DMEM medium for 30 min. In some experiments, cells were pretreated for 15 min with TBK1 and IKKε inhibitor MRT67307 (2 μM) (Tocris Bioscience). Depending on whether a murine or human cell line was used, cells were stimulated with murine or human cytokines as indicated. Subsequently, cells were lysed in 1% n-Dodecyl-β-D-Maltoside

(DDM) containing lysis buffer (30 mM Tris pH 7.4, 120 mM NaCl, 2 mM KCl, 2 mM ethylenediaminetetraacetic acid (EDTA), 10% glycerol, 10 mM chloroacetamide, 10 mM complete protease inhibitory cocktail and PhosSTOP tablets (Roche)). Samples were incubated at 4 °C for 30 min, cleared by centrifugation (21,130$g$, 30 min, 2 °C), mixed with sodium dodecyl sulfate (SDS) sample buffer, reduced by 50 mM dithiothreitol (DTT) and analyzed by immunoblotting.

## Immunoprecipitation

For each experimental condition, ST2 or HeLa cells were grown on one 15-cm dish. Cells were washed with serum-free DMEM and solubilized in 1% DDM lysis buffer. For isolation of IL-17RSC, cells were first incubated for 30 min in serum-free DMEM and subsequently stimulated with SF-IL-17A (500 ng ml$^{-1}$) for 15 min or left untreated and SF-IL-17A was added post lysis. Alternatively, cells expressing SF-IL-17RA or SF-IL-17RC were stimulated with IL-17A (500 ng ml$^{-1}$) for 15 min. Lysates were incubated on ice for 30 min and cleared by centrifugation (21,130$g$, 30 min, 2 °C). Parts of the lysates were mixed with reducing SDS sample buffer (with 50 mM DTT). The rest of the sample was subjected to immunoprecipitation with 10 µl of anti-FLAG M2 affinity agarose gel (Sigma) overnight. Subsequently, agarose beads were washed with 0.1% DDM containing lysis buffer three times and eluted by heating to 94 °C for 3 min in SDS sample buffer with 50 mM DTT. Samples were analyzed by immunoblotting.

Drosophila S2 cells were stably transfected with indicated constructs and the expression of the transfected genes was induced with 500 µM CuSO$_4$ for 3–4 days. Cells were collected, washed once with phosphate-buffered saline (PBS) buffer and lysates were subjected to IP with 10 µl of anti-FLAG M2 affinity agarose gel as described above.

For isolation of endogenous IL-17RC, cells were either stimulated or not with IL-17A (500 ng ml$^{-1}$) for 15 min. Lysates were incubated for 1 h with IL-17RC antibody or isotype control antibody and subsequently isolated via Protein A/G PLUS agarose (Santa Cruz Biotechnology). Alternatively, cells were incubated for 24 h with proteasome inhibitor bortezomib (10 nM) (Merck) before IL-17RC IP.

## PNGase F, Endo H, O-glycosidase treatment

Lysates of ST2 wild-type or $Cmtm4^{-/-}$ cells expressing SF-IL-17RC were subjected to IP with anti-FLAG M2 affinity agarose gel as described above. Each sample was subsequently denatured for 10 min at 100 °C and treated for 1 h at 37 °C with PNGase F, Endo H or O-glycosidase according to the manufacturer's instructions (New England Biolabs). Samples were eluted by heating to 94 °C for 3 min in SDS sample buffer and analyzed by immunoblotting.

## Tandem affinity purification and MS analysis

To isolate IL-17RSC, ST2 cells were grown on 6 × 15 cm dishes per condition, washed and incubated in serum-free DMEM for 30 min and activated with murine SF-IL-17A (500 ng ml$^{-1}$ in 10 ml per condition) for 15 min, and solubilized in 1% DDM lysis buffer. Control samples were left untreated and IL-17A was added post lysis. Lysates were incubated on ice for 30 min and cleared by centrifugation (21,130$g$, 30 min, 2 °C). Eleven out of 13 IL-17RSC MS analyses were published in our previous work, and are deposited in the PRIDE database under accession number PXD019020 (ref. [15]).

ST2 cells expressing SF-IL-17RC or control SF-GFP were grown on 3 × 15 cm dishes per condition. Cells were washed with serum-free medium and lysed in 1% DDM lysis buffer.

The first IP step was carried out by overnight incubation of samples with 50 µl of anti-FLAG M2 affinity agarose gel (Sigma). The beads were washed 3× with 0.1% DDM containing lysis buffer and isolated proteins were eluted by incubation of the beads in 250 µl of 1% DDM containing lysis buffer supplemented with 100 µg ml$^{-1}$ of 3xFlag peptide (Sigma) overnight. The supernatant was collected, and the elution step was repeated once again for 8 h.

The second purification step was carried out upon incubation of the samples with 50 µl of Strep-Tactin Sepharose beads (IBA Lifesciences) overnight. The samples were subsequently washed 3× with 0.1% DDM containing lysis buffer and 1× with lysis buffer alone. Bound proteins were eluted upon incubation of the beads with 220 µl of MS Elution buffer (2% sodium deoxycholate in 50 mM Tris pH 8.5).

The eluted protein samples (200 µl) were reduced with 5 mM tris(2-carboxyethyl)phosphine at 60 °C for 60 min and alkylated with 10 mM methyl methanethiosulfonate at room temperature for 10 min. Proteins were cleaved overnight with 1 µg of trypsin (Promega) at 37 °C. To remove sodium deoxycholate, samples were acidified with 1% trifluoroacetic acid, mixed with an equal volume of ethyl acetate, centrifuged (15,700$g$, 2 min) and an aqueous phase containing peptides was collected[46]. This step was repeated two more times. Peptides were desalted using in-house-made stage tips packed with C18 disks (Empore)[47] and resuspended in 20 µl of 2% acetonitrile with 1% trifluoroacetic acid.

The digested protein samples (12 µl) were loaded onto the trap column (Acclaim PepMap300, C18, 5 µm, 300 Å Wide Pore, 300 µm x 5 mm) using 2% acetonitrile with 0.1% trifluoroacetic acid at a flow rate of 15 µl min−1 for 4 min. Subsequently, peptides were separated on Nano Reversed-phase column (EASY-Spray column, 50 cm × 75 µm internal diameter, packed with PepMap C18, 2 µm particles, 100 Å pore size) using a linear gradient from 4% to 35% acetonitrile containing 0.1% formic acid at a flow rate of 300 nl min$^{-1}$ for 60 min.

Ionized peptides were analyzed on a Thermo Orbitrap Fusion (Q-OT-qIT, Thermo Scientific). Survey scans of peptide precursors from 350 to 1400 $m/z$ were performed at 120 K resolution settings with a $4 \times 10^5$ ion count target. Four different types of tandem MS were performed according to precursor intensity. The first three types were detected in Ion trap in rapid mode, and the last one was detected in Orbitrap with 15,000 resolution settings: (1) For precursors with intensity between $1 \times 10^3$ to $7 \times 10^3$ with collision-induced dissociation (CID) fragmentation (35% collision energy) and 250 ms of ion injection time; (2) For ions with intensity in range from $7 \times 10^3$ to $9 \times 10^4$ with CID fragmentation (35% collision energy) and 100 ms of ion injection time; (3) For ions with intensity in range from $9 \times 10^4$ to $5 \times 10^6$ with higher-energy-collisional-dissociation (HCD) fragmentation (30% collision energy) and 100 ms of ion injection time; and (4) For intensities $5 \times 10^6$ and more with HCD fragmentation (30% collision energy) and 35 ms of ion injection time. The dynamic exclusion duration was set to 60 s with a 10 ppm tolerance around the selected precursor and its isotopes. Monoisotopic precursor selection was turned on. The instrument was run in top speed mode with 3 s cycles.

All MS data were analyzed and quantified with the MaxQuant software (v.1.6.5.0)[48]. The false discovery rate was set to 1% for both proteins and peptides and the minimum length was specified as seven amino acids. The Andromeda search engine was used for the MS−MS spectra search against the murine Swiss-Prot database (downloaded from Uniprot in June 2019). Trypsin specificity was set as C-terminal to Arg and Lys, also allowing the cleavage at proline bonds and a maximum of two missed cleavages. β-methylthiolation, N-terminal protein acetylation, carbamidomethylation and Met oxidation were included as variable modifications. Label-free quantification was performed using the iBAQ algorithm, which divides the sum of all precursor-peptide intensities by the number of theoretically observable peptides[17]. Data analysis was performed using Perseus v.1.6.14.0 software[49]. Complex stoichiometry was estimated based on the iBAQ intensity ratio between individual components of the complex and IL-17RC.

## RNA sequencing analysis

The average expression of *IL17RA*, *Il17RC* and *CMTM4* in human tissues and *Il17ra*, *Il17rc* and *Cmtm4* in murine tissues measured by bulk RNA sequencing was analyzed using the Genevestigator software (v.8.3.0)[50]. Our analysis covered the Genevestigator category 'mRNA-Seq Gene

Level', which was filtered to exclude samples annotated as 'cell lines, neoplasms, unspecified organ/tissue/cells, organotypic, 3D culture'. In the case of murine samples, only samples of the wild-type genetic background were included. The total counts of included samples were 26,710 (human) and 5,251 (mouse).

The analysis of expression of selected transcripts in patient samples was based on Dataset HS-1529 (accession number GSE54456)[33] containing 90 healthy, 27 nonlesional psoriasis and 97 lesional psoriasis samples. Dataset HS-2913 (accession number GSE121212)[34] contained 38 healthy, 27 nonlesional psoriasis and 25 lesional psoriasis samples. The average values of the $\log_2$(expresion+1) per different tissues, cell types or patient samples were plotted.

Previously published single-cell RNA sequencing data of human tissues[51,52] were downloaded from the Gene Expression Omnibus (accession numbers GSE159929, GSE130973). Data were processed with the R (v.4.0.4) package Seurat (v.4.0.1)[53] following the methods of the original studies. Clusters were annotated with the metadata from original studies. For selected cell types, the percentages of cells expressing *Il17RA*, *Il17RC* and *CMTM4* (at least one detected transcript of the gene) were visualized using a heatmap.

### Fluorescence microscopy

Cells expressing IL-17RC-EGFP or CMTM4-mCherry as indicated were transferred to suspension, stained for 5–10 min with 4′,6-diamidino-2-phenylindole (DAPI) (BioLegend), washed with Hanks' Balanced Salt Solution (HBSS) and fixed in 3.7% formaldehyde for 10 min at room temperature. Fixed cells were washed, resuspended in HBSS medium with 10% FBS (HBSS/FBS), and placed into an eight-chambered cover glass system (Cellvis C8-1.5H-N). Image acquisition was performed on the Delta Vision Core microscope (inverted microscope v.IX-71, Olympus America Inc.) using the oil immersion objective (Plan-Apochromat 60× NA 1.42) and filters for DAPI (435/48) and fluorescein isothiocyanate (FITC) (523/36). The images were processed using the Fiji ImageJ.

For intracellular stainings, fixed cells were washed with HBSS/FBS and permeabilized with 0.1% Triton-X for 10 min, washed and stained with Calnexin AF647 antibody for 1 h. After staining, cells were washed, resuspended in HBSS/FBS and analyzed by inverted confocal microscope Leica TCS SP8 WLL SMD-FLIM. The images were processed using Leica Las AF software.

### Flow cytometry detection of surface receptors and CCL2 induction

To detect indicated surface receptors, cells were transferred to suspension, resuspended in FACS buffer (PBS, 2% FBS, 0.1% NaN₃) and stained with fluorescently labeled antibodies or with the indicated antibody on ice flowed by APC-labeled secondary antibody. To analyze the SF-IL-17A binding kinetics, ST2 cells were incubated on ice with indicated concentration of SF-IL-17A followed by APC-labeled Flag antibody. Propidium iodide solution was used for the discrimination of live and dead cells. To detect the production of CCL2, cells were washed with serum-free DMEM and stimulated for 4 h with the indicated concentration of IL-17A in the presence of 5 µg ml⁻¹ Brefeldin A (BioLegend). Cells were collected, fixed and permeabilized using Cyto-Fast Fix/Perm Buffer Set (BioLegend) and stained with fluorescently labeled CCL2 antibody. Samples were measured on a BriCyte E6 flow cytometer and data were analyzed using FlowJo software (TreeStar).

### Real-time PCR

Liquid nitrogen snap frozen ear samples were homogenized using a mortar and pestle. ST2 and HeLa cells were first stimulated with SF-IL-17A (500 ng ml⁻¹) or SF-IL-17F (500 ng ml⁻¹) for 2 and 4 h or left untreated. Total RNA was extracted using Quick RNA MiniPrep (Zymo Research). DNAse I treatment was performed according to the manufacturer's instructions. Reverse transcription was done using LunaScript

RT SuperMix (New England BioLabs). Quantitative PCR was performed using Luna Universal qPCR Master Mix (New England BioLabs) and measured on LightCycler 480 II (Roche) in technical triplicates for each experiment. Raw data were analyzed in The LightCycler 480 Software 1.5 using second derivation analysis. Obtained Ct values were normalized to GAPDH and quantified. The following primer pairs were used:

m*Gapdh*: CGTCCCGTAGACAAAATGGT and TTGATGGCAACAATCTCCAC

m*Cxcl1*: CTTGAAGGTGTTGCCCTCAG and TGGGGACACCTTTTAGCATC

m*Ccl2*: CCCAATGAGTAGGCTGGAGA and TCTGGACCCATTCCTTCTTG

m*Tnf*: CCACCACGCTCTTCTGTCTAC and AGGGTCTGGGCCATAGAACT

m*Il6*: ATGGATGCTACCAAACTGGAT and TGAAGGACTCTGGCTTTGTCT

m*DefB3*: TGTCTCCACCTGCAGCTTTT and GTGTTGCCAATGCACCGATT

m*S100a8*: AGTGTCCTCAGTTTGTGCAG and ACTCCTTGTGGCTGTCTTTG

m*S100a9*: CACCCTGAGCAAGAAGGAAT and TGTCATTTATGAGGCTTCATTT

m*Il17a*: TTTAACTCCCTTGGCGCAAAA and CTTTCCCTCCGCATTGACAC

m*Il17c*: CTGGAAGCTGACACTCACG and GGTAGCGGTTCTCATCTGTG

m*Il17f*: AATTCCAGAACCGCTCCAGT and TTGATGCAGCCTGAGTGTCT

m*Il23*: TGTGCCCCGTATCCAGTGT and CGGATCCTTTGCAAGCAGAA

h*GAPDH*: AAGGTGAAGGTCGGAGTCAA and AATGAAGGGGTCATTGATGG

h*CXCL2*: CTCAAGAATGGGCAGAAAGC and CTTCAGGAACAGCCACCAAT

h*CCL20*: CTGGCTGCTTTGATGTCAGT and CGTGTGAAGCCCACAATAAA

h*TNF*: TCAGCCTCTTCTCCTTCCTG and GCCAGAGGGCTGATTAGAGA

h*IL6*: AGTGAGGAACAAGCCAGAGC and GTCAGGGGTGGTTATTGCAT

### ELISA

Cells of the indicated genotype were stimulated with IL-17A (25, 100, 500 ng ml⁻¹). After 24 h, supernatants were collected and cleared by centrifugation (800*g*, 3 min). Cytokine levels in the cell supernatants were determined via ELISA, according to the manufacturer's instructions (R&D Systems).

### Mice

Frozen sperm from mouse strain C57BL/6N-A^tm1Brd Cmtm4^tm1a(EUCOMM)Wtsi/WtsiBiat was obtained from The European Mouse Mutant Archive repository (EM:06038) and used for in vitro fertilization. Mice carrying the targeted *Cmtm4* allele were crossed with Flp-deleter mouse strain B6.Cg-Tg(ACTFLPe)9205Dym/J from The Jackson Laboratory (005703). The resulting mouse strain with exons 2 and 3 flanked by LoxP sites was crossed with Cre-deleter strain Gt(ROSA)26Sor^tm1(ACTB-cre,-EGFP)Ics (MGI:5285392, Philippe Soriano) to obtain a germline knockout mouse. Animal protocols were approved by the Resort Professional Commission for Approval of Projects of Experiments on Animals of the Czech Academy of Sciences, Czech Republic.

*Cmtm4*^+/+ and *Cmtm4*^−/− littermates used in experiments were generated by mating heterozygous animals. Animals were kept on C57BL/6J background. All mice used in experiments were 5–12 weeks old. They were housed in a specific pathogen-free facility 12 h/12 h light/dark cycle, temperature and relative humidity were maintained at

$22 \pm 1\,°C$ and $55 \pm 5\%$, respectively. Both males and females were used for experiments, except for IMQ-induced psoriasis and EAE where female littermates were analyzed. If possible, littermates were equally divided into the experimental groups. Mice were genotyped by PCR reaction with Taq DNA polymerase (Top-Bio) using the following primer pairs:

*Cmtm4* wild-type allele: TTGGCTTTTCAGAAGAAATTGA and CTGTGGCCCTCTCAAAGAAG

*Cmtm4* knockout allele: AGCCTACGGCAGCTTAGTCA and GGTGTGAGCCACCACTAGGT

### Isolation and culture of primary keratinocytes
Primary keratinocytes were isolated from the tails of $Cmtm4^{+/+}$ and $Cmtm4^{-/-}$ adults as described previously[54]. Briefly, the skin was incubated with 0.25% trypsin in PBS overnight at 4 °C. On the following day, the dermis and epidermis were separated. Cell suspension from the epidermis was seeded in plates precoated with type I collagen solution (Advanced BioMatrix) and keratinocytes were cultured for 7 days in Eagle's minimal essential medium (Lonza) with 8% chelated fetal bovine serum (Gibco), penicillin/streptomycin (Biosera) and 2.5 mM $CaCl_2$ to induce keratinocyte differentiation. The medium was replaced every 2 days. Surface IL-17RC was analyzed by flow cytometry.

### Peritoneal lavage
Mice aged 8–12 weeks were injected intraperitoneally with 200 µl of PBS solution containing or not 0.5 µg of SF-IL-17A. After 4 h, mice were anesthetized with a mixture of 8 mg ml$^{-1}$ of ketamine (Narketan) and 2 mg ml$^{-1}$ of xylazine (Rometar). Anesthetized mice were beheaded to bleed out and peritoneal lavage was performed using 11 ml of PBS. Cell suspensions were stained with LIVE/DEAD near-IR dye (Life Technologies) and the mixture of primary antibodies (CD45.2, CD11b, F4/80, Ly6G, Ly6C) on ice and analyzed by flow cytometry using Cytek Aurora. CD45.2$^+$ cells were separated into the following subsets: residual macrophages (CD11b$^+$, F4/80$^+$), neutrophils (F4/80$^-$, Ly6G$^+$, CD11b$^+$) and inflammatory monocytes (F4/80$^-$, Ly6G$^-$, CD11b$^+$, Ly6C$^+$).

### Analysis of immune cell populations
Mice aged 8–12 weeks were sacrificed and the spleen, peripheral lymph nodes, and mesenteric lymph nodes were removed, and single-cell suspensions were prepared. In the case of the spleen, red blood cells were lysed in ACK buffer (150 mM $NH_4Cl$, 10 mM $KHCO_3$, 0.1 mM EDTA-$Na_2$, pH 7.4). Cells were resuspended in FACS buffer and stained on ice with LIVE/DEAD near-IR dye (Life Technologies) and separated into specific subsets based on staining with the following antibodies: T cell compartment (TCRβ, CD4, CD8, CD44, CD49d). T cells (TCRβ$^+$) were separated into the following subsets: CD4$^+$ T cells (CD4$^+$) and CD8$^+$ T cells (CD8$^+$), which were further divided into naïve CD8$^+$ (CD44$^-$), memory CD8$^+$ (CD44$^+$, CD49d$^+$) and antigen-inexperienced memory T cells (CD44$^+$, CD49d$^-$) cells[55]. B cell compartment (CD19, IgM, IgD, CD23, CD1d). B cells (CD19$^+$) were separated into the following subsets: T1 (IgM$^+$, CD23$^-$, CD1d$^-$), T2 (IgM$^+$, CD23$^+$, CD1d$^-$), marginal zone B cells (IgM$^+$, CD23$^-$, CD1d$^+$), mature (IgM$^-$, IgD$^+$) and isotype switched (IgM$^-$, IgD$^-$). Myeloid compartment (CD3, CD19, NK1.1, CD11b, CD11c, Ly6C, Ly6G). Myeloid cells (CD3$^-$, CD19$^-$, NK1.1$^-$, CD11b$^+$) were separated into the following subsets: neutrophils (CD11c$^-$, Ly6G$^+$) and monocytes/macrophages (CD11c$^-$, Ly6G$^-$). To distinguish Tregs and memory CD4 T cells, cells were fixed and permeabilized using Foxp3/Transcription Factor Staining Buffer Set (eBioscience) and stained with the following antibodies (CD4, CD8, CD25, CD44, FoxP3, GITR). CD4$^+$ T cells were divided into subsets: Tregs (FoxP3$^+$, CD25$^+$) and CD4$^+$ naïve cells (FoxP3$^-$, CD44$^-$) and CD4$^+$ memory cells (FoxP3$^-$, CD44$^+$). Samples were measured on Cytek Aurora and data were analyzed using FlowJo software (TreeStar).

### IMQ-induced psoriasis
Female mice aged 8–12 weeks were randomly distributed in cages, shaved with an electric razor followed by a hair removal cream

(Veet). To the shaved back skin and the ear, 62.5 mg of Aldara cream (Meda) containing 5% IMQ was applied daily for 5 consecutive days. The mice were monitored in a blinded manner to their genotype. The thickness of the back skin and the ear was measured every day using a digital micrometer (Kroeplin) and the pictures were taken daily of the mouse back skin. All procedures were performed under isoflurane anesthesia. The analysis of skin erythema and scaling was scored blind based on a 4-point scale. The score scale for scaling: 0, no scaling; 1, slight; 2, moderate; 3, substantial; 4, severe. The score scale for erythema: 0, no erythema; 1, slightly pink skin; 2, pink skin; 3, red skin; 4, dark violet skin. The score scale for back thickness was: 0, <800 µm; 1, 801–900 µm, 2, 901–1,000 µm; 3, 1,001–1,100 µm; 4, more than 1,100 µm. The cumulative score on a scale of 1–12 is based on these three parameters and is related to the Psoriasis Area and Severity Index used in patients[29].

Alternatively, mice were treated or not with IMQ on shaved back or ear for 4 consecutive days. Freshly removed samples of back skin were snap frozen in liquid nitrogen and stored at −80 °C for subsequent RNA isolation. Ears were fixed with 4% formaldehyde in PBS, transferred in 70% EtOH, and then embedded in paraffin. Sections of 5 µm were prepared and stained with hematoxylin and eosin. The average ear thickness was calculated from five measurements per ear.

### Bone marrow transfer
Bone marrow cells were isolated from long bones (femur, tibia) of $Cmtm4^{+/+}$ or $Cmtm4^{-/-}$ mice. In total, $2 \times 10^6$ cells in 200 µl of PBS were intravenously injected into irradiated (6 Gy) 6-week-old recipient Ly5.1 mice. Engraftment of primary bone marrow stem cells was confirmed via flow cytometry analysis of peripheral blood 7–8 weeks after the transfer using the following antibodies: CD45.1, CD45.2, TCRβ, B220, CD11b, Ly6C and Ly6G. The host mice were subjected to IMQ-induced psoriasis induction 8 weeks after the transfer.

### EAE
Female mice aged 8 weeks were randomly distributed in cages and subcutaneously injected with 200 µg MOG$_{35-55}$ (myelin oligodendrocyte glycoprotein 35-55; peptides&elephants GmbH), emulsified in incomplete Freund's adjuvant (Sigma) containing 500 µg inactivated *Mycobacterium tuberculosis* H37RA (Difco Laboratories) followed by intraperitoneal injection of 400 ng pertussis toxin (Merck) in a blinded manner to the genotype of mice. The pertussis toxin injection was repeated 2 days later. Mice were monitored daily in a blinded manner and the clinical score was determined as follows: 0, no clinical sign; 1, limp tail; 2, mild hind limb weakness; 3, complete unilateral hind limb paralysis or partial bilateral hind limb paralysis; 4, complete bilateral hind limb paralysis; 5, four limb paralysis; 6, death. In the case of mouse death, score 6 was recorded for all subsequent days.

### Statistics
The indicated statistical analyses were performed using Prism (Graph-Pad Software). Most of the experimental data were analyzed by a two-tailed nonparametric Mann–Whitney statistical test as indicated. In Extended Data Fig. 2b–d,e–h and Fig. 3b, data were analyzed by unpaired two-tailed Student's t-test; normality was assumed based on a similar set of experiments in Fig. 2a,b that passed the Kolmogorov–Smirnov test. In Fig. 4f–h, two-way analysis of variance (ANOVA) test was used, and data normality was tested via Kolmogorov–Smirnov test. Extended Data Fig. 5e,h data were analyzed by unpaired two-tailed Student's t-test; normality was assumed based on a similar set of experiments in Fig. 4d that passed the Kolmogorov–Smirnov test. In Fig. 4i, data were analyzed by one-way ANOVA and in Fig. 4j by unpaired two-tailed Student's t-test; normality was assumed based on the Shapiro–Wilk test. In Extended Data Fig. 6a–c, data were analyzed via Spearman's correlation test. In Extended Data Fig. 8f, data were analyzed by two-way ANOVA.

**Reporting summary**

Further information on research design is available in the Nature Research Reporting Summary linked to this article.

## Data availability

All MS proteomics data have been deposited to the ProteomeXchange Consortium via the PRIDE[56] under accession number PXD036366. RNA sequencing data were downloaded from Gene Expression Omnibus under accession numbers GSE159929, GSE130973, GSE54456 and GSE121212. Source data are provided with this paper.

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

## Acknowledgments

We thank L. Cupak for technical assistance. We thank K. Harant and P. Talacko from the Laboratory of Mass Spectrometry, Biocev, Charles University, Faculty of Science, where proteomics and mass spectrometric analysis had been done. This project was supported by the Czech Science Foundation (grant no. 20-24912Y to P.D.), EMBO Installation (grant no. 4420 to P.D.), Charles University (grant no. PRIMUS/20/MED/003 to P.D.) and the European Union (Horizon 2020 research and innovation program under grant agreement no. 802878, ERC Starting Grant FunDiT to O.S.). We acknowledge Charles University institutional funding (Cooperatio and UNCE/MED/016), core funding of the Institute of Molecular Genetics of the Czech Academy of Sciences, Czech Republic (IMG ASCR) (RVO 68378050), the project National Institute for Cancer Research (Programme EXCELES, LX22NPO5102) and the project National Institute of virology and bacteriology (Programme EXCELES, LX22NPO5103)—both funded by the European Union—Next Generation EU and grant no. SVV 260521 provided by the Ministry of Education, Youth and Sports of the Czech Republic (MEYS). The animal housing and *Cmtm4*[−/−] line rederivation and histology analysis were performed in the Czech Centre for Phenogenomics at IMG, ASCR supported by MEYS (LM2018126). We acknowledge the Light Microscopy Core Facility, IMG ASCR, supported by MEYS (LM2018129, CZ.02.1.01/0.0/0.0/18_046/0016045), for their help with the microscopy.

## Author contributions

P.D. conceived the study. D.K., M.P., H.D., T.S., T.T., A.S., A.U., J.S., A.D., M.H., O.T., A.N. and J.T. planned, performed and analyzed experiments. V.N. analyzed RNA sequencing data. P.D. and O.S. planned experiments and wrote the manuscript. All authors commented on the manuscript draft.

## Competing interests

The authors declare no competing interests.

## Additional information

**Extended data** is available for this paper at https://doi.org/10.1038/s41590-022-01325-9.

**Correspondence and requests for materials** should be addressed to Ondrej Stepanek or Peter Draber.

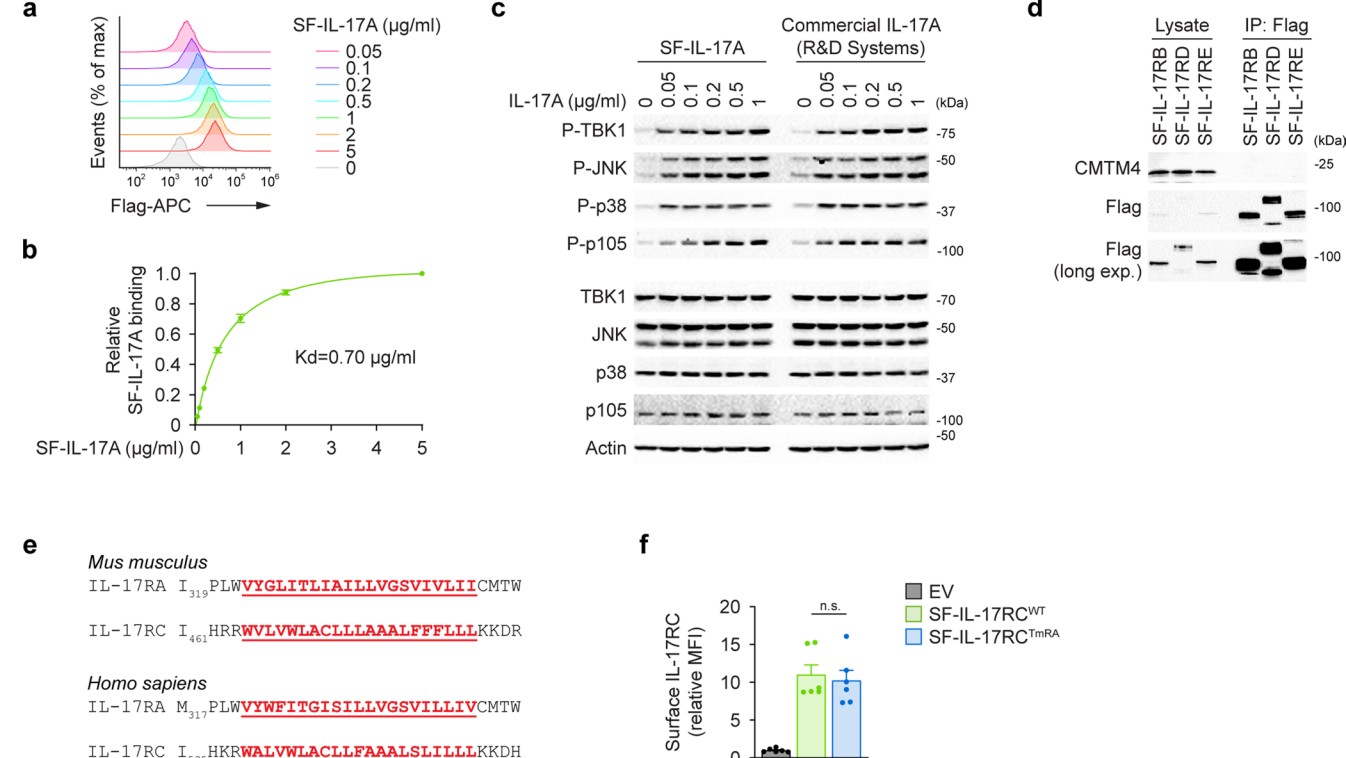

**Extended Data Fig. 1 | CMTM4 is specifically associated with IL-17RC. a, b,** Flow cytometry analysis of ST2 cells incubated with the indicated concentration of SF-IL-17A. Ligand binding was detected by Flag-APC antibody (**a**). Relative SF-IL-17A binding to ST2 cells was calculated from three independent experiments, mean ± s.e.m. (**b**). **c,** Immunoblotting analysis of lysates of wild-type ST2 cells that were stimulated with indicated concentrations of SF-IL-17A or commercial IL-17A (R&D Systems) for 15 min. **d,** Immunoblotting analysis of samples isolated through Flag immunoprecipitation from lysates of ST2 cells expressing SF-IL-17RB, SF-IL-17RD, or SF-IL-17RE. **e,** Comparison between murine and human IL-17RA and IL-17RC transmembrane domains. **f,** Flow cytometry analysis of surface expression of SF-IL-17RC$^{WT}$ and SF-IL-17RC$^{TmRA}$ ectopically expressed in ST2 cells. Mean + s.e.m. from six independent experiments. Two-tailed Mann–Whitney test. n.s. not significant. MFI, median fluorescence intensity. Data are representative of three (**a**–**d**) independent experiments.

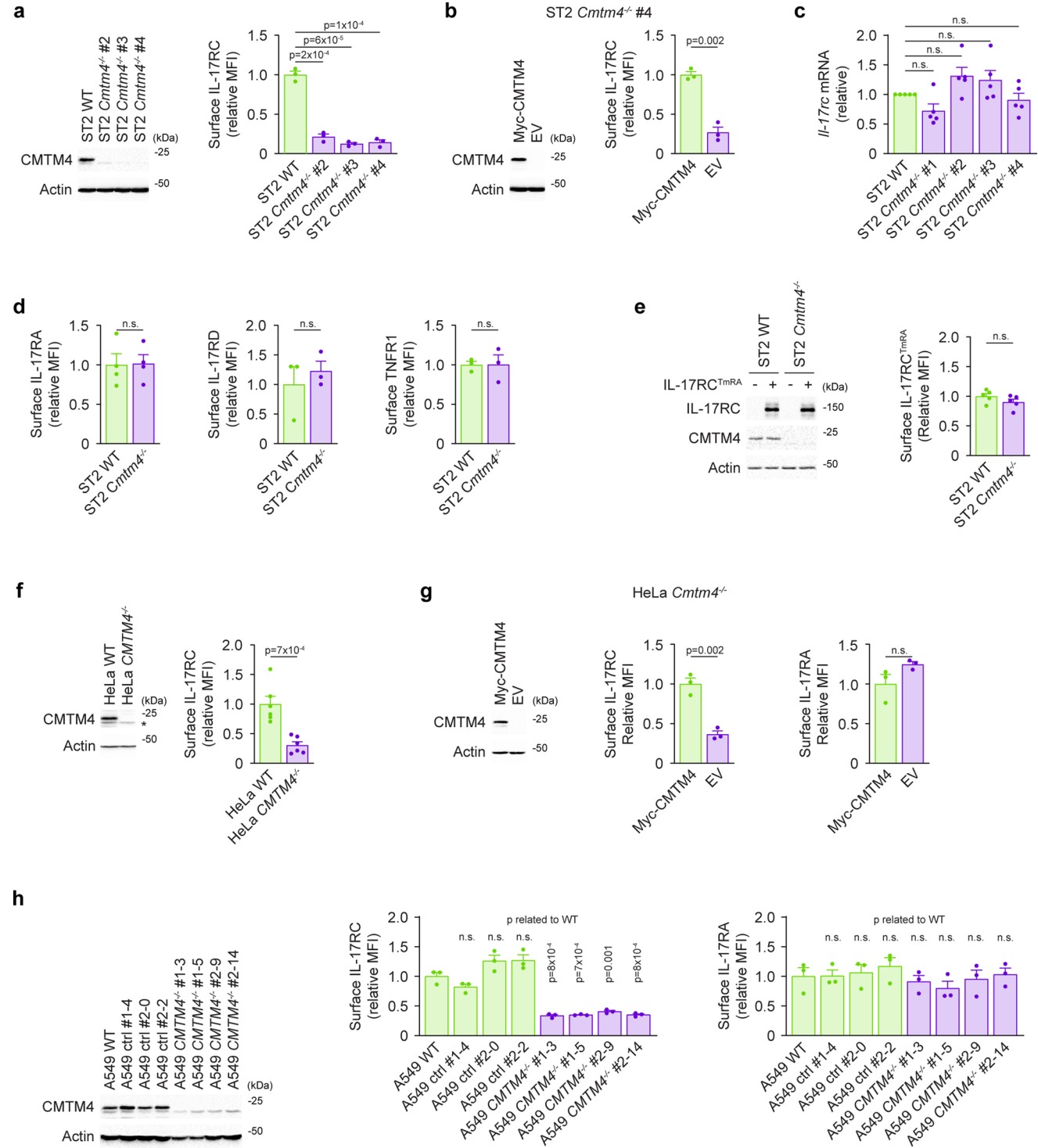

**Extended Data Fig. 2 | CMTM4 regulates IL-17RC plasma membrane localization. a, b,** Immunoblotting analysis of CMTM4 and actin and flow cytometry analysis of IL-17RC surface expression in wild-type or several *Cmtm4*$^{-/-}$ ST2 cell lines (**a**) or *Cmtm4*$^{-/-}$ ST2 cell line #4 transduced with the expression vector coding for Myc-CMTM4 or EV (**b**). **c,** Real-time quantitative PCR analysis of the relative level of *Cmtm4* transcript in wild-type and four *Cmtm4*$^{-/-}$ ST2 cell lines. Mean + s.e.m. from five independent experiments. Two-tailed Mann–Whitney test. **d,** Flow cytometry analysis of IL-17RA, IL-17RD, or TNFR1 surface expression in wild-type and *Cmtm4*$^{-/-}$ ST2 cells. **e,** Immunoblotting of IL-17RC, CMTM4, and actin, and flow cytometry analysis of IL-17RC surface expression in wild-type and *Cmtm4*$^{-/-}$ ST2 cells expressing IL-17RC$^{TmRA}$. **f-h,** Immunoblotting analysis of CMTM4 and actin and flow cytometry analysis of IL-17RC and IL-17RA surface expression in wild-type or *CMTM4*$^{-/-}$ HeLa cells (**f**), *CMTM4*$^{-/-}$ HeLa transduced with the expression vector coding for Myc-CMTM4 or EV (**g**), and A549 wild-type, *CMTM4*$^{-/-}$ or control cell lines where CRISPR/Cas9 transfection did not cause CMTM4 deficiency (**h**). *, unspecific band. Flow cytometry data are based on five (**e**, **f**), four (**d**) or three (**a**, **b**, **d**, **g**, **h**) independent experiments. Mean + s.e.m., two-tailed unpaired t-test. n.s. not significant. MFI, median fluorescence intensity.

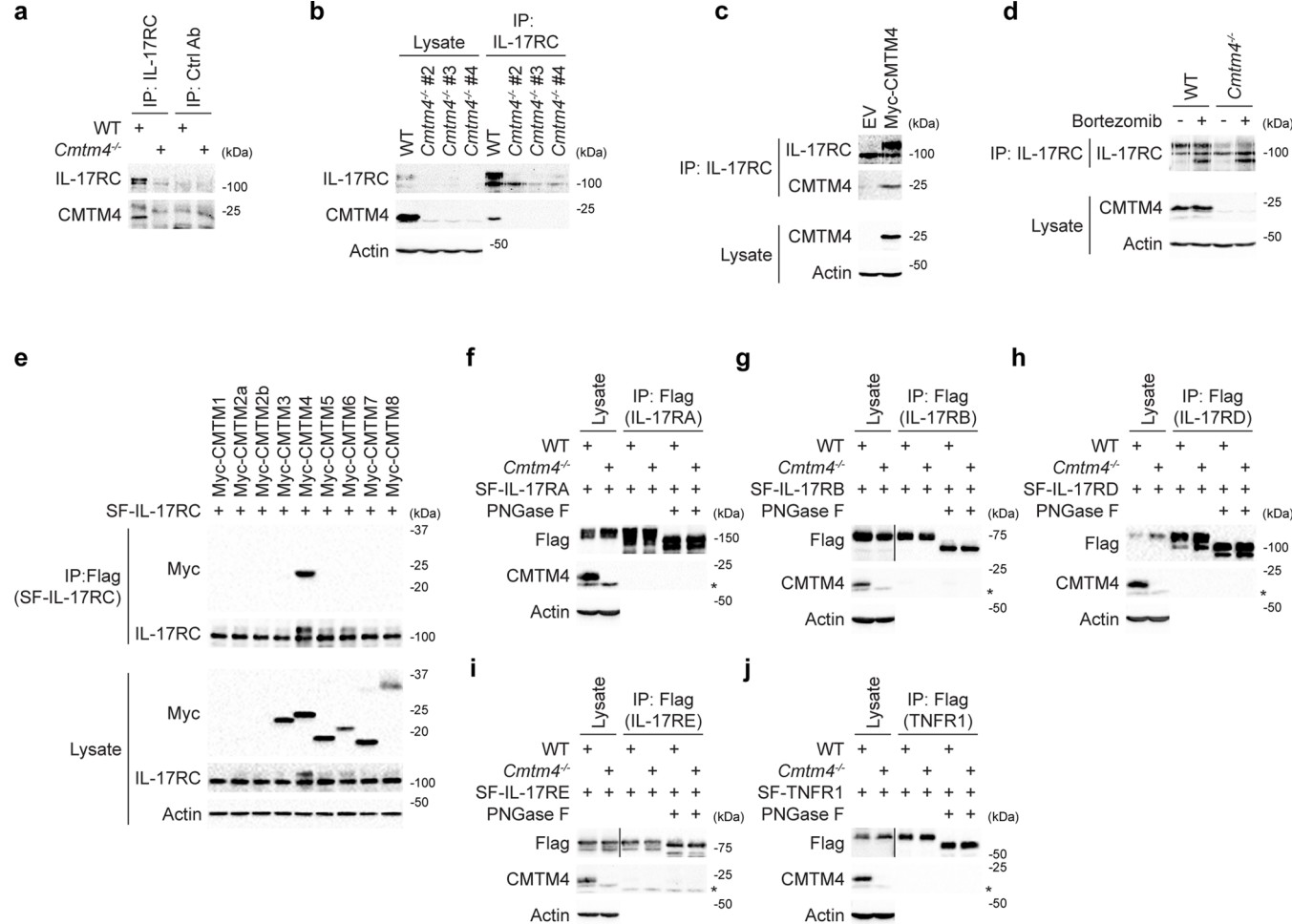

**Extended Data Fig. 3 | CMTM4 mediates IL-17RC glycosylation.**
**a**, Immunoblotting analysis of samples isolated via immunoprecipitation
with IL-17RC antibody or control isotype antibody from lysates of wild-type
or *Cmtm4*$^{-/-}$ ST2 cells. **b,c**, Immunoblotting analysis of samples isolated via
immunoprecipitation with IL-17RC antibody from lysates of wild-type or several
*Cmtm4*$^{-/-}$ ST2 cell lines (**b**) or *Cmtm4*$^{-/-}$ ST2 cells expressing Myc-CMTM4 or EV (**c**).
**d**, Immunoblotting analysis of samples isolated via immunoprecipitation with IL-
17RC antibody from lysates of wild-type or *Cmtm4*$^{-/-}$ ST2 cells treated or not with
proteasome inhibitor bortezomib (10 nM) for 24 h. **e**, Immunoblotting analysis of

samples isolated by Flag immunoprecipitation from lysates of *Cmtm4*$^{-/-}$ ST2 cells
transduced with expression vectors coding for indicated Myc-tagged members
of CMTM family together with the expression vector coding for SF-IL-17RC.
**f-j**, Immunoblotting analysis of samples isolated by Flag immunoprecipitation
from lysates of wild-type or *Cmtm4*$^{-/-}$ ST2 cells expressing SF-IL-17RA (**f**), SF-IL-
17RB (**g**), SF-IL-17RD (**h**), SF-IL-17RE (**i**), or SF-TNFR1 (**j**) and treated or not with
PNGase F. *, unspecific band. Data are representative of four (**a**), three (**b–d**), or
two (**e–j**) independent experiments.

**a**

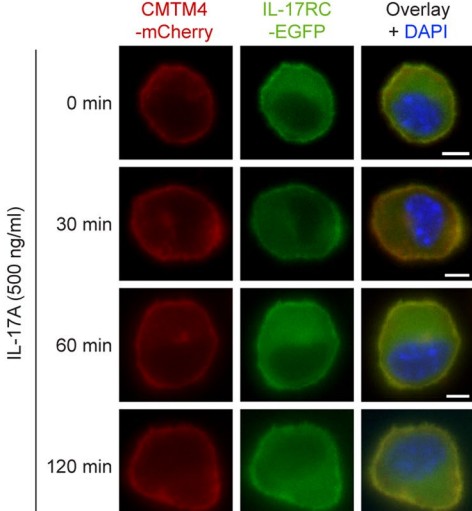

**b**

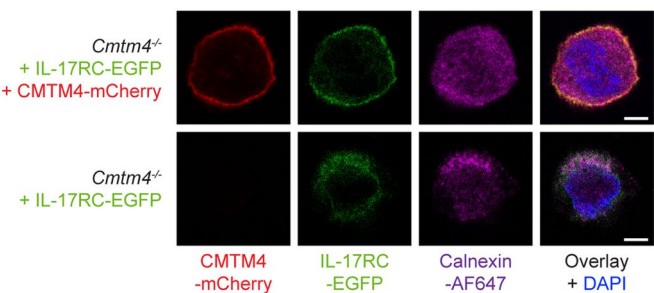

**Extended Data Fig. 4 | CMTM4 regulates IL-17RC trafficking to the plasma membrane. a**, Microscopy analysis of *Cmtm4*^−/−^ ST2 cells expressing IL-17RC-EGFP together with CMTM4-mCherry that were stimulated with IL-17A (500 ng ml⁻¹) for indicated time points. Scale bar, 5 μm. **b**, Confocal microscopy analysis of *Cmtm4*^−/−^ ST2 cells expressing IL-17RC-EGFP only or together with CMTM4-mCherry and stained intracellularly for endoplasmic reticulum marker Calnexin. Scale bar, 5 μm. Data (a,b) are representative of two independent experiments.

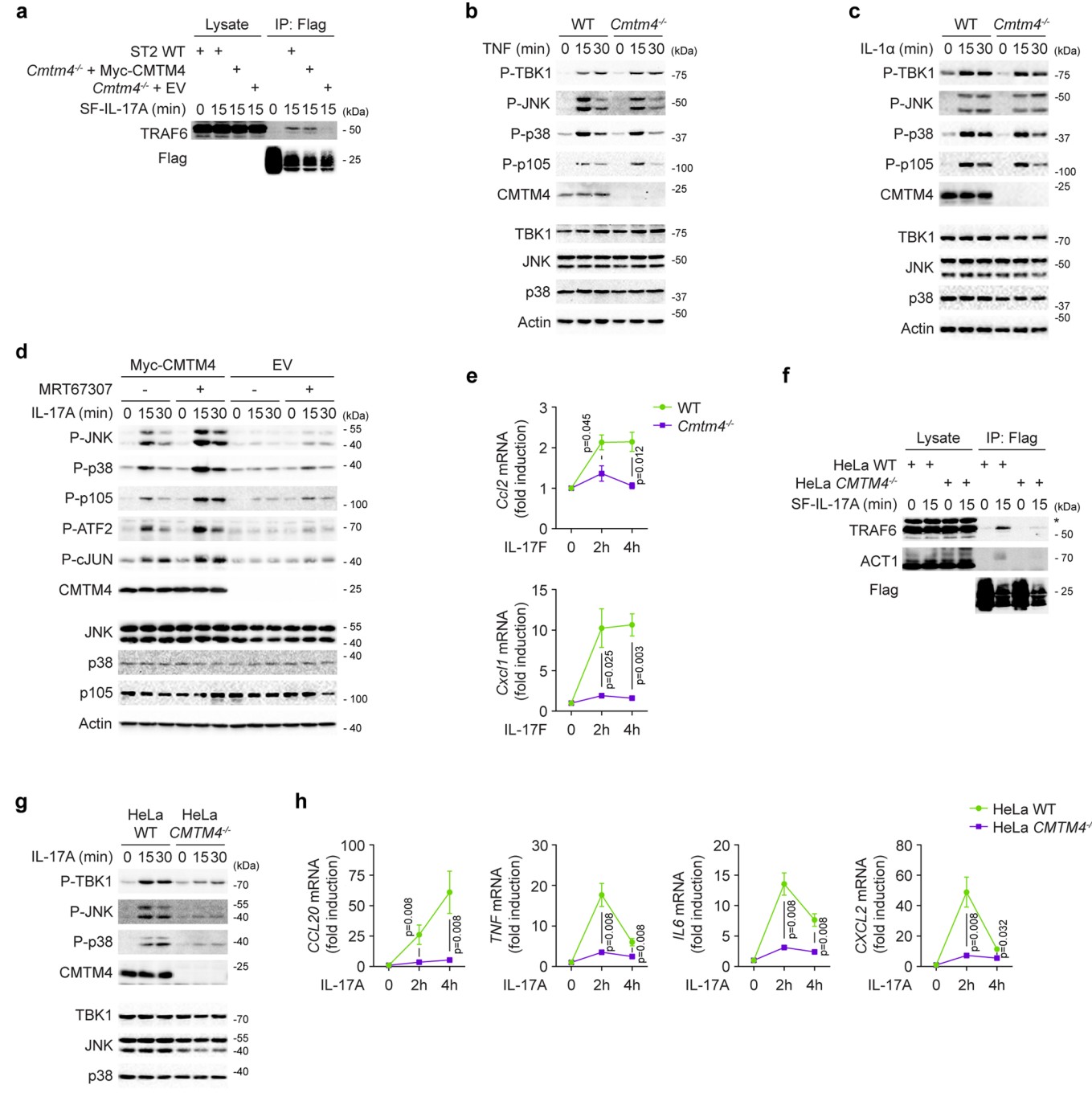

**Extended Data Fig. 5 | CMTM4 is required for IL-17A and IL-17F signaling.**
**a**, Immunoblotting analysis of samples isolated by Flag immunoprecipitation
from lysates of wild-type or *Cmtm4*−/− ST2 cells transduced with the expression
vector coding for Myc-CMTM4 or EV. Cells were stimulated with SF-IL-17A
(500 ng ml−1) for 15 min or left unstimulated and SF-IL-17A was added post lysis.
**b**, **c**, Immunoblotting analysis of lysates of wild-type or *Cmtm4*−/− ST2 cells
stimulated with TNF (50 ng ml−1) (**b**) or IL-1α (50 ng ml−1) (**c**) as indicated. **d**,
Immunoblotting analysis of lysates of *Cmtm4*−/− ST2 cells transduced with the
expression vector coding for Myc-CMTM4 or EV that were pretreated or not with
TBK1 and IKKε inhibitor MRT67307 (2 µM) and stimulated with SF-IL-17A (500 ng
ml−1) for indicated time points. **e**, Real-time quantitative PCR analysis of the
fold induction of *Ccl2* and *Cxcl1* upon stimulation of wild-type or *Cmtm4*−/− ST2

cells with SF-IL-17F (500 ng ml−1) for indicated time points. Mean ± s.e.m. from
three independent experiments. Two-tailed unpaired t-test. **f**, Immunoblotting
analysis of lysates isolated through Flag immunoprecipitation from wild-type
or *CMTM4*−/− HeLa cells that were stimulated with SF-IL-17A (500 ng ml−1) for
15 min or were left unstimulated and SF-IL-17A was added post lysis. *, unspecific
band. **g**, Immunoblotting analysis of lysates of wild-type or *CMTM4*−/− HeLa
cells stimulated with IL-17A (500 ng ml−1) for indicated time points. **h**, Real-time
quantitative PCR analysis of the fold induction of *CCl20*, *TNF*, *IL6*, and CXCL2
upon stimulation of wild-type or *CMTM4*−/− HeLa cells with SF-IL-17F (500 ng ml−1)
for indicated time points. Mean ± s.e.m. from five independent experiments.
Two-tailed unpaired t-test. Data are representative of four (**f**, **g**), three (**b**, **c**), or
two (**a**, **d**) independent experiments.

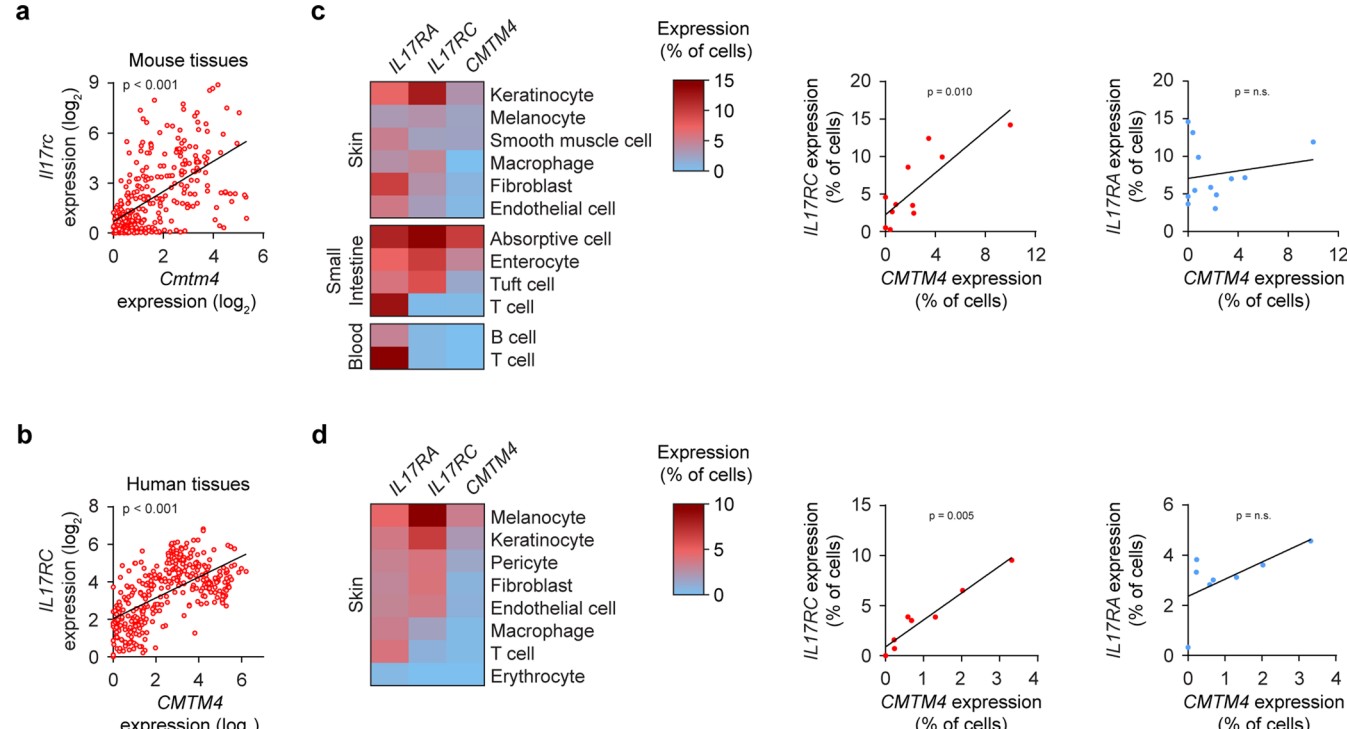

**Extended Data Fig. 6 | CMTM4 and IL-17RC share similar expression patterns.**
**a**, **b**, Relative expression of *Cmtm4* and *Il17rc* in mouse tissues (**a**) or relative expression of *CMTM4* and *IL17RC* in human tissues (**b**) was analyzed using the Genevestigator software. Each dot represents the log₂ of the average expression for different tissues or cell types of the Genevestigator mRNA-Seq Gene Level data. Spearman's correlation test. **c**, **d**, Expression of *IL17RA*, *IL17RC*, and *CMTM4* was analyzed in single-cell RNAseq datasets of human tissues available at the Gene Expression Omnibus. The percentage of cells with at least one detected transcript of *IL17RA*, *IL17RC*, or *CMTM4* in each cell type was visualized in a heatmap (left) or a scatterplot (right). Spearman's correlation test. Dataset GEO accession numbers are GSE159929 (**c**) or GSE130973 (**d**).

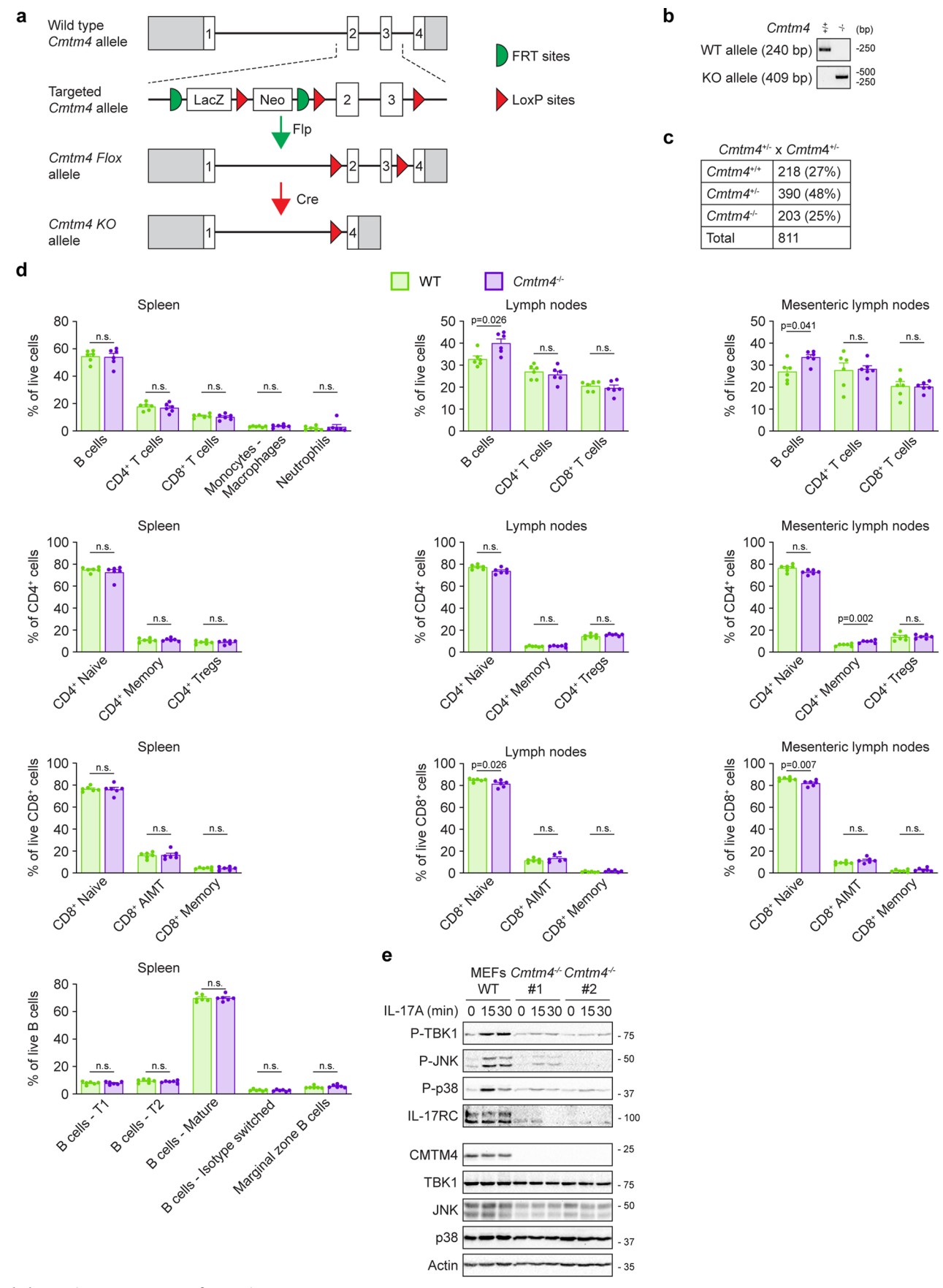

**Extended Data Fig. 7 | See next page for caption.**

**Extended Data Fig. 7 | Characterization of CMTM4-deficient mouse.**
**a**, Schematic representation of *Cmtm4* knockout strategy. *Cmtm4* coding
exons and position of FRT and LoxP sites are indicated. **b**, Example of routine
PCR genotyping of wild-type and *Cmtm4*[−/−] mice. **c**, Number of pups with the
indicated genotype born to *Cmtm4*[+/−] parents. **d**, Flow cytometry analysis of
different immune subtypes in indicated immune organs isolated from 8-12 weeks
old *Cmtm4*[+/+] (WT) or *Cmtm4*[−/−] mice. B cells (CD19[+]) were divided in T1 (IgM[+],
CD23[−], CD1d[−]), T2 (IgM[+], CD23[+], CD1d[−]), marginal zone B cells (IgM[+], CD23[−],
CD1d[+]), mature (IgM[−], IgD[+]), and isotype switched (IgM[−], IgD[−]) cells. Myeloid
cells (CD3[−], CD19[−], NK1.1[−], CD11b[+]) were divided into neutrophils (CD11c[−], Ly6G[+])
and monocytes · macrophages (CD11c[−], Ly6G[−]). CD8[+] T cells were divided in
naive (CD44[−]), memory (CD44[+], CD49d[+]), and antigen-inexperienced memory
T (AIMT) (CD44[+], CD49d[−]) cells. CD4[+] T cells were divided in Tregs (FoxP3[+]),
naive (FoxP3[−], CD44[−], GITR[−]) and memory (FoxP3[−], CD44[+], GITR[+]) cells. n = 6
mice per group in three independent experiments. Mean + s.e.m., two-tailed
Mann−Whitney test. n.s., not significant. **e**, Immunoblotting analysis of lysates of
mouse embryonic fibroblasts (MEFs) isolated from *Cmtm4*[+/+] or *Cmtm4*[−/−] sibling
embryos stimulated with SF-IL-17A (500 ng ml[−1]) for indicated time points. Data
(**e**) are representative of two independent experiments.

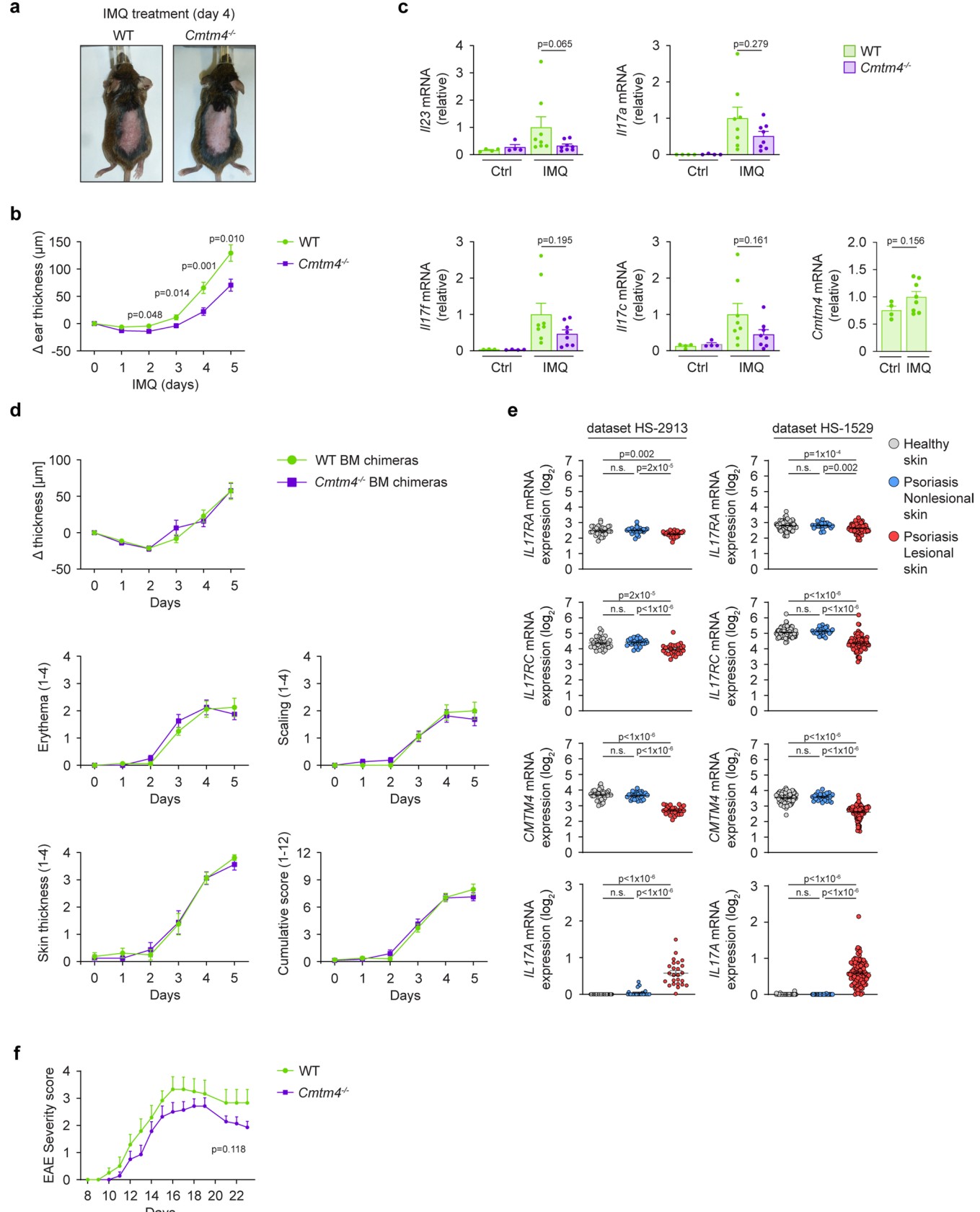

**Extended Data Fig. 8 | See next page for caption.**

**Extended Data Fig. 8 | CMTM4 has a substantial role in propagating experimental psoriasis, but not EAE. a,b,** Analysis of experimental psoriasis disease progression in *Cmtm4*[+/+] (WT) or *Cmtm4*[−/−] littermate mice treated with Aldara cream containing 5% imiquimod (IMQ) on the shaven back and ears daily for five consecutive days and monitored daily. n = 14 mice per group in 3 independent experiments. Photograph of mice on day 4 (**a**) and mean ± s.e.m. for change of ear thickness, two-tailed Mann–Whitney test (**b**). **c,** Real-time quantitative PCR analysis of the relative expression of *Il23, Il17a, Il17f, Il17c,* and *Cmtm4* transcripts in the back skin isolated from *Cmtm4*[+/+] or *Cmtm4*[−/−] littermate mice treated or not with Aldara cream for four consecutive days. n = 4 for control groups, and 8 for IMQ-treated groups. Mean ± s.e.m., two-tailed Mann–Whitney test. **d,** Analysis of experimental psoriasis disease progression in wild-type mice that were transplanted with bone marrow isolated from *Cmtm4*[+/+] or *Cmtm4*[−/−] mice. The experiment was performed as in a. Score for erythema, scaling, dorsal skin thickness, and combined cumulative score and changes in ear thickness are shown. n = 16 mice per group in 2 independent experiments analyzed. Mean ± s.e.m., two-tailed Mann–Whitney test. **e,** Analysis of the expression of *IL-17RA, IL-17RC, CMTM4,* and *IL-17A* mRNA in skin samples from patients with psoriasis (lesional or nonlesional samples) or healthy donors using the Genevestigator software. Each dot represents log$_2$ of the average expression in the given sample. GEO accession numbers are GSE121212 for dataset HS-2913 and GSE54456 for dataset HS-1529. Two-tailed Mann–Whitney test. n.s., not significant. **f,** Clinical scores after MOG$_{35–55}$-induced EAE. Mice were monitored daily in a blinded manner. n = 12 (*Cmtm4*[+/+]) or 14 (*Cmtm4*[−/−]). Mean ± s.e.m., two-way ANOVA.

# nature research

# Reporting Summary

Nature Research wishes to improve the reproducibility of the work that we publish. This form provides structure for consistency and transparency in reporting. For further information on Nature Research policies, see our Editorial Policies and the Editorial Policy Checklist.

## Statistics

For all statistical analyses, confirm that the following items are present in the figure legend, table legend, main text, or Methods section.

| n/a | Confirmed | |
|---|---|---|
| ☐ | ☒ | The exact sample size ($n$) for each experimental group/condition, given as a discrete number and unit of measurement |
| ☐ | ☒ | A statement on whether measurements were taken from distinct samples or whether the same sample was measured repeatedly |
| ☐ | ☒ | The statistical test(s) used AND whether they are one- or two-sided *Only common tests should be described solely by name; describe more complex techniques in the Methods section.* |
| ☐ | ☒ | A description of all covariates tested |
| ☐ | ☒ | A description of any assumptions or corrections, such as tests of normality and adjustment for multiple comparisons |
| ☐ | ☒ | A full description of the statistical parameters including central tendency (e.g. means) or other basic estimates (e.g. regression coefficient) AND variation (e.g. standard deviation) or associated estimates of uncertainty (e.g. confidence intervals) |
| ☐ | ☒ | For null hypothesis testing, the test statistic (e.g. $F$, $t$, $r$) with confidence intervals, effect sizes, degrees of freedom and $P$ value noted *Give P values as exact values whenever suitable.* |
| ☒ | ☐ | For Bayesian analysis, information on the choice of priors and Markov chain Monte Carlo settings |
| ☒ | ☐ | For hierarchical and complex designs, identification of the appropriate level for tests and full reporting of outcomes |
| ☒ | ☐ | Estimates of effect sizes (e.g. Cohen's $d$, Pearson's $r$), indicating how they were calculated |

*Our web collection on statistics for biologists contains articles on many of the points above.*

## Software and code

Policy information about availability of computer code

| Data collection | *Provide a description of all commercial, open source and custom code used to collect the data in this study, specifying the version used OR state that no software was used.* |
|---|---|
| Data analysis | All MS data were analyzed and quantified with the MaxQuant software (version 1.6.5.0) and subsequent data analysis was performed using Perseus 1.6.14.0 software. Cytometry data were analyzed using FlowJo software (TreeStar). |

For manuscripts utilizing custom algorithms or software that are central to the research but not yet described in published literature, software must be made available to editors and reviewers. We strongly encourage code deposition in a community repository (e.g. GitHub). See the Nature Research guidelines for submitting code & software for further information.

## Data

Policy information about availability of data

All manuscripts must include a data availability statement. This statement should provide the following information, where applicable:

- Accession codes, unique identifiers, or web links for publicly available datasets
- A list of figures that have associated raw data
- A description of any restrictions on data availability

The MS proteomics data have been deposited to the ProteomeXchange Consortium via the PRIDE under accession number PXD036366. RNASeq data were downloaded from Gene Expression Omnibus under accession numbers GSE159929, GSE130973, GSE54456, GSE121212. Source data are provided with this paper for all figures.

# Field-specific reporting

Please select the one below that is the best fit for your research. If you are not sure, read the appropriate sections before making your selection.

☒ Life sciences          ☐ Behavioural & social sciences          ☐ Ecological, evolutionary & environmental sciences

For a reference copy of the document with all sections, see nature.com/documents/nr-reporting-summary-flat.pdf

# Life sciences study design

All studies must disclose on these points even when the disclosure is negative.

| | |
|---|---|
| Sample size | All experiments were repeated several times as specified in the manuscript. In the case of graphs, the experiments were repeated at least 5 times to allow statistical analysis via Mann-Whitney test or t test the normality of data via Kolmogorov-Smirnov test. Alternatively for experimental results that were assumed to follow normal distribution, three repetitions were performed and statistical significance was calculated using Student's t-test. In the case of mice cohorts, generally accepted sample size were used based on prior experience with the model. |
| Data exclusions | No data were excluded from analyses. |
| Replication | The number of replication for each experiment is specified in each figure. All experiments were reliably reproduced. In particular, we confirmed all our findings in cell lines of both human and mouse origin. |
| Randomization | Mice were allocated to particular test groups based on their genotype. Littermates expressing or not CMTM4 were compared in blind manner. Experiment using cell lines were not randomized, the design of individual experiments allowed to control for covariants. |
| Blinding | In order to assess the increase of thickness and scaling during experimental psoriasis, mice were randomly distributed in cages prior to the experiment. Mice were photographed daily in a blinded manner to their genotype. Photographs were subsequently analyzed in a blinded manner in random order. In order to assess the severity of EAE, mice were randomly distributed in cages prior to the experiment. The induction of EAE and daily observation of the disease severity were performed in a blinded manner. In the case of experiments using cell lines, the blinding was not performed as we compared different cell lines side by side. |

# Reporting for specific materials, systems and methods

We require information from authors about some types of materials, experimental systems and methods used in many studies. Here, indicate whether each material, system or method listed is relevant to your study. If you are not sure if a list item applies to your research, read the appropriate section before selecting a response.

## Materials & experimental systems

| n/a | Involved in the study |
|---|---|
| ☐ | ☒ Antibodies |
| ☐ | ☒ Eukaryotic cell lines |
| ☒ | ☐ Palaeontology and archaeology |
| ☐ | ☒ Animals and other organisms |
| ☒ | ☐ Human research participants |
| ☒ | ☐ Clinical data |
| ☒ | ☐ Dual use research of concern |

## Methods

| n/a | Involved in the study |
|---|---|
| ☒ | ☐ ChIP-seq |
| ☐ | ☒ Flow cytometry |
| ☒ | ☐ MRI-based neuroimaging |

## Antibodies

| | |
|---|---|
| Antibodies used | Antibodies detecting β-Actin (Cat. number 3700), IκBα (9242), P-p105 (Ser932) (4806), P-p38 (Thr180/Tyr182) (9216), P-JNK (Thr183/Tyr185) (4671), P-TBK1(S172) (5483), Myc (2276), TBK1 (3013), JNK2 (9258), p105 (4717 or 13586) were purchased from Cell Signaling Technology; CMTM4 (HPA014704), FLAG (F3165), GFP (SAB4301138), human IL-17RC (HPA019885) were from Sigma-Aldrich; ACT1 (sc-398161), and p38 (sc-728) was from Santa Cruz Biotechnology; murine IL-17RA (MAB4481), IL-17RC (AF2270), IL-17RD (AF2276), and TNFR1 (AF-425-PB) were from R&D Systems; TRAF6 (ab40675) was from Abcam. Standard dillution of antibodies for immunoblotting was 1:1000. Secondary antibodies Anti-Rabbit IgG (H+L) HRP (711-035-152), Goat Anti-Mouse IgG1 HRP (115-035-205), Goat Anti-Mouse IgG2a HRP (115-035-206), Goat Anti-Mouse IgG2b HRP (115-035-207), Donkey anti-Goat IgG H+L HRP (705-035-147), Goat Anti-Rabbit IgG Fc fragment specific HRP (111-035-046), Goat Anti-Rat IgG (H+L) HRP (112-035-167), Donkey Anti-Rabbit IgG (H+L) AF488 (711-545-152), Donkey Anti-Goat IgG (H+L) AF647 (705-605-147) were purchased from Jackson Immunoresearch, dilution 1:10000 for HRP conjugated antibodies, 1:500 for fluorophore conjugated antibodies. For detection of ACT1, we used Mouse TrueBlot ULTRA (Rockland), dilution 1:5000. Fluorescently labelled antibodies against following antigens were used: CD1d Pe-Cy7 (123524), CD4 PerCP (100538), CD8a BV421 (100738), CD8a BV510 (100752), CD11b BV421 (101251), CD11c AF700 (117320), CD19 PE (115508), CD25 PE-Cy7 (102016), CD44 PE (103008), CD44 PerCP-Cy5.5 (103032), GITR APC (126312), F4/80 PE (123110), IgM BV421 (406518), IgD PerCP-Cy5.5 (405710), |

KLRG1 BV510 (138421), Ly6C PE-Cy7 (128018), Ly6G AF647 (127610), TCRβ PerCP (109212), MCP-1/CCL2 PE (505903), B220 AF700 (103232), TCRβ PE (109208), CD45.1 BV650 (110735), and human IL-17RA-APC (372305) were purchased from BioLegend; CD3 FITC (553062), CD19 FITC (553785), CD25 FITC (553071), CD45.2 FITC (553772), CD49d PE (553157), and NK-1.1 FITC (553164) were from BD Pharmingen; CD23 APC (1108095) was from SONY; FOXP3 PE-Cy7 (25-5773-82) was from eBioscience, Calnexin AF647 (MA3-027-A647) was from Invitrogen. Standard dilution of fluorescently conjugated antibodies was 1:200. LIVE/DEAD Fixable Near-IR Dead Cell Stain Kit was from Thermo Fisher Scientific.

| Validation | Most antibodies used in this study are commonly used by us and other laboratories. The validation of individual antibodies can be find on manufacturer's website. CMTM4, IL-17RC, Act1, TRAF6 antibodies were validated in knockout cell lines. |
| --- | --- |

# Eukaryotic cell lines

Policy information about cell lines

| Cell line source(s) | ST2 cells were kindly provided by Jana Balounova, HeLa, HEK293FT, Phoenix-Eco and Phoenix-Ampho were kindly provided by Tomas Brdicka, and S2 cells were kindly provided by Petr Draber (all from Institute of Molecular Genetics, Prague, Czech Republic). A549 cells were kindly provided by Zora Novakova (Institute of Biotechnology, Vestec, Czech Republic). These cell lines are also commercially available. MEF cells were derived from E11.5 mouse embryos and immortalized by lentiviral infection with the SV40 large T antigen. |
| --- | --- |
| Authentication | Listed cells are commonly used in our lab, no additional authentication was performed. |
| Mycoplasma contamination | All cell lines were regularly tested for mycoplasma contamination via PCR and were mycoplasma negative. |
| Commonly misidentified lines (See ICLAC register) | No commonly misidentified cell lines were used in the study. |

# Animals and other organisms

Policy information about studies involving animals; ARRIVE guidelines recommended for reporting animal research

| Laboratory animals | Frozen sperm from mouse strain C57BL/6N-Atm1Brd Cmtm4tm1a(EUCOMM)Wtsi/WtsiBiat was obtained from The European Mouse Mutant Archive repository (EM:06038) and used for in vitro fertilization. Mice carrying targeted Cmtm4 allele were crossed with Flp-deleter mouse strain B6.Cg-Tg(ACTFLPe)9205Dym/J from The Jackson Laboratory (005703). The resulting mouse strain with exon 2 and 3 flanked by LoxP sites was crossed with Cre-deleter strain Gt(ROSA)26Sortm1(ACTB-cre,-EGFP)Ics (MGI:5285392, Philippe Soriano) to obtain germ line knockout mouse. Animals were kept on C57BL/6J background. All mice used in experiments were 5-12 weeks old. TThey were housed in specific pathogen-free facility 12h/12h light/dark cycle, temperature and relative humidity are maintained at 22 ± 1 °C and 55 ± 5 %, respectively. Both males and females were used for experiments, except for IMQ-induced psoriasis and EAE where female littermate mice were analyzed. If possible, littermates were equally divided into the experimental groups. |
| --- | --- |
| Wild animals | Study did not involve wild animals. |
| Field-collected samples | Study did not involve samples collected from field. |
| Ethics oversight | Animal protocols were approved by the Resort Professional Commission for Approval of Projects of Experiments on Animals of the Czech Academy of Sciences, Czech Republic. |

Note that full information on the approval of the study protocol must also be provided in the manuscript.

# Flow Cytometry

## Plots

Confirm that:

☒ The axis labels state the marker and fluorochrome used (e.g. CD4-FITC).

☒ The axis scales are clearly visible. Include numbers along axes only for bottom left plot of group (a 'group' is an analysis of identical markers).

☒ All plots are contour plots with outliers or pseudocolor plots.

☒ A numerical value for number of cells or percentage (with statistics) is provided.

## Methodology

| Sample preparation | In order to detect surface IL-17RC, cells were transferred to suspension, resuspended in FACS buffer (PBS, 2% FCS, 0.1% NaN3) and stained with anti-IL17RC antibody on ice flowed by APC-labeled secondary antibody. Propidium iodide solution were used for discrimination of live and dead cells.

In order to detect production of CCL2, cells were washed with serum-free DMEM and stimulated for 4 hours with the indicated concentration of IL-17 in the presence of 5 μg/ml Brefeldin A (BioLegend). Cells were collected, fixed and |
| --- | --- |

permeabilized using Cyto-Fast™ Fix/Perm Buffer Set (BioLegend) and stained with fluorescently labeled CCL2 antibody. Samples were measured on BriCyte E6 flow cytometer and data were analyzed using FlowJo software (TreeStar).

Peritoneal lavage was performed using 11 ml of PBS. Cell suspensions were stained with LIVE/DEAD near-IR dye (Life Technologies) and the mixture of primary antibodies (CD45.2, CD11b, F4/80, Ly6G, Ly6C) on ice and analyzed by flow cytometry using Cytek Aurora.

In order to analyze the immune cell populations in WT and CMTM4 KO mice, mice were sacrificed and the spleen, peripheral lymph nodes, and mesenteric lymph nodes were removed and single cell suspensions were prepared. In the case of spleen, red blood cells were lysed in ACK buffer (150 mM NH4Cl, 10 mM KHCO3, 0.1 mM EDTA- Na2, pH 7.4). Cells were resuspended in FACS buffer and stained on ice with LIVE/DEAD near-IR dye (Life Technologies) and stained with ollowing sets of antibodies: T cell compartment (TCRβ, CD4, CD8, CD44, CD49d). B cell compartment (CD19, IgM, IgD, CD23, CD1d). Myeloid compartment (CD3, CD19, NK1.1, CD11b, CD11c, Ly6C, Ly6G). In order to distinguish Tregs and memory CD4 T cells, cells were fixed and permeabilized using Foxp3/Transcription Factor Staining Buffer Set (eBioscience, 00-5523-00) and stained with following antibodies (CD4, CD8, CD25, CD44, FoxP3, GITR).

| | |
|---|---|
| Instrument | BriCyte E6 flow cytometer or Cytek Aurora |
| Software | FlowJo software (TreeStar) |
| Cell population abundance | Retrovirally transduced cells were sorted as GFP positive and the purity was regularly tested via FACS. |
| Gating strategy | Immune cell populations isolated from mice were gated as follows. T cells (TCRβ+) were separated in following subsets: CD4+ T cells (CD4+) and CD8+ T cells (CD8+), which were further divided in naïve CD8+ (CD44-), memory CD8 (CD44+, CD49d+), and antigen-inexperienced memory T cells (AIMT) (CD44+, CD49d-) cells. B cells (CD19+) were separated in following subsets: T1 (IgM+, CD23-, CD1d-), T2 (IgM+, CD23+, CD1d-), marginal zone B cells (IgM+, CD23-, CD1d+), mature (IgM-, IgD+), and isotype switched (IgM-, IgD-).  Myeloid cells (CD3-, CD19-, NK1.1-, CD11b+) were separated in the following subsets: neutrophils (CD11c-, Ly6G+) and monocytes/macrophages (CD11c-, Ly6G-). CD4+ T cells were divided in subsets: Tregs (FoxP3+, CD25+) and CD4+ naïve cells (FoxP3-, CD44-) and CD4+ memory cells (FoxP3-, CD44+).Samples were measured on Cytek Aurora and data were analyzed using FlowJo software (TreeStar).

Cells isolated via peritoneal lavage, samples were first gated as CD45.2+ and subsequently separated in the following subsets: macrophages (CD11b+, F4/80+), neutrophils (CD11b+, Ly6G+) and inflammatory monocytes (CD11b+, Ly6G-, Ly6C+). |

☒ Tick this box to confirm that a figure exemplifying the gating strategy is provided in the Supplementary Information.

