## [Peer Review File · Nature Immunology]

Peer Review Information

Journal: Nature Immunology

Manuscript Title: CMTM4 is a subunit of the IL-17 receptor and mediates autoimmune pathology

Corresponding author name(s): Ondrej Stepanek, Peter Draber

Reviewer Comments & Decisions:

Decision Letter, initial version:

Subject: Decision on Nature Immunology submission NI-LE32255A-Z

Message: 6th Jul 2021

Dear Dr. Draber,

Thank you for your response to the referees' comments on your manuscript "CMTM4 is a subunit of the IL-17 receptor mediating autoimmune pathology". While we find your work of considerable potential interest, the reviewers have raised substantial concerns that must be addressed. As such, we cannot accept the current version of the manuscript for publication, but would be happy to consider a revised version that addresses these concerns, as long as novelty is not compromised in the interim.

Please revise along the lines specified in your letter and to address all issues raised by the referees. At resubmission, please include a "Response to referees" detailing, point-by-point, how you addressed each referee comment. If no action was taken to address a point, you must provide a compelling argument. This response will be sent back to the referees along with the revised manuscript.

Please include a revised version of any required reporting checklist. It will be available to referees to aid in their evaluation. The Reporting Summary can be found here: <https://www.nature.com/documents/nr-reporting-summary.pdf>

You may use the link below to submit your revised manuscript and related files:
[REDACTED]

We hope to receive the revised manuscript within 6 months. If you cannot send it within this time, please let us know. We will be happy to consider your revision so long as nothing similar has been accepted for publication at Nature Immunology or published elsewhere.

Nature Immunology is committed to improving transparency in authorship. As part of our efforts in this direction, we are now requesting that all authors identified as 'corresponding author' on published papers create and link their Open Researcher and Contributor Identifier (ORCID) with their account on the Manuscript Tracking System (MTS), prior to acceptance. ORCID helps the scientific community achieve unambiguous attribution of all scholarly contributions. You can create and link your ORCID from the home page of the MTS by clicking on 'Modify my Springer Nature account'. For more information please visit www.springernature.com/orcid.

Thank you for the opportunity to review your work.

Sincerely,

Ioana Visan, Ph.D.
Senior Editor
Nature Immunology

Tel: 212-726-9207
Fax: 212-696-9752
www.nature.com/ni

Reviewers' Comments:

Reviewer #1:

Remarks to the Author:

In this manuscript, Knizkova et al. exhibited in vitro and in vivo evidence that CMTM4 is an important component of the IL-17 receptor complex (IL-17RSC) mediating autoimmune pathology. Through tandem affinity purification of tagged IL-17A followed by mass-spectrometry, the authors detected CMTM4 as an unreported component of IL-17RSC. The authors hypothesized that CMTM4 stabilizes IL-17RC and promotes its trafficking into the plasma membrane given the fact the IL-17RC on the surface membrane was drastically reduced upon CMTM4 deficiency. Although the hypothesis of this manuscript is

interesting/innovative and phenotypes observed with CMTM4 KO cells are impressive, the authors did not provide sufficient data to draw convincing conclusions. Especially, the authors did not provide enough molecular details on how CMTM4 affects IL-17RC glycosylation that is critical for surface transportation. In addition, the phenotype of CMTM4 KO mice is mild on the experimental psoriasis model (is it because CMTM4 also affect other negative regulator such as PD-L1). Because the paper is making a big claim on the role of CMTM4 in IL-17 signaling, the authors need to provide more experimental data/evidence to address these concerns. Below are major concerning points:

1. Since CMTM4 affects IL17RC surface level, the authors should also investigate IL17F-induced signaling and gene expression since it is the preferred ligand for IL17RC.
2. Does CMTM4 also have impact on other receptor signaling (e.g. TNFR, IL1R/TIR)?
3. The phenotype of *Cmtm4*^{-/-} mice on imiquimod-induced psoriasis is not impressive (Fig 4). Please also show histology analysis and IL-17 target gene expression from the affected tissue (include control animals for both genotypes).
4. To strengthen the study, it is critical to Use another IL-17-dependent disease model (e.g., Th17 adopter transfer EAE model) to examine the phenotype of *Cmtm4*^{-/-} mice since this is the well-accepted disease model to study IL-17 signaling in vivo.
5. The authors showed that CMTM4 interacts only with IL-17RC but not IL-17RA (Fig. 1C). What would be the explanation for such specificity? (We know that IL-17RC should be associated with IL17RA in the IL17RSC).
6. Please map the domain/motif on CMTM4 that interact with IL17RC. What is the nature of the interaction-direct or indirect? Is it possible to do a pull-down experiment with purified proteins?
7. The authors showed endogenous IP between IL-17RC and CMTM4 in Fig 2C. Does IL-17A treatment have any impact on IL-17RC/CMTM4 interaction? Please show CMTM4 in Fig 3a.
8. Fig 3a legend should clearly indicate the cells were transfected with SF-IL17RC. Also, why no SF-17RC level were shown for whole cell lysate?
9. The authors found that IL-17RC is glycosylated and this process was CMTM4-dependent. This part of the study is very superficial, as authors did not provide the mechanism. There are several unanswered questions: What is the role of the glycosylation in this interaction? What sites on IL-17RC are N-glycosylated?; How does CMTM4 impact the glycosylation of IL17RC? -Please give detailed discussion and provide supporting evidences.
10. In Fig 3D the authors showed the mRNA level for several cytokines//chemokines in WT and CMTM4 KO. Please show the protein levels as well (e.g., ELISA).
11. Please discuss why the level of CMTM4 is greatly reduced in IL-17RC KO (Ext Fig 3B).

Reviewer #2:

Remarks to the Author:

IL-17 signaling drives pathology in autoimmune and pathologic inflammatory conditions, and the IL-17 receptor family comprises a unique subclass of receptors that are only

distantly related to better-studied cytokine receptors. As such, their fundamental biology is still surprisingly poorly understood, and therefore this topic is of considerable significance. IL-17(A) signals mainly through a heterodimer of IL-17RA and IL-17RC (though a few recent studies suggest alternative receptor configurations, notably IL-17RD). IL-17RC is generally believed to be non-redundant for IL-17 function, yet its expression patterns are very different from IL-17RA, and we know very little about its regulation.

In order to identify new mediators of IL-17 signaling that interact with the IL-17 receptor, Knizkova et al. performed mass-spectrometry analysis of an overexpressed Strep-Flag-tagged IL-17R complex (SF-IL-17R) in ST2 cells, a mouse stromal cell line often utilized for signaling studies. Their data reassuringly confirm association with numerous known signaling molecules (Act1, TRAFs, etc.), which lends confidence to the approach. They also identified the tetraspanning transmembrane protein CMTM4, the subject of this paper, which was recently shown to regulate PD-L1 protein levels. They use overexpressed chimeric receptors to show that CMTM4 constitutively associates with IL-17RC through its transmembrane domain, and appears to be essential for IL-17RC cell surface expression. CMTM4 depletion in several IL-17-responsive cells (Hela, ST2, MEFs) led to impaired IL-17 signaling that correlated with decreased IL-17RC surface expression. *Cmtm4*^{-/-} mice were partially resistant to imiquimod (IMQ)-induced experimental psoriasis, a model system known to be IL-17R-dependent. Collectively, the authors propose a model whereby CMTM4 enables IL-17 signaling by permitting cell surface expression of IL-17RC.

The overall findings of this study are important to the field, novel, and logically presented. A role for CMTM4 in the IL-17 signaling is especially intriguing, and could help to explain the unresolved questions about why IL-17RA and IL-17RC are distinctly expressed despite both being required to mediate IL-17 signaling. Data are performed clearly and convincingly, with exceptions noted below. However, the manuscript has some important shortcomings in its current state. The mechanism of how CMTM4 acts to regulate IL-17RC levels are unclear and findings are often correlative – glycosylation of IL-17RC is affected in absence of CMTM4, but is that the deciding determinant of surface expression? It seems unlikely that IL-17RC glycosylation by CMTM4 is unique to IL-17RC, but the specificity of this process on other receptors (especially IL-17RA and IL-17RD) needs to be clarified. Nor is it clear whether this is regulated by IL-17 or is a tonic event that establishes the capacity of a cell to respond to IL-17 in the first place. Furthermore, while *Cmtm4*^{-/-} mice are protected from IMQ-induced psoriasis, there are important missing controls, particularly related to how much of the effects are actually IL-17-driven events. Collectively, while CMTM4 emerges as a new component of the IL-17 receptor signaling, the underlying mechanism remains insufficiently supported.

Major points

1. While CMTM4 clearly reduces IL-17RC surface expression, this may simply be due to an overall decrease in IL-17RC total protein levels, which needs to be determined. Are IL-17RC mRNA levels affected in absence of CMTM4? (this would not be predicted based on PD-L1 findings but is very easy to test; if they are affected, then that would raise a very different scenario of how IL-17RC is controlled).
2. A major conclusion of this paper is that CMTM4 affects the glycosylation and by implication, the membrane expression of IL-17RC. It is unclear how this is happening at a mechanistic level. (1) No information is provided about whether N-glycosylation regulates IL-17RC protein stability/turnover/localization/internalization or some combination. The turnover rate for non-glycosylated IL-17RC compared to glycosylated IL-17RC in the presence of a protein synthesis inhibitor should be assessed. (2) the authors need to

address more precisely how CMTM4 affects IL-17RC glycosylation. Since most glycosylation reactions are regulated by Golgi pH, membrane integrity and cellular stress, it is important to check whether CMTM4 regulates these events and affects glycosylation more broadly. Along these lines, a much more controlled use of inhibitors (e.g., Endo H; O-Glycosidase) could potentially provide more information about the glycosylation type. (3) is the binding of CMTM4 to IL-17RC dependent on N-glycosylation sites? It does seem that inhibition of N-glycosylation decreases the interaction between CMTM4 and IL-17RC (Fig. 2d). Therefore, the binding of CMTM4 to IL-17RC with site-directed amino acid mutation of potential N-glycosylation site should be performed, which should be straightforward.

3. The authors performed receptor-complex analysis by stimulating cells with a recombinant Strep-Flag-IL-17 (SF-IL-17). The entire signaling complex is then isolated upon cell lysis via tandem affinity purification of the ligand and analyzed by MS. However, negative controls in this experiment were not clearly articulated. First, if only unstimulated cells were used as negative controls, they are not in the same activation state as those stimulated – treating cells with standard recombinant IL-17A would be a more appropriate negative control. Second, the authors need to verify that the SF-IL-17 when added only after cell lysis does not interact with CMTM4. Third, the authors use very high doses of SF-IL-17 to stimulate cells (500 ng/ml), and they do not compare results to commercial rIL-17. This is essential. Since IL-17RC is actually thought to bind more to IL-17F than to IL-17A, they also need to include IL-17F, and ideally the IL-17AF heterodimer. IL-17C is also implicated in IMQ-psoriasis (though mostly in keratinocytes) so this should also be examined.

4. A more comprehensive description of the changes in gene expression in the absence of CMTM4, with and without IL-17 stimulation, is required in these experiments (Fig 3). Moreover, how was the fold induction calculated in Fig 3d ? Were all samples normalized to WT untreated cells?

5. What is the expression of IL-17RA in the absence of CMTM4? This is a key control that is lacking in many of these experiments. In a related manner, is IL-17RD affected? (this was recently reported to be an alternative co-receptor in the IL-17R complex)

6. The authors report that CMTM4 and IL-17RC interact constitutively, but whether this interaction is affected by IL-17 (A, F, AF) stimulation was not assessed.

7. *Cmtm4*^{-/-} mice were partially, but not fully resistant to IMQ-induced experimental psoriasis, reminiscent of a recent report describing IL-17RD in this process (PMID 31175175). However, there is little to prove that this phenotype is a consequence of CMTM4 regulating IL-17 signaling as opposed to any other receptor. The authors could look at the expression of IL-17A- F- and IL-17C-dependent target genes implicated in this process, for example. Does CMTM4 regulate neutrophil infiltration during psoriasis?

8. It is unclear if the effects in *Cmtm4*^{-/-} mice arise from effects on the IL-17 target cells (in this model, keratinocytes) or alternatively could be linked to a dysfunctional hematopoietic compartment (e.g., deficiency in IL-17 producing cells such as gamma-delta T cells, ILC3 or Th17 cells). This could readily be addressed with bone marrow chimeras and followed up with specific conditional KOs. In addition, *Il17a*, *Il17f*, *Il17c* and *Il23* mRNA levels in IMQ-treated mice of both genotypes should be evaluated to assess potential contributions for CMTM4 in IL-17-producing cells.

9. Immunofluorescence images in Fig. 2e-f are low resolution and do not reveal much about specific subcellular compartments. Staining should be done in both untreated and IL-17-treated cells.

10. Information about CMTM4 expression in psoriasis patients would provide clinical support.

11. The chimeric receptors studies in Extended Fig 3 are clever and would be appropriate in the main text. However, they do not demonstrate that each chimeric protein is expressed at similar levels on the cell surface, which is essential to validate conclusions

12. Total protein level controls of TBK1, p38, JNK, p105 are missing from the majority of western blot figures. This is essential to include as changes in their protein expression could affect their phosphorylation.

13. Authors claim that *Cmtm4*^{-/-} mice do not present major changes in their immune cell compartment sat baseline. However, the percentage of B cells is increased and the percentage of CD8⁺ T cells is decreased in *Cmtm4* KO mice when compared to WT mice. Please reconcile.

Minor

1. The expression levels of CMTM4 in skin of IMQ-treated WT mice compared to control mice should be evaluated.

2. Do CMTM4 and IL17RC deficient cells generated via CRISPR/Cas9 originate from a single clone? If this is the case, did the authors check multiple clones to avoid potential inter-clonal variation?

3. Extended Fig 2d (PNGase) should be in main figures as this is an important point.

4. Extended fig 3b is unnecessary -- there is no need to re-prove that IL-17RC is needed for signaling

Author Rebuttal to Initial comments

CMTM4 is a subunit of the IL-17 receptor mediating autoimmune pathology Knizkova et al.

Response to reviewer questions

Reviewer #1

(Remarks to the Author)

In this manuscript, Knizkova et al. exhibited *in vitro* and *in vivo* evidence that CMTM4 is an important component of the IL-17 receptor complex (IL-17RSC) mediating autoimmune pathology. Through tandem affinity purification of tagged IL-17A followed by mass-spectrometry, the authors detected CMTM4 as an unreported component of IL-17RSC. The authors hypothesized that CMTM4 stabilizes IL-17RC and promotes its trafficking into the plasma membrane given the fact the IL-17RC on the surface membrane was drastically reduced upon CMTM4 deficiency. Although the hypothesis of this

manuscript is interesting/innovative and phenotypes observed with CMTM4 KO cells are impressive, the authors did not provide sufficient data to draw convincing conclusions. Especially, the authors did not provide enough molecular details on how CMTM4 affects IL-17RC glycosylation that is critical for surface transportation. In addition, the phenotype of CMTM4 KO mice is mild on the experimental psoriasis model (is it because CMTM4 also affect other negative regulator such as PD-L1). Because the paper is making a big claim on the role of CMTM4 in IL-17 signaling, the authors need to provide more experimental data/evidence to address these concerns. Below are major concerning points:

We would like to thank the Reviewer for the appreciation of our hypothesis presented in this manuscript and also for valuable comments on how to further improve and strengthen our experimental evidence.

1. Since CMTM4 affects IL17RC surface level, the authors should also investigate IL17F-induced signaling and gene expression since it is the preferred ligand for IL17RC.

In accord with published literature, IL-17F is weak activator of signaling responses compared to IL-17A (Zheng, Y., et al. Nat Med. 2008 Mar;14(3):282-9). Stimulation of wild-type ST2 cells with IL-17F leads to very weak response at or even below the detection limit of the Western blot methodology:

For that reason, we decided not to include data from Western blot analysis of signaling pathways comparing WT and CMTM4 knockout cell lines. Instead we focused on more quantitative and sensitive real-time PCR analysis of transcription of Cxcl1 and Ccl2, which are readily detected upon IL-17F stimulation, albeit at approximately 10-times lower levels than in the case of IL-17A. Indeed, we demonstrated that cells deficient in CMTM4 were severely compromised in their ability to trigger transcription of these cytokines upon IL-17F stimulation. These data are now included in the manuscript and provide further support for the role of CMTM4 as a major regulator of IL-17RC-mediated signaling (newly added Extended Data Fig. 6e).

2. Does CMTM4 also have impact on other receptor signaling (e.g. TNFR, IL1R/TIR)?

We have previously analyzed the composition of TNFR1 signaling complex using similar approach as presented in this manuscript, i.e. we purified the complex via Strep-Flag (SF-) tagged TNF and subjected the samples to mass-spec analysis. We did not detect CMTM4 or any other member of CMTM family as a part of

TNFR1 signaling complex, which suggested that TNF signaling is not regulated by CMTM4 (Lafont, Draber et al. *Nat Cell Biol.* 2018 Dec;20(12):1389-1399).

In order to provide further evidence that CMTM4 does not regulate TNFR1 function, we first demonstrated by flow cytometry that there is no difference in TNFR1 expression between ST2 WT and CMTM4 KO cell lines (newly added Extended Data Fig. 3a). Subsequently, we transfected WT or CMTM4deficient cells with SF-tagged TNFR1. We isolated SF-TNFR1 via anti-Flag immunoprecipitation and treated samples with PNGase F to analyze the glycosylation of TNFR1 in these two cell types. We showed that CMTM4 does not interact with TNFR1 and does not impact its glycosylation (newly added Extended Data Fig. 4h). In accord, the analysis of signaling pathways triggered upon TNF stimulation showed no significant difference between WT and CMTM4 KO cells (newly added Extended Data Fig. 6b). Moreover, we demonstrated that CMTM4 is not required for IL-1 α signaling (newly added Extended Data Fig. 6c). Altogether, these data confirmed that CMTM4 is not a general regulator of signaling from proinflammatory receptors.

3. The phenotype of *Cmtm4*^{-/-} mice on imiquimod-induced psoriasis is not impressive (Fig 4). Please also show histology analysis and IL-17 target gene expression from the affected tissue (include control animals for both genotypes).

We have repeated the psoriasis experiments to provide these additional data and we performed the histological analysis of ear sections and of the expression of IL-17A target genes in the psoriatic skin. H&E staining of ears from control and IMQ-treated animals revealed substantial increase of skin thickness in IMQ-treated WT animals, while it was much less pronounced in CMTM4-deficient animals (newly added Fig. 4e). The analysis of proinflammatory genes transcription by real-time PCR revealed significantly stronger upregulation of IL-17-responsive genes, such as *DefB3*, *S100a8*, or *S100a9* in the psoriatic skin in WT mice in comparison to CMTM4 KO animals (newly added Fig. 4f). Altogether, these newly added experiments provide further evidence for the role of CMTM4 in IMQ-induced psoriasis.

4. To strengthen the study, it is critical to Use another IL-17-dependent disease model (e.g., Th17 adopter transfer EAE model) to examine the phenotype of *Cmtm4*^{-/-} mice since this is the well-accepted disease model to study IL-17 signaling in vivo.

We have performed the classical EAE model (immunization with MOG peptide, CFA, and pertussis toxin) in *Cmtm4*^{-/-} mice and WT littermates. Although we observed a mild improvement in the EAE progression in CMTM4 KO mice, the differences were not statistically significant (newly added Extended Data Fig. 8f).

It should be noted that, in contrast to psoriasis, the role of IL-17 in EAE progression is still debated. IL-17 was reported both to be an important driver of EAE (Komiyama et al. *J Immunol.* 2006 Jul 1;177(1):56673), but also to have a negligible effect similar to the one we observed in this study (Haak et al. *J Clin Invest.* 2009 Jan;119(1):61-9). Recently, this discrepancy has been resolved by the observation that IL-17 enhances the sensitivity to the EAE via an indirect mechanism involving long-term changes in the microbial colonization of the intestine (Regen et al. *Sci Immunol.* 2021 Feb 5;6(56):eaaz6563). This recent

study actually demonstrated that IL-17 deficient mice are largely protected from EAE progression only when housed separately from controls. However, when IL-17 deficient and WT animals were co-housed, the protective role of IL-17 was strongly diminished, due to the normalization of the microbiota in these two strains. In all our *in vivo* experiments, we used littermate controls born to *Cmtm4*^{+/-} parents and both *Cmtm4*^{-/-} and WT animals shared cages since birth. This setup disables any divergence in the intestinal bacterial colonization, which explains the relatively mild effect of CMTM4 in the EAE progression. Taken these new findings into account, the IMQ-induced psoriasis is the only mouse autoimmune model, which is, to our knowledge, unanimously established as being dependent on the direct role of IL-17.

We would like to stress that both induced autoimmune models in our study are very complex and several proinflammatory pathways contribute to the progression of the disease. Notably, IMQ-based model of psoriasis is induced by the activation of a strong pro-inflammatory TLR7 pathway. For this reason, blocking one particular signaling pathway, in this case IL-17A signaling, is unlikely to make animals completely resistant as has been previously documented in the literature by showing that even ablation of IL-17RA does not completely prevent IMQ-induced skin inflammation (El Malki et al. *J Invest Dermatol.* 2013 Feb;133(2):441-51 and Moos et al. *J Invest Dermatol.* 2019 May;139(5):1110-1117). Similar data showing improved, but not abolished, progression of IMQ-induced psoriasis were also shown for animals deficient in a critical component of IL-17A signaling pathways *Act1* (Ha et al. *PNAS* August 19, 2014 111 (33) E3422-E3431) and animals deficient in IL-17RC (Su et al. *Sci Immunol.* 2019 Jun 7;4(36):eaau9657).

Therefore, besides these disease models, we measured the leukocyte infiltration upon the administration of IL-17A in the peritoneum, in order to directly address the *in vivo* response to IL-17A in *Cmtm4*^{-/-} mice (Fig. 4a). In this well-controlled model, the IL-17A-induced responses are largely compromised in the *Cmtm4*^{-/-} mice, which provide direct *in vivo* evidence that CMTM4 is required for IL17A-induced signaling.

5. The authors showed that CMTM4 interacts only with IL-17RC but not IL-17RA (Fig. 1C). What would be the explanation for such specificity? (We know that IL-17RC should be associated with IL17RA in the IL17RSC).

IL17RA and IL17RC do not interact with each other in unstimulated cells and are spatially separated on cell surface. Upon addition of dimeric IL-17A, the two receptors are crosslinked, which leads to the recruitment of intracellular signaling molecules (Hu, Y., et al. *J Immunol*, 2010 Apr 15;184(8):4307-16). In accord, our mass-spec analysis of proteins associated directly with IL-17RC in unstimulated cells revealed a strong interaction with CMTM4, but not IL-17RA (Table 2).

Our data demonstrate that IL17RC, but not IL-17RA, interacts strongly with CMTM4 via its transmembrane domain (Fig. 1e). The amino acid sequence comparison between IL-17RC and IL-17RA transmembrane domains show that they are indeed very different (newly added Extended Data Fig. 1i). We further show that CMTM4 interaction with IL-17RC is highly specific, as it does not bind a variety of other transmembrane proteins, such as IL-17RB, IL-17RD, IL-17RE, or TNFR1 (newly added Extended Data Fig. 1d, 4d-h).

Upon the stimulation of cells with IL-17A, the interaction between IL-17RC and CMTM4 is preserved (newly added Fig. 1d and Extended Data Fig. 1g) and CMTM4, together with IL-17RC, becomes a part of the IL-17

receptor signaling complex (IL-17RSC) together with IL-17RA (Fig. 1a, 3a). In order to clarify this point, we decided to provide schematic representation of our results as Extended Data Fig. 9.

6. Please map the domain/motif on CMTM4 that interact with IL17RC. What is the nature of the interaction-direct or indirect? Is it possible to do a pull-down experiment with purified proteins?

In order to map the IL-17RC-interaction domain on CMTM4, we made several chimeric molecules between CMTM4 and its paralogue CMTM6, which is not a component of the IL-17RSC and does not bind IL-17RC. We swapped individual transmembrane domains of CMTM4 for the ones from CMTM6 and measured the binding of these mutants to IL-17RC. Our data demonstrated that the binding of IL-17RC is mediated collectively by the first two transmembrane domains of CMTM4 (newly added Fig. 1f).

Concerning the nature of the interaction, all the evidence supports the view that the interaction is direct. The stoichiometry between IL-17RC and CMTM4 is nearly 1:1 in both unstimulated cells and in IL17RSC (Fig. 1a, c and newly added Extended Data Fig. 1e). Moreover, if there were any additional protein(s) linking CMTM4 and IL-17RC, we would very likely detect it/them in our pull-downs followed by mass-spectrometry. However, our mass-spec analysis of both the IL-17RC interactome and the IL-17RSC did not reveal any additional transmembrane proteins that could be responsible for mediating the interaction between the transmembrane domain of IL-17RC and CMTM4 (Table 1 and 2 and newly added Extended data Fig. 1e).

Both IL-17RC and CMTM4 are transmembrane proteins and the interaction of their membrane domains is crucial for their interaction, which might require a specific environment in the membrane of the compartments of the secretory pathway. To directly test whether IL-17RC and CMTM4 can interact as isolated proteins after cell lysis, we performed the following experiment. Cells deficient in CMTM4 were transfected with SF-tagged IL-17RC and subjected to cellular lysis. Subsequently, we mixed this lysate with the lysate from WT cells and we immunoprecipitated SF-IL-17RC via Flag antibody. In this setup, SF-IL17RC was unable to interact with CMTM4 post-lysis:

Lysates from ST2 CMTM4 KO cells transfected with SF-IL-17RC were mixed with lysates from WT and subjected to anti-Flag immunoprecipitation. As a control, we used lysates from ST2 WT cells expressing SF-IL-17RC. Samples were analyzed by immunoblotting. Data are representative of two independent experiments.

These data exclude the possibility that the interaction of IL-17RC with CMTM4 is an artifact happening post-lysis. Instead, the CMTM4-IL-17RC complex assembly requires specific environment in intact cellular

membranes. However, this experiment also demonstrates that it is not feasible (or at least beyond our expertise) to use purified proteins for studying this interaction.

Therefore, we decided to study the interaction between CMTM4 and IL-17RC in transfected insect S2 cells. The logic behind this experiment is that the insect cells are evolutionarily distant from vertebrate cells and do not have IL-17 signaling components (Huang et al. PLoS One. 2015 Jul 28;10(7):e0132802). It is very unlikely that these cells would express a hypothetical protein or proteins assisting the assembly of IL-17RC and CMTM4 in mammalian cells (in the hypothetical case of the indirect character of the CMTM4-IL-17RC interaction). We expressed either SF-IL-17RA or SF-IL-17RC together with CMTM4-Myc in S2 cells and subsequently we isolated the receptors via anti-Flag immunoprecipitation. In accord with our previous data, only IL-17RC, but not IL-17RA could bind to CMTM4 (newly added Extended Data Fig. 1f).

Altogether, our data strongly indicate that the interaction between IL-17RC and CMTM4 is direct.

7. The authors showed endogenous IP between IL-17RC and CMTM4 in Fig 2C. Does IL-17A treatment have any impact on IL-17RC/CMTM4 interaction? Please show CMTM4 in Fig 3a.

We have now added several figures in which we show that the association between IL-17RC and CMTM4 is constitutive and is not changed upon IL-17A stimulation. (1) We show that CMTM4 coprecipitated with endogenous IL-17RC in both unstimulated and IL-17A stimulated cells. Isotype antibody was used for the immunoprecipitation as negative control (newly added Fig. 1d). (2) We expressed SF-IL17RC or empty vector in WT ST2 cells. Subsequently cells were stimulated with IL-17A and subjected to anti-Flag immunoprecipitation. CMTM4 specifically co-immunoprecipitated with SF-IL-17RC in lysates from both unstimulated and IL-17A stimulated cells (newly added Extended Data Fig. 1g). (3) We show that IL-17RC expression and glycosylation is substantially reduced in CMTM4-deficient cells as compared to control cells. The glycosylation of IL-17RC or its association with CMTM4 is not changed upon IL-17A stimulation (newly added Fig. 2c). The previously shown Fig. 2c is now presented as Extended Data Fig.

4a.

We have added CMTM4 staining to Fig 3a.

8. Fig 3a legend should clearly indicate the cells were transfected with SF-IL17RC. Also, why no SF-17RC level were shown for whole cell lysate?

Fig. 3a does not show cells transfected with SF-IL-17RC. We pulled down the endogenous IL-17RC using recombinant SF-tagged IL-17. Specifically, we stimulated either WT or CMTM4 KO cells with recombinant SF-IL-17A and subsequently, we purified the formed signaling complex via anti-Flag immunoprecipitation. As a control, cells were first lysed and SF-IL-17A was added only post-lysis. We provide below the modified

scheme of this experiment adapted from our previous publication (Draberova et al. EMBO J. 2020 Sep 1;39(17):e104202). We adjusted the text to be clearer.

9. The authors found that IL-17RC is glycosylated and this process was CMTM4-dependent. This part of the study is very superficial, as authors did not provide the mechanism. There are several unanswered questions: What is the role of the glycosylation in this interaction? What sites on IL-17RC are N-glycosylated?; How does CMTM4 impact the glycosylation of IL17RC? -Please give detailed discussion and provide supporting evidences.

We have dedicated substantial effort to further understand the role of CMTM4 in IL-17RC glycosylation. Our data show that IL-17RC upper band corresponds to the fully glycosylated protein resistant to enzyme Endo H, which cannot cleave N-linked glycans modified by trans-Golgi-resident enzymes (Stanley, P. Cold Spring Harb Perspect Biol. 2011 Apr 1;3(4):a005199) (newly added Extended Data Fig. 4b). CMTM4-deficient cells are unable to promote the maturation of IL-17RC into this Endo H-resistant fully glycosylated form, which suggests that CMTM4 is required for the transport of IL-17RC to the Golgi compartment, where the full maturation of its N-linked glycans takes place (newly added Fig.

2f).

IL-17RC contains nine potential glycosylation motif sites defined as NxS/T. In order to assess which of these sites are crucial for IL-17RC glycosylation, we prepared constructs coding for various mutations of

IL-17RC putative glycosylation sites (N₁₁₄Q, N₁₈₂Q, N₂₀₉Q, N_{249/255/259}Q, N₃₄₅Q, N_{401/402}Q) and transfected them in ST2 WT cells. However, we observed that all these constructs are glycosylated, demonstrating that there is not just one major glycosylation site (newly added Extended Data Fig. 5a). Therefore, we prepared a series of mutants where different combinations of these glycosylation sites were mutated. We noted that the mutation of multiple potential glycosylation sites decreases the level of glycosylation. However, only the combined mutation of all nine potential glycosylation sites prevented the IL-17RC glycosylation completely (newly added Extended Data Fig. 5b). These data demonstrated that the glycosylation of IL-17RC is a relatively complex process involving a number of different sites. The fully deglycosylated IL-17RC mutant had a severe defect in plasma membrane localization (newly added Extended Data Fig. 5c), although it was still

associated with CMTM4 (newly added Extended Data Fig. 5d). These results show that binding of CMTM4 to IL-17RC is required for the transport of IL-17RC through the Golgi network, its full glycosylation and subsequent targeting to the plasma membrane.

When we considered all these data altogether, we proposed the following model. The interaction of CMTM4 with non-glycosylated or partially glycosylated IL-17RC facilitates the transport of IL-17RC to the Golgi apparatus, where the full glycosylation is accomplished and IL-17RC in complex with CMTM4 reaches the plasma membrane (newly added Extended Data Fig. 9).

10. In Fig 3D the authors showed the mRNA level for several cytokines//chemokines in WT and CMTM4 KO.

Please show the protein levels as well (e.g., ELISA).

We stimulated WT and two different CMTM4 KO ST2 cell lines with different concentrations of IL-17A and analyzed the production of CCL2, CXCL1, or IL-6 using ELISA. We confirmed that CMTM4 is crucial for the production of these proinflammatory cytokines upon IL-17A stimulation (newly added Fig. 3i). Furthermore, we show that the reconstitution of CMTM4 KO cells with CMTM4-Myc, but not empty vector, rescued this phenotype (newly added Fig. 3j).

These data complement our previous observation that the production of CCL2 protein was markedly decreased in CMTM4 KO cells upon stimulation with a broad range of IL-17A concentrations measured in individual cells by FACS. This effect could be rescued by re-expression of CMTM4 (Fig. 3e-h).

11. Please discuss why the level of CMTM4 is greatly reduced in IL-17RC KO (Ext Fig 3B).

We analyzed several IL-17RC KO ST2 cell lines and we noticed markedly decreased CMTM4 expression in all of them (please see the figure below). These data pointed towards the mutual stabilization between CMTM4 and IL-17RC. However, when we tried to validate these data in human cell line A549, we did not observe a similar phenotype, albeit the deficiency in CMTM4 still led to drastically decreased surface levels of IL-17RC (newly added Extended Data Fig. 3g).

The most likely explanation for this discrepancy is that A549 cells express additional protein(s) that can bind and stabilize CMTM4. Since Reviewer 2 asked us to exclude this figure from the manuscript (as the role of IL-17RC in IL-17A signaling is well established), we have decided to remove these data.

Reviewer #2

(Remarks to the Author)

IL-17 signaling drives pathology in autoimmune and pathologic inflammatory conditions, and the IL-17 receptor family comprises a unique subclass of receptors that are only distantly related to better-studied cytokine receptors. As such, their fundamental biology is still surprisingly poorly understood, and therefore this topic is of considerable significance. IL-17(A) signals mainly through a heterodimer of IL-17RA and IL17RC (though a few recent studies suggest alternative receptor configurations, notably IL-17RD). IL-17RC is generally believed to be non-redundant for IL-17 function, yet its expression patterns are very different from IL-17RA, and we know very little about its regulation.

In order to identify new mediators of IL-17 signaling that interact with the IL-17 receptor, Knizkova et al. performed mass-spectrometry analysis of an overexpressed Strep-Flag-tagged IL-17R complex (SF-IL-17R) in ST2 cells, a mouse stromal cell line often utilized for signaling studies. Their data reassuringly confirm association with numerous known signaling molecules (Act1, TRAFs, etc.), which lends confidence to the approach. They also identified the tetraspanning transmembrane protein CMTM4, the subject of this paper, which was recently shown to regulate PD-L1 protein levels. They use overexpressed chimeric receptors to show that CMTM4 constitutively associates with IL-17RC through its transmembrane domain, and appears to be essential for IL-17RC cell surface expression. CMTM4 depletion in several IL-17 responsive cells (Hela, ST2, MEFs) led to impaired IL-17 signaling that correlated with decreased IL-17RC surface expression. *Cmtm4*^{-/-} mice were partially resistant to imiquimod (IMQ)-induced experimental psoriasis, a model system known to be IL-17R-dependent. Collectively, the authors propose a model whereby CMTM4 enables IL-17 signaling by permitting cell surface expression of IL-17RC.

The overall findings of this study are important to the field, novel, and logically presented. A role for CMTM4 in the IL-17 signaling is especially intriguing, and could help to explain the unresolved questions about why IL-17RA and IL-17RC are distinctly expressed despite both being required to mediate IL-17 signaling. Data are performed clearly and convincingly, with exceptions noted below. However, the manuscript has some important shortcomings in its current state. The mechanism of how CMTM4 acts to regulate IL-17RC levels are unclear and findings are often correlative – glycosylation of IL-17RC is affected in absence of CMTM4, but is that the deciding determinant of surface expression? It seems unlikely that IL-17RC glycosylation by CMTM4 is unique to IL-17RC, but the specificity of this process on other receptors (especially IL-17RA and IL-17RD) needs to be clarified. Nor is it clear whether this is regulated by IL-17 or

is a tonic event that establishes the capacity of a cell to respond to IL-17 in the first place. Furthermore, while *Cmtm4*^{-/-} mice are protected from IMQ-induced psoriasis, there are important missing controls, particularly related to how much of the effects are actually IL-17-driven events. Collectively, while CMTM4 emerges as a new component of the IL-17 receptor signaling, the underlying mechanism remains insufficiently supported.

We would like to thank the Reviewer for the appreciation of our findings and we are also glad for pointing out how to further support our experimental evidence establishing CMTM4 as a new component of the IL-17 receptor signaling machinery.

Major points

1. While CMTM4 clearly reduces IL-17RC surface expression, this may simply be due to an overall decrease in IL-17RC total protein levels, which needs to be determined. Are IL-17RC mRNA levels affected in absence of CMTM4? (this would not be predicted based on PD-L1 findings but is very easy to test; if they are affected, then that would raise a very different scenario of how IL-17RC is controlled).

We show that CMTM4 deficient cells have severely downregulated fully matured IL-17RC protein (Fig. 2 and Extended Data Fig. 4 and 5). IL-17RC mRNA levels in several ST2 CMTM4 KO clones are not significantly changed (newly added Extended Data Fig. 3d). This control was indeed missing in the original manuscript and we appreciate reviewer's comment to include it.

2. A major conclusion of this paper is that CMTM4 affects the glycosylation and by implication, the membrane expression of IL-17RC. It is unclear how this is happening at a mechanistic level. (1) No information is provided about whether N-glycosylation regulates IL-17RC protein stability/turnover/localization/internalization or some combination. The turnover rate for nonglycosylated IL-17RC compared to glycosylated IL-17RC in the presence of a protein synthesis inhibitor should be assessed. (2) the authors need to address more precisely how CMTM4 affects IL-17RC glycosylation. Since most glycosylation reactions are regulated by Golgi pH, membrane integrity and cellular stress, it is important to check whether CMTM4 regulates these events and affects glycosylation more broadly. Along these lines, a much more controlled use of inhibitors (e.g., Endo H; O-Glycosidase) could potentially provide more information about the glycosylation type. (3) is the binding of CMTM4 to IL-17RC dependent on N-glycosylation sites? It does seem that inhibition of N-glycosylation decreases the interaction between CMTM4

and IL-17RC (Fig. 2d). Therefore, the binding of CMTM4 to IL-17RC with sitedirected amino acid mutation of potential N-glycosylation site should be performed, which should be straightforward.

We believe it is unlikely that CMTM4 regulates global protein N-glycosylation substantially. In such a scenario, we would expect altered function of a huge number of glycosylated proteins manifesting with severe (likely lethal) phenotype in *Cmtm4*^{-/-} mice (Stanley, *J Mol Biol.* 2016 Aug 14; 428(16): 3166–3182). On the contrary, *Cmtm4*^{-/-} mice are viable with no apparent phenotype at the steady-state. In order to provide direct evidence whether or not CMTM4 is a general regulator of glycosylation, we expressed various SF-tagged receptors, namely IL-17RA, IL-17RB, IL-17RD, IL-17RE, and TNFR1, in either WT or CMTM4 KO cells. Subsequently we immunoprecipitated these receptors and treated or not the samples with PNGase F enzyme that removes all N-linked glycans. Our data show that all these receptors are modified by N-linked glycosylation. However, in contrast to IL-17RC, none of them interacts with CMTM4 and their glycosylation is not altered in CMTM4 KO cells (newly added Extended Data Fig. 4d-h). Furthermore, we show that although CMTM-family contains 8 members, only CMTM4 can interact with IL-17RC and rescue its glycosylation (newly added Extended Data Fig. 4c). Overall, we can conclude with a high level of certainty that the regulation of N-glycosylation by CMTM4 is specific for IL-17RC, although we cannot exclude that there are few additional proteins specifically regulated by CMTM4 in a similar manner.

We subsequently tested what type of IL-17RC glycosylation is mediated by CMTM4. We immunoprecipitated SF-IL-17RC and treated this purified protein with enzymes PNGase F that removes all N-glycans or Endo H that cleaves nonmatured N-glycans but not glycans modified by Golgi resident enzymes or O-glycosidase that removes O-linked glycans, or with a combination of PNGase F and Oglycosidase. We observed that PNGase F completely removed all glycans while O-glycosidase had no effect. Interestingly, Endo H was unable to cleave the highest molecular form of IL-17RC (newly added Extended Data Fig. 4b). This is the form that is almost undetectable in CMTM4 KO cells (newly added Fig. 2f). Altogether, our data based on the treatment of purified IL-17RC with specific enzymes revealed that CMTM4 is not required for the initial steps of glycosylation, but is crucial for the full maturation of the glycan moieties on IL-17RC in the Golgi compartment and for the transport of mature IL-17RC to the plasma membrane.

In order to establish which sites on IL-17RC are glycosylated, we expressed SF-IL-17RC protein with mutations in putative glycosylation sites defined as NxS/T (N₁₁₄Q, N₁₈₂Q, N₂₀₉Q, N_{249/255/259}Q, N₃₄₅Q, N_{401/402}Q). However, none of these mutants showed strong effect on the glycosylation level (newly added Extended Data Fig. 5a). Therefore, we subsequently prepared constructs with various combinations of mutated sites showing reduced glycosylation to a variable degree. However, only the mutation of all potential glycosylation sites prevented the glycosylation completely (newly added Extended Data Fig. 5b). The fully deglycosylated IL-17RC mutant showed strongly impaired plasma membrane localization (newly added Extended Data Fig. 5c), although it was equally capable to interact with CMTM4. This demonstrated that IL-17RC can interact with CMTM4 before it becomes fully glycosylated (newly added Extended Data Fig. 5d). Interestingly, in CMTM4 KO cells, a small portion of IL-17RC is still glycosylated and can reach the plasma membrane, while the glycosylation mutant has an even more severe defect in reaching the cell surface (newly added Extended Data Fig. 5e-f). Altogether these data show that CMTM4 promotes the trafficking of IL-17RC through the Golgi network where it is glycosylated and subsequently IL-17RC associated with CMTM4 can reach plasma membrane.

In order to provide further evidence that CMTM4 specifically regulates IL-17RC trafficking, we took advantage of our IL-17RC constructs in which the transmembrane domain is changed for the one from IL17RA, that cannot bind CMTM4. The resulting construct, termed IL-17RC^{Tm RA}, is equally localized at the cell surface in both WT and CMTM4 KO (newly added Extended Data Fig. 3h). These data demonstrate that CMTM4 is not general regulator of protein trafficking, but is specifically attuned to IL-17RC.

3. The authors performed receptor-complex analysis by stimulating cells with a recombinant Strep-Flag-IL17 (SF-IL-17). The entire signaling complex is then isolated upon cell lysis via tandem affinity purification of the ligand and analyzed by MS. However, negative controls in this experiment were not clearly articulated. First, if only unstimulated cells were used as negative controls, they are not in the same activation state as those stimulated – treating cells with standard recombinant IL-17A would be a more appropriate negative control. Second, the authors need to verify that the SF-IL-17 when added only after cell lysis does not interact with CMTM4. Third, the authors use very high doses of SF-IL-17 to stimulate cells (500 ng/ml), and they do not compare results to commercial rIL-17. This is essential. Since IL-17RC is actually thought to bind more to IL-17F than to IL-17A, they also need to include IL-17F, and ideally the IL17AF heterodimer. IL-17C is also implicated in IMQ-psoriasis (though mostly in keratinocytes) so this should also be examined.

We apologize that the proteomic assay including its controls was not described sufficiently. The negative controls were pull-downs with recombinant SF-IL-17A added post cell lysis. In this setup, SF-IL17A binds only IL-17RA, but not IL-17RC, as described previously (Kuestner RE et al. *J Immunol* 2007 179: 5462-5473; Draberova et al. *EMBO J.* 2020 Sep 1;39(17):e104202). In accord, neither IL-17RC, CMTM4 or any downstream signaling molecules were detected in control samples (Table 1). Therefore, we can confirm that CMTM4 does not bind non-specifically to either beads used for the pull-down or to SF-IL-17A ligand.

The reviewer is correct that theoretically the stimulation of cells with IL-17A might modify some proteins (in this case CMTM4) in a way that it non-specifically binds to the beads used for tandem affinity purification of IL-17RSC. We can formally exclude this scenario for CMTM4 because we confirmed the interaction between IL-17RC and CMTM4 also at the steady-state (i.e. without IL-17A stimulation). This interaction was specific, because CMTM4 did not interact with SF-IL-17RA (Fig. 1b, Extended data Fig. 1f, 4d) nor with IL-17RC chimeric protein with the transmembrane domain switched for that of IL-17RA (Fig. 1e). We also provide evidence that CMTM4 does not bind to IL-17RB, IL-17RD, IL-17RE, or TNFR1 (newly added Extended Data Fig. 1d, 4e-h).

We now included our preliminary experiments to find the optimal concentration for IL-17A stimulation. We have incubated cells with different concentrations of SF-IL-17A and showed that at the concentration 0.5 µg/ml slightly less than half of the IL-17A receptors were occupied. We also showed that at this concentration, SF-IL-17A induced strong signaling response comparable to the commercial IL-17A (obtained from R&D Systems) (newly added Extended Data Fig. 1a-c).

IL-17RC functions also as receptor for IL-17F. We confirmed that IL-17F induced signaling is markedly impaired in CMTM4 KO cells (newly added Extended Data Fig. 6e). We did not include signaling triggered by IL-17A/F heterodimer, as our technology is not suited for isolation of such heterodimer and we are not aware of any commercially available IL-17A/F heterodimer produced in eukaryotic cells, which is crucial to ensure proper glycosylation and folding of the recombinant dimer protein. Since both IL-17A- and IL-17F-induced signaling is strongly impaired in CMTM4 KO cells, it is very likely that signaling of the combined heterodimer would be affected in a comparable manner.

IL-17C employs IL-17RA and IL-17RE for signaling. We show that IL-17RE does not associate with CMTM4 (newly added Extended Data Fig. 1d) and the glycosylation of this receptor is not impacted by CMTM4 deficiency (newly added Extended Data Fig. 4g). Therefore, it is unlikely that CMTM4 would have an impact on the signaling via this cytokine.

4. A more comprehensive description of the changes in gene expression in the absence of CMTM4, with and without IL-17 stimulation, is required in these experiments (Fig 3). Moreover, how was the fold induction calculated in Fig 3d? Were all samples normalized to WT untreated cells?

We modified the manuscript to provide more clarity on this issue. The fold induction is normalized to unstimulated cells. We now also provide ELISA levels of cytokines induced upon IL-17A stimulation in WT and two different CMTM4 KO cell lines. The production of cytokines CCL2, CXCL1, and IL-6 is substantially reduced over a wide range of IL-17A concentrations in CMTM4-deficient cells (newly added Fig. 3i). Reconstitution of CMTM4 KO cells with CMTM4-Myc rescues the production of these cytokines (newly added Fig. 3j). Altogether, combined with our results from real-time PCR, our data provide compelling evidence that induction of pro-inflammatory cytokines upon IL-17A stimulation is severely impaired.

5. What is the expression of IL-17RA in the absence of CMTM4? This is a key control that is lacking in many of these experiments. In a related manner, is IL-17RD affected? (this was recently reported to be an alternative co-receptor in the IL-17R complex)

Neither IL-17RA or IL-17RD are associated with CMTM4 in unstimulated cells and the glycosylation of these proteins does not require CMTM4 (Fig. 1b and newly added Extended Data Fig. 1d, 4d, 4f). In accord, the absence of CMTM4 does not impact IL-17RA or IL-17RD expression measured by FACS (newly added Extended Data Fig. 3a).

6. The authors report that CMTM4 and IL-17RC interact constitutively, but whether this interaction is affected by IL-17 (A, F, AF) stimulation was not assessed.

Our mass-spectrometry based analysis of the stoichiometry between IL-17RC and CMTM4 showed that these two proteins interact in nearly 1:1 ratio both prior to the stimulation and also within IL-17RSC (Fig. 1c and Extended Data Fig. 1e). These data suggested that CMTM4 is constitutively associated with IL-17RC and this is not changed upon IL-17A stimulation. However, we performed several additional experiments to fully

address this issue. We first immunoprecipitated endogenous IL-17RC from ST2 cells stimulated or not with IL-17A. We demonstrated that stimulation with IL-17A did not lead to any changes in the binding between CMTM4 and IL-17RC (newly added Fig. 1d, 2c). Similarly, we stimulated ST2 cells overexpressing SF-IL-17RC with IL-17A or left them untreated and subsequently immunoprecipitated IL-17RC via anti-Flag antibody. We confirmed that the interaction between CMTM4 and SF-IL-17RC was not changed upon stimulation (newly added Extended Data Fig. 1g). Altogether, our data show that CMTM4 interacts with IL-17RC prior to the stimulation and remains associated with IL-17RC also in the context of IL-17 receptor signaling complex.

7. Cmtm4^{-/-} mice were partially, but not fully resistant to IMQ-induced experimental psoriasis, reminiscent of a recent report describing IL-17RD in this process (PMID 31175175). However, there is little to prove that this phenotype is a consequence of CMTM4 regulating IL-17 signaling as opposed to any other receptor. The authors could look at the expression of IL-17A- F- and IL-17C-dependent target genes implicated in this process, for example. Does CMTM4 regulate neutrophil infiltration during psoriasis?

In order to provide more experimental data concerning the role of CMTM4 in IMQ-induced psoriasis, we repeated the IMQ-induced psoriasis experiment and analyzed the induction of IL-17A responsive genes DefB3, S100a8 and S100a9 (Matsumoto et al. JCI Insight. 2018 Aug 9; 3(15): e121175). Induction of these genes was significantly decreased in CMTM4 deficient animals (newly added Fig. 4f).

We show that intraperitoneal injection of IL-17A leads to a massive recruitment of neutrophils and inflammatory monocytes in WT but not Cmtm4^{-/-} animals (Fig. 4A). These experiments provide direct evidence that CMTM4 is required for the cellular response to IL-17A in vivo, in accord with our data obtained from carefully controlled experiments using cell lines (Fig. 3) and also directly show that IL-17A signaling leading to the recruitment of neutrophils requires CMTM4. In accord, we provide the histology analysis of the ear showing significantly decreased thickening of the skin in CMTM4 KO mice upon IMQ treatment (newly added Fig. 4e). Because the analysis of neutrophil recruitment to IMQ-treated site would require new experimental cohorts of animals and would provide only confirmatory data (defective IL-17-dependent recruitment of neutrophils was demonstrated in the experiments with IL-17A intraperitoneal injection), we felt that the data obtained in such experiments would not justify the suffering of animals in the course of the experiment and would not comply with the rule of 3R for animal experiments.

8. It is unclear if the effects in Cmtm4^{-/-} mice arise from effects on the IL-17 target cells (in this model, keratinocytes) or alternatively could be linked to a dysfunctional hematopoietic compartment (e.g., deficiency in IL-17 producing cells such as gamma-delta T cells, ILC3 or Th17 cells). This could readily be addressed with bone marrow chimeras and followed up with specific conditional KOs. In addition, Il17a, Il17f, Il17c and Il23 mRNA levels in IMQ-treated mice of both genotypes should be evaluated to assess potential contributions for CMTM4 in IL-17-producing cells.

We agree with the reviewer that our experimental approach cannot exclude the role of CMTM4 in the hematopoietic compartment and we appreciate his/her suggestion to analyze this issue using bone marrow chimeras. We have transferred bone marrow cells from WT and *Cmtm4*^{-/-} mice to irradiated WT recipients and analyzed the progression of IMQ-induced psoriasis in these animals eight weeks later. We did not observe any difference between the mice with transplantation of either WT or CMTM4 KO bone marrow (newly added Extended Data Fig. 8c-d). These data are in a good accord with our analysis of single-cell RNAseq data showing that CMTM4 and IL-17RC are not present in immune cells (Extended Data Fig. 2)

16

and also with our data that the hematopoietic compartment is only minimally impacted in CMTM4 deficient animals (Extended Data Fig. 7).

We also provide analysis of Il-17f, Il17c, Il23 mRNA levels in IMQ treated and control mice in WT or CMTM4 KO animals (newly added Extended Data Fig. 8b). It has been previously described that IL-17A stimulates keratinocytes to produce IL-23, which can promote feed-forward amplification of the inflammation leading to aggravation of the inflammation via the IL-17 axis (Su et al. *Sci Immunol.* 2019 Jun 7;4(36):eaau9657). In accord, mice deficient in *Act1*, which is a crucial adaptor activating IL-17A signaling, have decreased IMQ-induced transcription of Il-23, Il-17a, Il-17c, Il-17f in the skin as compared to control animals (Ha et al. *PNAS* August 19, 2014 111 (33) E3422-E3431) and this effect was observed also in mice with keratinocytes-specific ablation of TRAF6, another crucial component of IL-17A signaling pathway (Matsumoto et al. *JCI Insight.* 2018 Aug 9; 3(15): e121175). We observed that CMTM4-deficient mice treated with IMQ exhibited a similar phenotype, i.e. the transcription of all these genes was mildly downregulated compared to IMQ-treated WT animals, although the differences did not reach statistical significance.

9. Immunofluorescence images in Fig. 2e-f are low resolution and do not reveal much about specific subcellular compartments. Staining should be done in both untreated and IL-17-treated cells.

We now complemented the microscopy images in the Fig. 2 with flow cytometry analysis showing that CMTM4 is required for plasma membrane localization of IL-17RC-EGFP which can be rescued by the coexpression of CMTM4-mCherry. We provide evidence that IL-17RC remains co-localized with CMTM4 upon IL-17A stimulation (newly added Extended Data Fig. 4i), which complements our biochemical results demonstrating that the stimulation of cells with IL-17A does not alter the association between these two proteins (newly added Fig. 1d, 2c and Extended Data Fig. 1g).

10. Information about CMTM4 expression in psoriasis patients would provide clinical support.

We re-analyzed two publicly available datasets containing the RNAseq data of the psoriatic lesions, non-lesion skin, and skin from healthy patients. We identified a downregulation of both IL-17RC and CMTM4 in the psoriatic skin in comparison to both controls. This is likely reflective of the infiltration of the inflamed skin with immune cells, which do not express IL-17RC and CMTM4 (Extended Data Fig. 2b-

c). These data are in accord with the previously reported decrease of IL-17RC in the skin of psoriatic patients (Johansen et al. Br J Dermatol. 2009 Feb;160(2):319-24).

11. The chimeric receptors studies in Extended Fig 3 are clever and would be appropriate in the main text.

However, they do not demonstrate that each chimeric protein is expressed at similar levels on the cell surface, which is essential to validate conclusions

We thank the reviewer for his praise of our approach that employ chimeric receptors to study the interaction between IL-17RC and CMTM4 and we included these data in the main Figure 1.

We also include the results showing that both IL-17RC wild type and chimeric mutant with transmembrane domain changed for that of IL-17RA (termed IL-17RC^{Tm RA}) are expressed on the cell surface at similar levels (newly added Extended Data Fig. 1h). Since IL-17RC^{Tm RA} does not bind CMTM4, we aimed to find out whether the surface localization of this constructs would be changed in CMTM4 KO as compared to WT cells. We showed that IL-17RC^{Tm RA} is equally localized on cell surface in both WT and CMTM4 KO cells (newly added Extended Data Fig. 3h). These data show that the transmembrane domain of IL-17RC must be coupled with CMTM4 in order for IL-17RC to promote surface expression. Changing the transmembrane domain for the one from IL-17RA renders CMTM4 dispensable. Altogether, these data provide strong evidence that binding of IL-17RC to CMTM4 in 1:1 stoichiometry is a prerequisite for Golgi trafficking, maturation, and translocation of IL-17RC to the plasma membrane, as summarized in newly added Extended Data Fig. 9.

12. Total protein level controls of TBK1, p38, JNK, p105 are missing from the majority of western blot figures.

This is essential to include as changes in their protein expression could affect their phosphorylation.

There were no changes in the expression of these proteins in CMTM4-deficient cells. We now included these controls in the revised manuscript. We also show that the activation of these signaling intermediates upon TNF or IL-1 stimulation is not impaired in CMTM4 knockout cells (newly added Extended Data Fig. 6b-c). Altogether, these data provide further evidence that CMTM4 directly impacts the assembly of the IL-17 receptor signaling complex, which is then reflected by decreased activation of downstream IL-17A signaling pathways.

13. Authors claim that *Cmtm4*^{-/-} mice do not present major changes in their immune cell compartment at baseline. However, the percentage of B cells is increased and the percentage of CD8⁺ T cells is decreased in *Cmtm4* KO mice when compared to WT mice. Please reconcile.

We changed the text accordingly to indicate the small changes in these populations.

Minor

1. The expression levels of CMTM4 in skin of IMQ-treated WT mice compared to control mice should be evaluated.

The expression of CMTM4 mRNA in skin isolated from control and IMQ-treated animals is not significantly different (newly added Extended Data Fig. 8b).

2. Do CMTM4 and IL17RC deficient cells generated via CRISPR/Cas9 originate from a single clone? If this is the case, did the authors check multiple clones to avoid potential inter-clonal variation?

We showed that CMTM4 deficiency impacts the surface expression of IL-17RC in four different ST2 clones and we further showed that re-expression of CMTM4 in two different clones rescued this phenotype (Fig. 1 and Extended Data Fig. 3b-c). We now also provide ELISA-based analysis of cytokine production in two different ST2 CMTM4 KO clones and show that both these cell lines are impaired in the induction of CCL2, CXCL1, and IL-6 upon the stimulation with a wide range of IL-17A concentrations and again provided evidence that reconstitution of CMTM4 KO with CMTM4 can rescue this phenotype (Fig. 3i-j). We confirmed the key findings using human cell line HeLa and showed that re-expression of CMTM4 can rescue IL-17RC surface expression (newly added Extended Data Fig. 3f).

In order to provide additional evidence for our data, we newly prepared human A549 cells deficient in CMTM4 and we show that in four different CMTM4 KO cell lines the surface level of IL-17RC is markedly decreased as compared to WT cells or three control clones. In contrast, levels of IL-17RA are not significantly changed in A549 CMTM4 KO cells (newly added Extended Data Fig. 3g). Altogether, we believe that provided data are very robust in showing that CMTM4 regulates the surface level of IL-17RC and consequently downstream signaling.

3. Extended Fig 2d (PNGase) should be in main figures as this is an important point.

We moved this panel to the main figure (as Fig 2d).

4. Extended fig 3b is unnecessary -- there is no need to re-prove that IL-17RC is needed for signaling We removed this figure from the manuscript.

Knizkova et al. - Revision plan

Reviewer #2

(Remarks to the Author)

In this revision, the authors have addressed to my satisfaction their contention that CMTM4 is a novel regulator of the IL-17RC receptor, and thus is essential for conventional IL-17 signal transduction

through the IL-17RA and IL-17RC complex. This work is timely, novel, and fills an important gap in the field.

We thank the Reviewer for the appreciation of our work. We are pleased that the Reviewer is satisfied with how we revised the manuscript.

Reviewer #3

(Remarks to the Author)

IL-17RC is the specific receptor for the cytokines IL-17A and IL17F. This manuscript identified CMTM4 as a critical regulator for IL-17RC cell surface expression. CMTM4 was shown to specifically associate with IL-17RC among IL-17 receptor members (IL-17RA to IL-17RE), and specifically required for IL-17RC glycosylation. The IL-17RC glycosylation was essential for its cell surface expression (potentially due to its surface transportation). Consistently, IL-17A-induced signaling and downstream genes were dramatically reduced in CMTM4-deficient cells. Similarly in vivo, IL-17A-induced leukocyte infiltration (upon the administration of IL-17A in the peritoneum) was reduced in CMTM4-deficient mice. CMTM4deficient mice also had partially protective phenotypes in the IMQ-induced mouse psoriasis, an IL-17Aassociated autoimmune disease model.

Although the authors have addressed most of the reviewers' comments in the revised manuscript, some comments still need to be addressed or clarified.

1. While the authors demonstrated that CMTM4 was required for IL-17RC glycosylation and its cell surface localization, it is still not clear how CMTM4 promotes IL-17RC glycosylation. The authors in the Extended Data Figure 9 propose that CMTM4 is required for the transport of IL-17RC to the Golgi compartment for IL-17RC glycosylation. Can the authors provide experimental evidence for the proposal? The data by using the enzymes PNGase F and Endo H are not enough and just provide indirect suggestion. Direct evidence is helpful.

Our data show that CMTM4 interacts with IL-17RC independent of its glycosylation. Treatment of IL-17RC isolated from WT or CMTM4 KO with enzyme EndoH shows that in the absence of CMTM4, IL17RC is not modified by trans-Golgi network resident glycosyltransferases (this experiment was actually recommended by Reviewer#2, who is satisfied with how we addressed it). In accord, deglycosylated IL-17RC has a severe defect in reaching the cell surface. We also show that CMTM4 is not required for the glycosylation and plasma membrane localization of IL-17RC bearing the transmembrane domain of IL-17RA (IL-17RC^{TmA}). This documents that CMTM4 is not required for the recognition of IL-

IL-17RC glycosylation sites as substrates by glycosyltransferases. Combined, these findings provide compelling data showing that binding of CMTM4 to IL-17RC promotes transport of IL-17RC to the trans-Golgi compartment for full glycosylation and subsequent transport to the cell surface.

The original questions raised by the Reviewer#1 (point 9) were answered based on the new evidence as follows:

What is the role of the glycosylation in this interaction? Glycosylation is not required for the interaction as revealed by the glycosylation mutant (Extended data 5d)

What sites on IL-17RC are N-glycosylated? All nine N-glycosylation sites of IL-17RC were identified (Extended data Fig. 5a and 5b).

How does CMTM4 impact the glycosylation of IL-17RC? CMTM4 is not required for the recognition of the glycosylation sites by glycosyltransferases as revealed by the mutant IL-17RC^{TmA}, which is normally glycosylated and localized on the plasma membrane independently on CMTM4 (Extended data Fig 3a). Furthermore, CMTM4 is not required for the early glycosylation of WT IL-17RC, but it is essential for the full glycosylation of IL-17RC by enzymes residing in the trans-Golgi network, which render N-linked glycans resistant to EndoH treatment (Fig 2f and Extended Data Fig. 4b). Combined these data provide strong support for the model that CMTM4 binding to IL-17RC enables its transport via the trans-Golgi network and subsequently to the plasma membrane.

The original questions raised by the Reviewer#2 (point 2) were answered based on the new evidence as follows:

No information is provided about whether N-glycosylation regulates IL-17RC protein stability/turnover/localization/internalization or some combination. N-glycosylation of IL-17RC is required for its localization at the plasma membrane (Extended Data Fig. 5c and 5f).

The authors need to address more precisely how CMTM4 affects IL-17RC glycosylation. Since most glycosylation reactions are regulated by Golgi pH, membrane integrity and cellular stress, it is important to check whether CMTM4 regulates these events and affects glycosylation more broadly. Along these lines, a much more controlled use of inhibitors (e.g., Endo H; O-Glycosidase) could potentially provide more information about the glycosylation type. We showed that the regulation of IL-17RC glycosylation by CMTM4 is very specific, since the N-glycosylation of IL-17RA, IL-17RB, IL-17RD, IL-17RE, or TNFR1, which are not associated with CMTM4, is not altered in CMTM4 KO cells (Extended Data Fig. 4d-h). We used the treatments suggested by this reviewer to learn that CMTM4 does not regulate the initial glycosylation, but only the final glycosylation by enzymes residing in the trans-Golgi network (Fig. 2f and Extended Data Fig. 4b).

Is the binding of CMTM4 to IL-17RC dependent on N-glycosylation sites? The binding of CMTM4 to IL-17RC is independent of the glycosylation as the IL-17RC with mutated glycosylation sites still interacts with CMTM4 (Extended Data Fig. 5d).

We are convinced that we answered all these specific questions of Reviewer#1 and Reviewer#2 concerning glycosylation. We also believe that our model of how CMTM4 regulates the trafficking of IL-17RC to the late compartments of the secretion pathway is the only probable explanation for our data. Moreover, we are not convinced that it is very relevant for the major conclusions of our study to know where exactly IL-17RC is stuck in the secretion pathway in the absence of CMTM4.

However, if required by the editor, we can address this additional concern of Reviewer#3 by examining the colocalization of trans-Golgi markers and IL-17RC in WT and CMTM4 KO, if required.

2. While the authors argued that the data out of the drosophila S2 cells can answer the question about the direct binding of CMTM4 to IL-17RC, the interactions out of the S2 cells might still be indirectly, considering exogenously expressed CMTM4 or IL-17RC might form unknown complex in the S2 cells. It would be more convincing to use direct protein-protein interaction approaches, such as individual proteins derived from in vitro transcription and translation system, bacterial purified proteins, or yeast two hybrid system, etc.

We would like to point to the data that are currently present in the manuscript that provides very compelling evidence for direct interaction between IL-17RC and CMTM4:

- (1) We show that the transmembrane domain of IL-17RC and the first two transmembrane domains of CMTM4 are mediating the interaction. If there were a putative protein required to connect IL-17RC and CMTM4, it would have to be a transmembrane protein. However, our MS analysis of the IL-17A receptor signaling complex, which also identified known components of this complex, shows that the only transmembrane proteins present are IL17RA (which does not bind CMTM4), IL-17RC, and CMTM4. There is no other protein that could mediate such interaction.
- (2) Our MS analysis of IL-17RC interactors demonstrates that CMTM4 is by far the strongest interaction partner of IL-17RC and shows that the stoichiometry of their interaction is close to 1:1 ratio. If there would be a potential third component connecting CMTM4 and IL-17RC, we would detect it via MS. Furthermore, such protein would have to be detected also in the IL-17 receptor signaling complex. Again, no such protein was found.

(3) The interaction between IL-17RC and CMTM4 is very specific, as we did not detect binding of CMTM4 with TNFR1, IL-17RA, IL-17RB, IL-17RD, or IL-17RE. Similarly, IL-17RC is not binding any other member of the CMTM family including closely related CMTM6 which shares 38% of amino acids identity with CMTM4. If there were a potential protein connecting CMTM4 and IL-17RC, it would have to evolve to contain two highly selective interaction sites: one for CMTM4 and one for IL-17RC, while not recognizing other transmembrane receptors nor other closely related members of the CMTM family.

In *Drosophila melanogaster* S2 cells we observed strong interaction between CMTM4 and IL17RC, but not between CMTM4 and IL-17RA. Insect cells are very distant from human and murine cells and do not have IL-17 signaling components. It is difficult to imagine, that S2 cells would have a protein, that would very specifically and strongly simultaneously interact with murine IL-17RC and CMTM4, while not expressing similar proteins in endogenous form.

If the interaction of IL-17RC and CMTM4 was indirect, the following very improbable hypothetical scenario would have to occur. (i) There would have to be a hypothetical protein X that is present in the IL-17 receptor signaling complex and in the steady-state IL-17RC/CMTM4 complex at the equimolar stoichiometry with IL-17RC and CMTM4. (ii) This protein would be undetected in our MS analysis for some unknown reason, although our MS was able to detect known components of the IL17 receptor signaling complex with much lower stoichiometries. (iii) This protein would have to specifically interact with IL-17RC (and not with other IL-17 receptor family members or TNFR1) and CMTM4 (and not with other CMTM family members). (iv) The orthologue of protein X would have to be expressed in insect S2 cells, which are devoid of CMTM4, IL-17RC, and IL-17 signaling in general. (v) This *Drosophila melanogaster* orthologue would possess the interaction motifs to specifically bind both mammalian CMTM4 and IL-17RC, while it would not interact with mammalian IL-17RA. We believe that this scenario is very improbable. What would be the role of this protein X in *Drosophila melanogaster*? What would be the evolutionary pressure for this Protein X to specifically interact with mammalian IL-17RC and CMTM4, while no orthologous or closely related proteins are present in insects? Why would such a protein specifically interact only with mammalian IL-17RC, but not IL-17RA?

Using the same argumentation, it is not possible to employ yeast two-hybrid (Y2H) system to study the interaction between IL-17RC and CMTM4, since we cannot formally exclude that there is an orthologue of protein X in yeast cells. Furthermore, the conventional Y2H cannot be used to study the interaction of transmembrane proteins, as it requires the interacting proteins to be targeted to the nucleus to initiate transcription of the reporter gene (although there are some rarely used modifications of Y2H which allow this).

We think that the existence of protein X orthologue that is conserved from insects to mammals to simultaneously bind mammalian IL-17RC (but not other members of IL-17R family) and CMTM4 (but not CMTM1-3 or CMTM5-8) is extremely unlikely. However, we have no tools to formally exclude it. The interaction between IL-17RC and CMTM4 requires that both proteins are expressed in the same cellular membrane. We aimed previously to test the interaction between IL-17RC and CMTM4 mixed post-lysis. We transfected cells CMTM4-deficient with SF-IL-17RC and subjected cells to cellular lysis. Subsequently, we mixed these lysates with lysates from WT cells and we isolated SF-IL-17RC via antiFlag immunoprecipitation. In this setup, SF-IL-17RC was unable to interact with CMTM4 post lysis. As a control, SF-IL-17RC expressed in WT cells interacted very strongly with CMTM4 and was fully glycosylated:

Lysates from ST2 CMTM4 KO cells transfected with SF-IL-17RC were mixed with lysates from WT and subjected to anti-Flag immunoprecipitation. As a control, we used lysates from ST2 WT cells expressing SF-IL-17RC. Samples were analyzed by immunoblotting. Data are representative of two independent experiments.

These data establish that the interaction of CMTM4 and IL-17RC is not an artifact induced by cell lysis. However, this experiment also demonstrates that in order to interact, IL-17RC and CMTM4 must be properly folded and inserted into the membrane in the correct orientation. This is not achievable by simply mixing purified proteins together (in contrast to soluble proteins).

In order to recapitulate the binding of CMTM4 and IL-17RC using purified proteins, we would have to ensure that they are (i) inserted in the membrane vesicles having lipid composition similar to biological membranes, (ii) properly folded – this is important, especially in regard to CMTM4 which has four transmembrane domains, and (iii) in the correct orientation. The structural studies using transmembrane complexes with several subunits are notoriously difficult, as opposed to cytoplasmic and soluble proteins and it is beyond our expertise.

It could be argued that for many described and commonly accepted protein-protein interactions, it was not formally excluded that additional “protein X” is mediating their interaction. To mention one related example – the recently published interaction between PD-L1 and CMTM6 (Burr, M., et al., Nature 549, 101–105 (2017) and Mezzadra R, et al., Nature 549, 106–110 (2017)). Overall, we do not believe that any additional experiments in this direction are required to support our conclusions.

3. In term of the binding domain analysis, the authors showed the nice data that the chimeric IL-17RCTmRA (with Tm domain of IL-17RC replaced by that of IL-17RA) did not bind CMTM4. It would

be better check if Tm domain of IL-17RC alone is sufficient for its binding to CMTM4 by generating the chimeric IL-17RA-TmRC (with Tm domain of IL-17RA replaced by that of IL-17RC) or IL-17RB/RD/RE-TmRC.

We show that IL-17RC strongly interacts with CMTM4, while IL-17RA does not. IL-17RC^{TmA} mutant, in which IL-17RC transmembrane domain is changed for the one from IL-17RA, cannot interact with

CMTM4, albeit having normal surface expression. This shows that the transmembrane domain of IL17RC is required for the interaction with CMTM4.

This additionally raised experiment would show whether or not the IL-17RC transmembrane domain alone is sufficient for CMTM4 binding. There are two possible outcomes:

- (1) IL-17RA-TmRC will interact with CMTM4. In such a case, we can conclude that the IL-17RC TM is sufficient for the interaction.
- (2) IL-17RA-TmRC will not interact with CMTM4. In such a case, we can conclude that the IL-17RC TM is required but not sufficient for the interaction. This would indicate that some other parts of IL17RC might be also involved in the interaction.

In either case, the result of this experiment will not change the main message of our paper at all. Furthermore, this experiment was not proposed by Reviewer#1 and Reviewer#2 in the first round of the peer review. For these reasons, we believe that this is a kind of experiment which is not necessary.

On the other hand, this experiment is a relatively simple one, albeit time- and money-consuming. If required by the editor, we will express SF-IL17RA or SF-IL17RA^{TmC} harboring IL-17RC transmembrane domain in WT cells, isolate these proteins via immunoprecipitation and check the binding of endogenous CMTM4 by western blotting.

4. In term of the mild and not statistic significant EAE phenotype in the CMTM4 KO mice, the authors may consider the suggested Th17 adoptive transfer EAE model. The authors could also use separated housed mice for the EAE model as they concerned the microbiota effect on EAE. The authors showed clear in vitro and in vivo functions of CMTM4 in IL-17A-mediated effects. Considering partial effects of CMTM4 deficiency on psoriasis, if the authors like to claim the role of CMTM4 in IL-17-associated autoimmune diseases and CMTM4 as a potential target, it would be better they try other models, such as EAE mentioned. CIA could be another option. Otherwise, the authors need to discuss on this point that CMTM4 is essential for IL-17A signaling but may not be that critical for autoimmune diseases as CMTM4

may also target other proteins such as the reported PD-L1 or undefined ones that may compromised its effect on IL-17RC in autoimmune conditions.

We will modify the discussion to reflect the Reviewer's point that albeit CMTM4 is a major regulator of IL-17A signaling in vitro and in vivo, we cannot exclude that CMTM4 regulates PD-L1 or other undefined proteins, which might have an effect on the phenotype in the complex autoimmune models. This might compromise its effect in EAE, in which IL-17A is mainly responsible for the regulation of gut homeostasis and microbiome, which indirectly impacts EAE progression (Regen et al.

Sci Immunol. 2021 Feb 5;6(56):eaaz6563).

5. The authors claimed that in unstimulated cells CMTM4 associates with IL-17RC but not IL-17RA, but in IL-17A stimulated cells, IL-17RA, IL-17RC and CMTM4 are in the same complex (the model in Extended Data Figure 9). The authors only showed the relevant data from MS in Fig.1a, which seems not enough and needs to be confirmed by IP experiments in other relevant Figures, such as in Extended Data Figure 1g, or in Extended Data Figure 6a or 6f.

In order to signal, IL-17A must form a signaling complex containing both IL-17RA and IL-17RC. Our data show that IL-17RC is constitutively associated with CMTM4 and our MS data-based quantification shows that IL-17RC binds CMTM4 in close to 1:1 stoichiometry in unstimulated cells. The same ratio is also observed in the IL-17 receptor complex isolated via SF-IL-17A. Furthermore, we show that IL17RC is dramatically decreased from the cell surface when CMTM4 is missing, which demonstrates that IL17RC must associate with CMTM4 in order to be available for subsequent signaling upon IL-17 stimulation. Combined, these data show that CMTM4 is constitutively associated with IL-17RC on the plasma membrane and upon stimulation, they move together in the IL-17 receptor signaling complex. We did also provide validation for this conclusion via the IP/western blotting experiment in Fig. 3a. We stimulated WT and CMTM4 KO cells with SF-IL-17A and isolated the formed IL-17 receptor signaling complex via anti-Flag immunoprecipitation. Indeed, this experiment confirmed that CMTM4 is part IL-17 receptor signaling complex (as requested by Reviewer#1).

We are not completely sure what interaction(s) is(are) questioned by this additional point of Reviewer#3 (no similar question was raised by Reviewer#1 or #2 in the first round of peer-review). Our evidence for the existence/non-existence of the individual interactions is described below:

(1) CMTM4 constitutively interacts with IL-17RC. This is demonstrated by the immunoprecipitation of both endogenous and Flag-tagged IL-17RC. We identified the constitutive interaction between IL-17RC and CMTM4 using MS detection (Extended Data Fig. 1e) and also by

western blotting in multiple experiments (Fig 1b, 1d, 1e, 1f, 2c, 2d, 2e, 2f and Extended Data Fig. 1f, 1g, 4a, 4c, 5d).

(2) CMTM4 does not constitutively interact with IL-17RA. This is shown by immunoprecipitation of IL-17RA followed by western blotting experiments (Fig. 1a, 1e, and Extended Data Fig. 1f, 1d).

(3) CMTM4 is a part of the IL-17 receptor signaling complex. This was shown by the isolation of the IL-17 receptor signaling complex using SF-IL-17A ligand as bait, followed by the detection of proteins recruited to this complex via MS (Fig. 1a) and western blotting (Fig. 3a). Moreover, the interaction between IL-17RC and CMTM4 was not altered upon IL-17A stimulation (Fig. 1d and Extended Data Fig. 1g), demonstrating that CMTM4 is a part of the IL-17 receptor signaling complex along with IL-17RC.

(4) IL-17RA and IL-17RC are parts of the IL-17 receptor signaling complex. It is true that we are showing that both IL-17A receptor subunits are present in the IL-17 receptor signaling complex only in the IP followed by MS (Fig 1a). However, IL-17RA and IL-17RC are parts of the IL-17A receptor signaling complex by definition and this has been established and shown many times and described in many reviews (for example McGeachy, M., et al. *Immunity*. 892–906 (2019)).

Overall, we employed both MS and western blotting-based analysis of our IP experiments to provide substantial evidence that CMTM4 interacts with IL-17RC, but not IL-17RA, at the steady-state, and CMTM4 is a part of the IL-17A receptor signaling complex. In addition, we provide data documenting that CMTM4 is important for the proper assembly of the IL-17A receptor complex since ACT1 and TRAF6 recruitment to this complex is severely compromised in CMTM4 KO cells (Fig. 3a, Extended Data 6a, 6f), which subsequently leads to strongly reduced activation of downstream signaling and cytokine production. All these data are in good accord and provide strong support for our model. For these reasons, we do not feel that additional control experiments would provide further support for our conclusions.

If required by the editor, we can provide additional control stainings showing that CMTM4 is recruited to IL-17 receptor signaling complex also in CMTM4 KO cells reconstituted with CMTM4-Myc (Extended data Fig. 6a) or in HeLa (Extended data Fig. 6f) and we can show that IL-17RC co-precipitates with IL-17RA upon IL-17 stimulation (Extended data Fig. 1g).

6. It appears the authors did not address if CMTM4 affects total IL-17RC protein expression, while they showed it did not affect IL-17RC mRNA level. In Fig. 2c, the total IL-17RC protein level seems to be dramatically reduced in CMTM4 KO lysate. Similarly in Fig. 2f, Flag-IL-17RC was lower in the KO cells. But in Fig. 2e and 2g, the SF-IL-17RC protein levels seem not much reduced in CMTM4 KO cell lysate. Are these due to experimental variations? If yes, the representative data need to be shown. If not, that means that CMTM4 affects total IL-17RC protein levels, and what is the mechanism? Does it affect IL-17RC mRNA translation or IL-17RC protein stability (turnover)?

Our data show that in the absence of CMTM4, IL-17RC is not fully glycosylated and arrested in the secretion pathway, unable to reach the trans-Golgi network. It is well expected that IL-17RC stuck somewhere in the secretory compartment has a faster degradation rate than properly posttranslationally modified and properly localized IL-17RC. This effect is more apparent on the level of the endogenous protein than in the case of the exogenous expression of flagged IL-17RC, probably because of the overexpression of the latter. This explains the difference between experiments shown in Fig. 2c (endogenous protein, similar data are also shown in Extended Data Fig. 4a) and Fig. 2e, g (both exogenous IL-17RC). The relatively small differences between Fig. 2f and 2e, g (all exogenous IL17RC) are caused by the random experiment variation and the semi-quantitative nature of western blotting. This can be documented by showing data from additional experiments.

It is difficult for us to imagine how would CMTM4 regulate the translation of IL-17RC. This would be a very unlikely co-incidence that a transmembrane protein interacting with IL-17RC and regulating its maturation and trafficking would at the same time, somehow very specifically regulate the translation of IL-17RC which occurs on ribosomes.

If required by the editor, we can further address this issue by using inhibitors of protein degradation, which should normalize the levels of IL-17RC in WT and CMTM4 KO cells.

7. In Extended Data Figure 8e, p-values need to be shown.

We will include the p-values in this experiment.

Decision Letter, first revision:

Subject: Decision on Nature Immunology submission NI-LE32255B

Message: 11th Apr 2022

Dear Dr. Draber,

Thank you for your response to the reviewers' comments on your Letter, "CMTM4 is a subunit of the IL-17 receptor mediating autoimmune pathology". Although we are interested in the possibility of publishing your study in Nature Immunology, the issues raised by the referees need to be addressed.

Please revise along the lines specified in your letter and as discussed. At resubmission, please include a "Response to referees" detailing, point-by-point, how you addressed each referee comment. If no action was taken to address a point, you must provide a compelling argument. This response will be sent back to the referees along with the revised manuscript.

Please include a revised version of any required reporting checklist. It will be available to the referees to aid in their evaluation. The Reporting Summary can be found here:

Please use the link below to submit your revised manuscript and related files:
[REDACTED]

We hope to receive your revised manuscript within two-three months. If you cannot send it within this time, please let us know. We will be happy to consider your revision so long as nothing similar has been accepted for publication at Nature Immunology or published elsewhere.

Nature Immunology is committed to improving transparency in authorship. As part of our efforts in this direction, we are now requesting that all authors identified as 'corresponding author' on published papers create and link their Open Researcher and Contributor Identifier (ORCID) with their account on the Manuscript Tracking System (MTS), prior to acceptance. ORCID helps the scientific community achieve unambiguous attribution of all scholarly contributions. You can create and link your ORCID from the home page of the

MTS by clicking on 'Modify my Springer Nature account'. For more information please visit please visit www.springernature.com/orcid.

Sincerely,

Ioana Visan, Ph.D.
Senior Editor
Nature Immunology

Tel: 212-726-9207
Fax: 212-696-9752
www.nature.com/ni

Reviewers' Comments:

Reviewer #2:

Remarks to the Author:

In this revision, the authors have addressed to my satisfaction their contention that CMTM4 is a novel regulator of the IL-17RC receptor, and thus is essential for conventional IL-17 signal transduction through the IL-17RA and IL-17RC complex. This work is timely, novel, and fills an important gap in the field.

Reviewer #3:

Remarks to the Author:

IL-17RC is the specific receptor for the cytokines IL-17A and IL17F. This manuscript identified CMTM4 as a critical regulator for IL-17RC cell surface expression. CMTM4 was shown to specifically associate with IL-17RC among IL-17 receptor members (IL-17RA to IL-17RE), and specifically required for IL-17RC glycosylation. The IL-17RC glycosylation was essential for its cell surface expression (potentially due to its surface transportation). Consistently, IL-17A-induced signaling and downstream genes were dramatically reduced in CMTM4-deficient cells. Similarly in vivo, IL-17A-induced leukocyte infiltration (upon the administration of IL-17A in the peritoneum) was reduced in CMTM4-deficient mice. CMTM4-deficient mice also had partially protective phenotypes in the IMQ-induced mouse psoriasis, an IL-17A-associated autoimmune disease model.

Although the authors have addressed most of the reviewers' comments in the revised manuscript, some comments still need to be addressed or clarified.

1. While the authors demonstrated that CMTM4 was required for IL-17RC glycosylation and its cell surface localization, it is still not clear how CMTM4 promotes IL-17RC glycosylation. The authors in the Extended Data Figure 9 propose that CMTM4 is required for the transport of IL-17RC to the Golgi compartment for IL-17RC glycosylation. Can the authors provide experimental evidence for the proposal? The data by using the enzymes PNGase F and Endo H are not enough and just provide indirect suggestion. Direct evidence is helpful.

2. While the authors argued that the data out of the drosophila S2 cells can answer the question about the direct binding of CMTM4 to IL-17RC, the interactions out of the S2 cells might still be indirectly, considering exogenously expressed CMTM4 or IL-17RC might form unknown complex in the S2 cells. It would be more convincing to use direct protein-protein interaction approaches, such as individual proteins derived from in vitro transcription and translation system, bacterial purified proteins, or yeast-two hybrid system, etc.

3. In term of the binding domain analysis, the authors showed the nice data that the chimeric IL-17RC-TmRA (with Tm domain of IL-17RC replaced by that of IL-17RA) did not bind CMTM4. It would be better check if Tm domain of IL-17RC alone is sufficient for its binding to CMTM4 by generating the chimeric IL-17RA-TmRC (with Tm domain of IL-17RA replaced by that of IL-17RC) or IL-17RB/RD/RE-TmRC.

4. In term of the mild and not statistic significant EAE phenotype in the CMTM4 KO mice, the authors may consider the suggested Th17 adoptive transfer EAE model. The authors could also use separated housed mice for the EAE model as they concerned the microbiota effect on EAE. The authors showed clear in vitro and in vivo functions of CMTM4 in IL-17A-mediated effects. Considering partial effects of CMTM4 deficiency on psoriasis, if the authors like to claim the role of CMTM4 in IL-17-associated autoimmune diseases and CMTM4 as a potential target, it would be better they try other models, such as EAE mentioned. CIA could be another option. Otherwise, the authors need to discuss on this point that CMTM4 is essential for IL-17A signaling but may not be that critical for autoimmune diseases as CMTM4 may also target other proteins such as the reported PD-L1 or undefined ones that may compromised its effect on IL-17RC in autoimmune conditions.

5. The authors claimed that in unstimulated cells CMTM4 associates with IL-17RC but not IL-17RA, but in IL-17A stimulated cells, IL-17RA, IL-17RC and CMTM4 are in the same complex (the model in Extended Data Figure 9). The authors only showed the relevant data from MS in Fig.1a, which seems not enough and needs to be confirmed by IP experiments in other relevant Figures, such as in Extended Data Figure 1g, or in Extended Data Figure 6a or 6f.

6. It appears the authors did not address if CMTM4 affects total IL-17RC protein expression, while they showed it did not affect IL-17RC mRNA level. In Fig. 2c, the total IL-17RC protein level seems to be dramatically reduced in CMTM4 KO lysate. Similarly in Fig. 2f, Flag-IL-17RC was lower in the KO cells. But in Fig. 2e and 2g, the SF-IL-17RC protein levels seem not much reduced in CMTM4 KO cell lysate. Are these due to experimental variations? If yes, the representative data need to be shown. If not, that means that CMTM4 affects total IL-17RC protein levels, and what is the mechanism? Does it affect IL-17RC mRNA translation or IL-17RC protein stability (turnover)?

7. In Extended Data Figure 8e, p-values need to be shown.

Author Rebuttal, first revision:

Knizkova et al. - Revision plan

Reviewer #2

(Remarks to the Author)

In this revision, the authors have addressed to my satisfaction their contention that CMTM4 is a novel regulator of the IL-17RC receptor, and thus is essential for conventional IL-17 signal transduction through the IL-17RA and IL-17RC complex. This work is timely, novel, and fills an important gap in the field.

We thank the Reviewer for the appreciation of our work. We are pleased that the Reviewer is satisfied with how we revised the manuscript.

Reviewer #3

(Remarks to the Author)

IL-17RC is the specific receptor for the cytokines IL-17A and IL17F. This manuscript identified CMTM4 as a critical regulator for IL-17RC cell surface expression. CMTM4 was shown to specifically associate with IL-17RC among IL-17 receptor members (IL-17RA to IL-17RE), and specifically required for IL-17RC glycosylation. The IL-17RC glycosylation was essential for its cell surface expression (potentially due to its surface transportation). Consistently, IL-17A-induced signaling and downstream genes were dramatically reduced in CMTM4-deficient cells. Similarly in vivo, IL-17A-induced leukocyte infiltration (upon the administration of IL-17A in the peritoneum) was reduced in CMTM4-deficient mice. CMTM4deficient mice also had partially protective phenotypes in the IMQ-induced mouse psoriasis, an IL-17Aassociated autoimmune disease model.

Although the authors have addressed most of the reviewers' comments in the revised manuscript, some comments still need to be addressed or clarified.

1. *While the authors demonstrated that CMTM4 was required for IL-17RC glycosylation and its cell surface localization, it is still not clear how CMTM4 promotes IL-17RC glycosylation. The authors in the Extended Data Figure 9 propose that CMTM4 is required for the transport of IL-17RC to the Golgi compartment for IL-17RC glycosylation. Can the authors provide experimental evidence for the proposal? The data by using the enzymes PNGase F and Endo H are not enough and just provide indirect suggestion. Direct evidence is helpful.*

Our data show that CMTM4 interacts with IL-17RC independently of its glycosylation (Fig. 3e). Treatment of IL-17RC isolated from WT or CMTM4 KO with enzyme EndoH shows that in the absence of CMTM4, IL-17RC is not modified by trans-Golgi network resident glycosyltransferases (Fig. 2f). In accord, deglycosylated IL-17RC has a severe defect in reaching the cell surface (Fig. 3d). We also show that CMTM4 is not required for the glycosylation and plasma membrane localization of IL-17RC bearing the transmembrane domain of IL-17RA (IL-17RC^{TmA}) (Extended Data Fig. 2h). This documents that CMTM4 is not required for the recognition of IL-17RC glycosylation sites as substrates by glycosyltransferases. Combined, these findings provide compelling data showing that the binding of CMTM4 to IL-17RC promotes its full glycosylation and subsequent transport to the cell surface.

We newly provide evidence that in CMTM4 KO cells IL-17RC-EGFP signal overlapped with endoplasmic reticulum marker Calnexin (newly added Extended Data Fig. 4d), which again shows that the trafficking of IL-17RC to the plasma membrane is severely restricted. We also modified the scheme to indicate that CMTM4 is required for IL-17RC transport through the secretory pathway (now present as Fig. 6).

2. While the authors argued that the data out of the drosophila S2 cells can answer the question about the direct binding of CMTM4 to IL-17RC, the interactions out of the S2 cells might still be indirectly, considering exogenously expressed CMTM4 or IL-17RC might form unknown complex in the S2 cells. It would be more convincing to use direct protein-protein interaction approaches, such as individual proteins derived from in vitro transcription and translation system, bacterial purified proteins, or yeast two hybrid system, etc.

We would like to point to the data that are currently present in the manuscript that provides very compelling evidence for direct interaction between IL-17RC and CMTM4:

- (1) We show that the transmembrane domain of IL-17RC and the first two transmembrane domains of CMTM4 are mediating the interaction (Fig. 1e-f). If there was a putative protein required to connect IL-17RC and CMTM4, it would have to be a transmembrane protein. However, our MS analysis of the IL-17A receptor signaling complex, which also identified known components of this complex, shows that the only transmembrane proteins present are IL-17RA (which does not bind CMTM4), IL-17RC, and CMTM4 (Fig. 1a). There is no other protein that could mediate such interaction.
- (2) Our MS analysis of IL-17RC interactors demonstrates that CMTM4 is by far the strongest interaction partner of IL-17RC and shows that the stoichiometry of their interaction is close to a 1:1 ratio (Extended Data Fig. 1d). If there would be a potential third component connecting CMTM4 and IL-17RC, we would detect it via MS. Again, no such protein was found.
- (3) The interaction between IL-17RC and CMTM4 is very specific, as we did not detect binding of CMTM4 with TNFR1, IL-17RA, IL-17RB, IL-17RD, or IL-17RE (Extended Data Fig. 3f-j). Similarly, IL-17RC is not binding any other member of the CMTM family including closely related CMTM6 which shares 38% of amino acid identity with CMTM4 (Extended Data Fig. 3e). If there were a potential protein connecting CMTM4 and IL-17RC, it would have to evolve to contain two highly selective interaction sites: one for CMTM4 and one for IL-17RC, while not recognizing other receptors from IL-17R family nor other closely related members of the CMTM family.

(4) In *Drosophila melanogaster* S2 cells, we observed a strong interaction between CMTM4 and IL-17RC, but not between CMTM4 and IL-17RA (Extended Data Fig. 1h). Insect cells are very distant from human and murine cells and do not have IL-17 signaling components. It is difficult to imagine, that S2 cells would have a protein, that would very specifically and strongly simultaneously interact with murine IL-17RC and CMTM4, while not expressing similar proteins in endogenous form.

If the interaction of IL-17RC and CMTM4 was indirect, the following very improbable hypothetical scenario would have to occur. (i) There would have to be a hypothetical protein X that is present in the IL-17 receptor signaling complex and in the steady-state IL-17RC/CMTM4 complex at the equimolar stoichiometry with IL-17RC and CMTM4. (ii) This protein would be undetected in our MS analysis for some unknown reason, although our MS was able to detect known components of the IL17 receptor signaling complex with much lower stoichiometries. (iii) This protein would have to specifically interact with IL-17RC (and not with other IL-17 receptor family members or TNFR1) and CMTM4 (and not with other CMTM family members). (iv) The orthologue of protein X would have to be expressed in insect S2 cells, which are devoid of CMTM4, IL-17RC, and IL-17 signaling in general. (v) This *Drosophila melanogaster* orthologue would possess the interaction motifs to specifically bind both mammalian CMTM4 and IL-17RC, while it would not interact with mammalian IL-17RA. We believe that this scenario is very improbable. What would be the role of this protein X in *Drosophila melanogaster*? What would be the evolutionary pressure for this Protein X to specifically interact with mammalian IL-17RC and CMTM4, while no orthologous or closely related proteins are present in insects? Why would such a protein specifically interact only with mammalian IL-17RC, but not mammalian IL-17RA?

Using the same argumentation, it is not possible to employ yeast two-hybrid (Y2H) system to study the interaction between IL-17RC and CMTM4, since we cannot formally exclude that there is an orthologue of protein X in yeast cells. Furthermore, the conventional Y2H cannot be used to study the interaction of transmembrane proteins, as it requires the interacting proteins to be targeted to the nucleus to initiate transcription of the reporter gene.

We think that the existence of protein X orthologue that is conserved from insects to mammals to simultaneously bind mammalian IL-17RC (but not other members of the IL-17R family) and CMTM4 (but not other members of the CMTM family) is extremely unlikely. However, we have no tools to formally exclude it. The interaction between IL-17RC and CMTM4 requires that both proteins are expressed in the same cellular membrane. We aimed previously to test the interaction between IL17RC and CMTM4 mixed post-lysis. We transfected CMTM4-deficient cells with SF-IL-17RC and subjected cells to cellular lysis. Subsequently, we mixed these lysates with lysates from WT cells and we isolated SF-IL-17RC via anti-Flag immunoprecipitation. In this setup, SF-IL-17RC was unable to interact with CMTM4 post lysis.

As a control, SF-IL-17RC expressed in WT cells interacted very strongly with CMTM4 and was fully glycosylated:

Lysates from ST2 CMTM4 KO cells transfected with SF-IL-17RC were mixed with lysates from WT and subjected to anti-Flag immunoprecipitation. As a control, we used lysates from ST2 WT cells expressing SF-IL-17RC. Samples were analyzed by immunoblotting. Data are representative of two independent experiments.

These data confirm that the interaction of CMTM4 and IL-17RC is not an artifact induced by cell lysis. However, this experiment also demonstrates that in order to interact, IL-17RC and CMTM4 must be properly folded and inserted into the membrane in the correct orientation. This is not achievable by simply mixing purified proteins together (in contrast to soluble proteins).

In order to recapitulate the binding of CMTM4 and IL-17RC using purified proteins, we would have to ensure that they are (i) inserted in the membrane vesicles having lipid composition similar to biological membranes, (ii) properly folded – this is important, especially in regard to CMTM4 which has four transmembrane domains, and (iii) in the correct orientation. The structural studies using transmembrane complexes with several subunits are notoriously difficult, as opposed to cytoplasmic and soluble proteins and it is beyond our expertise.

It could be argued that for many described and commonly accepted protein-protein interactions, it was not formally excluded that additional “protein X” is mediating their interaction. To mention one related example – the recently published interaction between PD-L1 and CMTM6 (Burr, M., et al., *Nature* **549**, 101–105 (2017) and Mezzadra R, et al., *Nature* **549**, 106–110 (2017)).

We agreed with the handling editor, Dr. Visan, to discuss this matter in the revised version of the manuscript that, albeit all the evidence is pointing towards direct interaction between CMTM4 and IL17RC, we cannot formally exclude indirect interaction.

3. In term of the binding domain analysis, the authors showed the nice data that the chimeric IL-17RCTmRA (with Tm domain of IL-17RC replaced by that of IL-17RA) did not bind CMTM4. It would be better check if Tm domain of IL-17RC alone is sufficient for its binding to CMTM4 by generating the chimeric IL-17RA-TmRC (with Tm domain of IL-17RA replaced by that of IL-17RC) or IL-17RB/RD/RE-TmRC.

We have expressed SF-tagged chimeric construct in which the transmembrane domain of IL-17RA is swapped with that of IL-17RC (IL17RA^{TmRC}-SF). Immunoprecipitation of this construct shows that it binds

CMTM4, which provides evidence that the transmembrane domain of IL-17RC is both required and sufficient for CMTM4 interaction (newly added Fig. 1e).

4. *In term of the mild and not statistic significant EAE phenotype in the CMTM4 KO mice, the authors may consider the suggested Th17 adoptive transfer EAE model. The authors could also use separated housed mice for the EAE model as they concerned the microbiota effect on EAE. The authors showed clear in vitro and in vivo functions of CMTM4 in IL-17A-mediated effects. Considering partial effects of CMTM4 deficiency on psoriasis, if the authors like to claim the role of CMTM4 in IL-17-associated autoimmune diseases and CMTM4 as a potential target, it would be better they try other models, such as EAE mentioned. CIA could be another option. Otherwise, the authors need to discuss on this point that CMTM4 is essential for IL-17A signaling but may not be that critical for autoimmune diseases as CMTM4 may also target other proteins such as the reported PD-L1 or undefined ones that may compromised its effect on IL-17RC in autoimmune conditions.*

We modified the discussion to reflect the Reviewer's point that albeit CMTM4 is a major regulator of IL-17A signaling *in vitro* and *in vivo*, we cannot exclude that CMTM4 regulates PD-L1 or other undefined proteins, which might have an effect on the phenotype in the complex autoimmune models. This might compromise its effect in EAE, in which IL-17A is mainly responsible for the regulation of gut homeostasis and microbiome, which indirectly impacts EAE progression (Regen et al.

Sci Immunol. 2021 Feb 5;6(56):eaaz6563).

5. *The authors claimed that in unstimulated cells CMTM4 associates with IL-17RC but not IL-17RA, but in IL-17A stimulated cells, IL-17RA, IL-17RC and CMTM4 are in the same complex (the model in Extended Data Figure 9). The authors only showed the relevant data from MS in Fig.1a, which seems not enough and needs to be confirmed by IP experiments in other relevant Figures, such as in Extended Data Figure 1g, or in Extended Data Figure 6a or 6f.*

In unstimulated cells, IL-17RA and IL-17RC are separated on the cell surface. Stimulation of dimeric IL-17A leads to the binding of both receptors to the IL-17A dimer and this, in turn, triggers the recruitment of downstream signaling molecules (Hu et al. J Immunol. 2010 Apr 15;184(8):4307-16).

We extended the introduction part of the manuscript to provide more clarity on this issue.

Our data show that IL-17RC is constitutively associated with CMTM4 and our MS data-based quantification shows that IL-17RC binds CMTM4 in close to 1:1 stoichiometry in unstimulated cells (Fig.

1b-c, Extended Data Fig. 1g). The same ratio is also observed in the IL-17 receptor complex isolated via SF-IL-17A (Fig. 1d). Furthermore, we show that IL-17RC is dramatically decreased from the cell surface when CMTM4 is missing, which demonstrates that IL-17RC has to associate with CMTM4 in order to be available for subsequent signaling upon IL-17 stimulation (Fig. 2a-b, Extended Data Fig. 2a-c, e-g). Combined, these data demonstrate that CMTM4 is constitutively associated with IL-17RC on the plasma membrane and upon stimulation, they move together in the IL-17 receptor signaling complex. We previously provided validation for this conclusion via the IP/western blotting experiment in Fig. 4a: we stimulated WT and CMTM4 KO cells with SF-IL-17A and isolated the formed IL-17 receptor signaling complex via anti-Flag immunoprecipitation. Indeed, this experiment confirmed that CMTM4 is part of IL-17 receptor signaling complex.

In order to further strengthen our experimental evidence, we show that in unstimulated cells, IL17RA is not associated with either IL-17RC or CMTM4. However, upon IL-17A stimulation, IL-17RA interacts with both IL-17RC and CMTM4 (newly added Extended Data Fig. 1f). Altogether, these data provide strong evidence that CMTM4 is recruited to IL-17 receptor signaling complex together with IL17RC.

6. It appears the authors did not address if CMTM4 affects total IL-17RC protein expression, while they showed it did not affect IL-17RC mRNA level. In Fig. 2c, the total IL-17RC protein level seems to be dramatically reduced in CMTM4 KO lysate. Similarly in Fig. 2f, Flag-IL-17RC was lower in the KO cells. But in Fig. 2e and 2g, the SF-IL-17RC protein levels seem not much reduced in CMTM4 KO cell lysate. Are these due to experimental variations? If yes, the representative data need to be shown. If not, that means that CMTM4 affects total IL-17RC protein levels, and what is the mechanism? Does it affect IL-17RC mRNA translation or IL-17RC protein stability (turnover)?

We have provided further clarification on this issue. We added data showing that various ST2 clones and MEFs deficient in CMTM4 have decreased protein levels of endogenous IL-17RC while the glycosylated form is not detectable (newly added Extended Data Fig. 3b and 7e). Reconstitution of CMTM4 KO ST2 cells with CMTM4-Myc leads to increased protein expression of endogenous IL-17RC that is fully glycosylated (newly added Extended Data Fig. 3c).

This effect is more apparent on the level of the endogenous protein than in the case of the exogenous expression of flagged IL-17RC, likely due to the overexpression of the latter. This explains the difference between experiments shown in Fig. 2c (endogenous protein, similar data are also shown in Extended Data Fig. 4a-b) and Fig. 2e, 3f (Fig. 3f was previously Fig. 2g) (both exogenous IL17RC). The relatively small differences between Fig. 2f and 2e and 3f (previously Fig. 2g) (all exogenous IL-17RC) are caused by the random experiment variation and the semi-quantitative nature of western blotting. However, all our experiments show that in the CMTM4 KO cells even the overexpression of IL-17RC is decreased and the fully glycosylated IL-17RC is almost absent.

Both our FACS and microscopy analysis show that in CMTM4 knockout, IL-17RC is unable to reach the plasma membrane and is stuck in the secretory pathway (Fig. 2a-b and Fig. 3f-k). Similarly, deglycosylated IL-17RC is unable to reach plasma membrane (Fig. 3d). It can be well expected that IL17RC stuck somewhere in the secretory compartment has a faster degradation rate than properly post-translationally modified and properly localized IL-17RC, as described previously (Helenius, A. and M. Aebi, Science. 2001 Mar 23;291(5512):2364-9). In accord, the treatment of both wild-type and CMTM4 deficient cells with proteasome inhibitor bortezomib leads to the accumulation of fully deglycosylated form of IL-17RC (newly added Extended data Fig. 3d).

Combined, our data show that CMTM4 promotes maturation and full glycosylation of IL-17RC and its transport to the plasma membrane. In CMTM4 KO cells, the endogenous IL-17RC is deglycosylated and degraded, instead of being transported to the cell surface.

7. In Extended Data Figure 8e, p-values need to be shown.

The p-values were included in this experiment.

Decision Letter, second revision:

Subject: Your manuscript, NI-A32255C

Message: Our ref: NI-A32255C

17th Aug 2022

Dear Dr. Draber,

Thank you for your patience as we've prepared the guidelines for final submission of your Nature Immunology manuscript, "CMTM4 is a subunit of the IL-17 receptor mediating autoimmune pathology" (NI-A32255C). Please carefully follow the step-by-step instructions provided in the attached file, and add a response in each row of the table to indicate the changes that you have made. Please also check and comment on any additional marked-up edits we have proposed within the text. Ensuring that each point is addressed will help to ensure that your revised manuscript can be swiftly handed over to our production team.

We would like to start working on your revised paper, with all of the requested files and forms, as soon as possible (preferably by August 22nd). Please get in contact with us if you anticipate delays.

When you upload your final materials, please include a point-by-point response to any remaining reviewer comments and please make sure to upload your checklist.

If you have not done so already, please alert us to any related manuscripts from your group that are under consideration or in press at other journals, or are being written up for submission to other journals (see: <https://www.nature.com/nature-portfolio/editorial-policies/plagiarism#policy-on-duplicate-publication> for details).

In recognition of the time and expertise our reviewers provide to Nature Immunology's

editorial process, we would like to formally acknowledge their contribution to the external peer review of your manuscript entitled "CMTM4 is a subunit of the IL-17 receptor mediating autoimmune pathology". For those reviewers who give their assent, we will be publishing their names alongside the published article.

Nature Immunology offers a Transparent Peer Review option for new original research manuscripts submitted after December 1st, 2019. As part of this initiative, we encourage our authors to support increased transparency into the peer review process by agreeing to have the reviewer comments, author rebuttal letters, and editorial decision letters published as a Supplementary item. When you submit your final files please clearly state in your cover letter whether or not you would like to participate in this initiative. Please note that failure to state your preference will result in delays in accepting your manuscript for publication.

Cover suggestions

As you prepare your final files we encourage you to consider whether you have any images or illustrations that may be appropriate for use on the cover of Nature Immunology.

Nature Immunology has now transitioned to a unified Rights Collection system which will allow our Author Services team to quickly and easily collect the rights and permissions required to publish your work. Approximately 10 days after your paper is formally accepted, you will receive an email in providing you with a link to complete the grant of rights. If your paper is eligible for Open Access, our Author Services team will also be in touch regarding any additional information that may be required to arrange payment for your article.

Please note that *Nature Immunology* is a Transformative Journal (TJ). Authors may publish their research with us through the traditional subscription access route or make their paper immediately open access through payment of an article-processing charge (APC). Authors will not be required to make a final decision about access to their article until it has been accepted. [Find out more about Transformative Journals](https://www.springernature.com/gp/open-research/transformative-journals).

If you have any questions about costs, Open Access requirements, or our legal forms,

please contact ASJournals@springernature.com.

Authors may need to take specific actions to achieve compliance with funder and institutional open access mandates. If your research is supported by a funder that requires immediate open access (e.g. according to Plan S principles) then you should select the gold OA route, and we will direct you to the compliant route where possible. For authors selecting the subscription publication route, the journal's standard licensing terms will need to be accepted, including self-archiving policies. Those licensing terms will supersede any other terms that the author or any third party may assert apply to any version of the manuscript.

Please use the following link for uploading these materials: [REDACTED]

Best regards,

Elle Morris
Senior Editorial Assistant
Nature Immunology
Phone: 212 726 9207
Fax: 212 696 9752
E-mail: immunology@us.nature.com

On behalf of

Ioana Visan, Ph.D.
Senior Editor
Nature Immunology

Tel: 212-726-9207
Fax: 212-696-9752
www.nature.com/ni

Reviewer #3:
Remarks to the Author:
The authors have addressed my points by new data and explanations in the revision. No further comments.

Final Decision Letter:

Subject: Decision on Nature Immunology submission NI-A32255D

Message: In reply please quote: NI-A32255D

Dear Dr. Draber,

I am delighted to accept your manuscript entitled "CMTM4 is a subunit of the IL-17 receptor and mediates autoimmune pathology" for publication in an upcoming issue of Nature Immunology.

Over the next few weeks, your paper will be copyedited to ensure that it conforms to Nature Immunology style. Once your paper is typeset, you will receive an email with a link to choose the appropriate publishing options for your paper and our Author Services team will be in touch regarding any additional information that may be required.

Please note that *Nature Immunology* is a Transformative Journal (TJ). Authors may publish their research with us through the traditional subscription access route or make their paper immediately open access through payment of an article-processing charge (APC). Authors will not be required to make a final decision about access to their article until it has been accepted. [Find out more about Transformative Journals](https://www.springernature.com/gp/open-research/transformative-journals).

Your paper will be published online soon after we receive your corrections and will appear in print in the next available issue. Content is published online weekly on Mondays and Thursdays, and the embargo is set at 16:00 London time (GMT)/11:00 am US Eastern

time (EST) on the day of publication. Now is the time to inform your Public Relations or Press Office about your paper, as they might be interested in promoting its publication. This will allow them time to prepare an accurate and satisfactory press release. Include your manuscript tracking number (NI-A32255D) and the name of the journal, which they will need when they contact our office.

About one week before your paper is published online, we shall be distributing a press release to news organizations worldwide, which may very well include details of your work. We are happy for your institution or funding agency to prepare its own press release, but it must mention the embargo date and Nature Immunology. Our Press Office will contact you closer to the time of publication, but if you or your Press Office have any enquiries in the meantime, please contact press@nature.com.

Also, if you have any spectacular or outstanding figures or graphics associated with your manuscript - though not necessarily included with your submission - we'd be delighted to consider them as candidates for our cover. Simply send an electronic version (accompanied by a hard copy) to us with a possible cover caption enclosed.

Please note that we encourage the authors to self-archive their manuscript (the accepted version before copy editing) in their institutional repository, and in their funders' archives, six months after publication. Nature Portfolio recognizes the efforts of funding bodies to increase access of the research they fund, and strongly encourages authors to participate in such efforts. For information about our editorial policy, including license agreement and author copyright, please visit www.nature.com/ni/about/ed_policies/index.html

Sincerely,

Ioana Visan, Ph.D.
Senior Editor
Nature Immunology

Tel: 212-726-9207
Fax: 212-696-9752
www.nature.com/ni